# Beyond Total Impervious Area: A New Lumped Descriptor of Basin-Wide Hydrologic Connectivity for Characterizing Urban Watersheds

Francesco Dell'Aira, and Claudio I. Meier

University of Memphis, Department of Civil Engineering. Memphis, TN 38152, United States.

*Correspondence to*: Francesco Dell'Aira (fdllaira@memphis.edu)

**Abstract.** Urbanization impacts on hydrologic response are typically indexed as a function of the fraction of total impervious area (TIA), i.e., the proportion of impervious areas in a basin. This implicitly assumes that changes in flood characteristics are somehow proportional to the extents of land-development, without considering that such impacts may vary widely depending on the location of the developed areas with respect to each other, the less-developed land patches, the stream network, and the

basin outlet. In other words, TIA is blind to the spatial arrangement of the different types of land patches within a basin, and to the nuanced ways in which runoff volumes are differentially generated over them and then subsequently retained or detained, as they are routed towards the stream network and then the outlet. To overcome such limitations, we propose a new lumped index that measures the impacts of urbanization on basin response in terms of the emerging hydrologic connectivity, defined here as the distributed property that explains the ability of any hillslope location to quickly receive and transfer runoff to the

stream network, as driven by topographically induced runoff pathways and locally affected by the different land-use/land-cover types present in a watershed. This alternative, hydrologic-connectivity-based index of urbanization (HCIU) displays sensitivity to the spatial arrangement of both fully developed as well as less developed or undeveloped patches, each with different degrees of imperviousness, roughness, and other characteristics affecting their abilities to either generate or else hydrologically retain or detain runoff, reflecting their distinct localized effects on hydrologic connectivity. The proposed HCIU

can be readily obtained in a GIS environment from easily available raster geospatial data. We found that HCIU improves the predictive power of regional equations for peak flow in three large case-study homogeneous regions, when used in place of the traditional TIA.

## 1 Introduction

The ongoing expansion in land development across many regions of the world is a major driver of alterations in the

hydrologic response of watersheds (Nirupama and Simonovic, 2007; Sillanpää and Koivusalo, 2015), with subsequent impacts on urban stream ecosystems (Walsh et al., 2005; Vietz, et al., 2016a, 2016b). Along with climate change, this trend in urbanization is expected to pose formidable challenges for water resources management in the years to come (Praskievicz & Chang, 2009; Bell et al., 2017; Zölch et al., 2017). Because developed land patches have quite different infiltration and interception capacities, as well as surface roughness characteristics, than undeveloped sectors, they have strong effects on

stormwater runoff dynamics; this is why quantifications of the level of urbanization based on a basin's impervious area are widely used in the domains of engineering hydrology and urban river ecology (Bauer et al., 2007; Roy & Shuster, 2009; Lee et al., 2018; Gong et al., 2020; Liang et al., 2022). Urbanization descriptors specified as a fraction of the total watershed area (Bell et al., 2016) are extensively adopted in a range of stormwater management (Kong et al., 2017; Sultana et al., 2020) as well as flood risk assessment and mitigation practices (Suharyanto et al., 1997; Loperfido et al., 2014; Sohn et al., 2020).

Popular applications of these metrics include the study of stormwater runoff dynamics (Meierdiercks et al., 2010; Fletcher et al., 2013; Yao et al., 2016), water quality assessments (Fletcher et al., 2014; Lee et al., 2012; Li et al., 2021), and peak flow prediction in ungauged (urban) basins (PUBs; Kennedy & Paretti, 2014; Southard, 2010), also involving urban planning and regulation (Smucker et al., 2016).

The total impervious area of a basin (TIA) has been historically the most widely adopted urbanization metric in hydrology (Shuster et al., 2005). Other representations of the level of urbanization are the directly connected impervious area (DCIA, i.e., the subset of TIA connected to the stream network through constructed drainage or other fully impervious pathways; Han & Burian, 2009; Sytsma et al., 2020) and the effective impervious area (EIA, i.e., an indirect estimate of DCIA; Boyd et al., 1993, 1994; Ebrahimian et al., 2016a, 2016b, 2018). Despite their widespread use, these methods have intrinsic limitations, as highlighted by a growing body of research (Shuster et al., 2005; Law et al., 2009; Beck et al., 2016; Bell et al., 2016). One drawback relates to their inability to explicitly account for differences in the spatial arrangements of impervious patches (Shuster et al., 2005; Beck et al., 2016; Bell et al., 2016). However, basins with similar levels of land development can exhibit distinct hydrologic behaviors, because the actual locations of the developed sectors within a watershed significantly impact surface runoff dynamics (Corbett et al., 1997; Pappas et al., 2008). Another issue with this kind of descriptors is that they are not able to capture the complex, distributed interactions between urbanized patches (with varying land-development intensities, depending on location) and undeveloped sectors with heterogenous hydrologic characteristics (Bell et al., 2016; Law et al., 2009), even though different spatial configurations will significantly affect a watershed's hydrologic response (Loperfido et al., 2014).

Despite the conceptual limitations of TIA and similar indices used to depict a watershed's impervious areas, this kind of simple descriptors are needed to characterize the degree of urbanization in lumped hydrologic models, with their inherent trade-offs between basin representation detail versus the spatial heterogeneity of the captured hydrologic processes (Hrachowitz and Clark, 2017). More recent urbanization metrics proposed by Yang et al. (2011) and Beck et al. (2016) try to incorporate information on the spatial distribution and geometric fragmentation of developed and undeveloped patches. These indices implicitly attempt to capture the degree of contiguity and interconnection of different land-use/land-cover (LULC) sectors based on their spatial density and granularity, but overlook the effects of basin relief on runoff routing. However, spatial contiguity does not fully explain the hydrologic connectivity of patches with distinct LULC types, as this property is ultimately determined by water pathways induced by topographic gradients. Following a conceptually different approach, Zhang & Shuster (2014) proposed the following two indices: 1) the average distance of impervious patches to the outlet, measured along topographic pathways, and 2) the mean number of pervious cells along those routes, regarded as a proxy for hydrologic disconnection. These metrics do consider the effects of topography but fall short in accounting for the heterogeneity across different LULC types and their varying effects on surface runoff dynamics, as they adopt a binary, simplified pervious-vs-impervious LULC classification. Undeveloped areas located downstream of urbanized land patches can mitigate their adverse hydrologic impacts to varying degrees and through different mechanisms, depending on specific factors such as soil infiltrability and vegetation type (and its effects on interception and roughness; Law et al., 2009). The fact that Zhang & Shuster's (2014) indices cannot account for this continuum of behaviors likely explains their poor correlation with simulated stormwater runoff volumes in their two case-study basins.

In summary, ongoing efforts in urban watershed characterization have brought to light the limitations of traditional impervious-area-based indicators like TIA as proxies for the effects of land development on stormwater hydrologic processes. Researchers have proposed alternative approaches that shift the focus towards hydrologic or geometric properties, considering either the degree of dispersion and granularity of patches with different LULC types, or else the interconnections of pervious and impervious patches as driven by topographic gradients. However, despite these advances, existing methods have yet to simultaneously address the synergistic impacts of topography and of the existence of a wide spectrum of LULC types with heterogeneous effects on stormwater runoff dynamics.

We propose a conceptual approach for deriving lumped urbanization metrics starting from the DEM and LULC map of a watershed, by measuring the impacts of land-development on hydrologic response in terms of its distributed effects on hydrologic connectivity, the spatially distributed property that explains the ability of any hillslope location to quickly receive and transfer runoff to the stream network, as driven by topographically induced runoff pathways and influenced by the spatial

arrangement of LULC patches with different hydrologic properties, such as roughness and infiltrability (Hooke et al., 2021). Distinct LULC patches intensify or mitigate the hydrologic response of their contributing area to varying extents, depending on the hydrologic processes that take place under specific LULC conditions (e.g., high canopy and litter interception as well as detention due to high surface roughness in densely vegetated areas, increased runoff volumes and peaks because of negligible infiltration losses as well as shorter travel times due to low roughness in urbanized sectors, enhanced retention due to higher infiltrability in undisturbed areas, etc.). Hence, basin cells with different land-use/land-cover types contribute to hydrologic connectivity in different ways. Conceptually, highly urbanized, fully impervious patches represent one extreme in the continuous spectrum of LULC potentials for generating runoff and increasing connectivity, with their smooth surfaces and absence of infiltration losses. On the other extreme instead, are forested areas and other natural LULC types with high vegetation densities, because of their ability for reducing runoff volumes and travel speeds through canopy and litter interception and temporary storage, high surface roughness, and soil infiltration.

Because it displays sensitivity to both the topographic structure and the heterogenous mosaic of LULC types of a basin, hydrologic connectivity provides a methodological framework for a conceptual, yet quantitative and comprehensive assessment of the impacts of land development, that not only considers the spatial arrangement of urban sectors with distinct land-development intensities, but also their interactions with undeveloped patches, with their range of flood-mitigating capabilities. Connectivity analyses have gained popularity in recent years within hydrology and geomorphology (Bordoni et al., 2018; Heckmann et al., 2018; Husic and Michalek, 2022; Martini et al., 2022). These methods define a connectivity index (Eq. 1) for each basin cell, that can be interpreted as a measure of its potential for affecting runoff or sediment fluxes (Hooke et al., 2021), depending on its location within the watershed. However, a map of connectivity values, with possible local peaks induced by the presence of fully developed patches, does not represent a lumped measure of the impacts of land development per se, nor does it allow for a straightforward comparison of the effects of urbanization across basins, or of different urbanization scenarios within a given watershed; thus, further conceptualizations are required.

As the traditional definition of the connectivity index only accounts for topographically induced runoff pathways (Borselli et al., 2008), additional adjustments may be needed, depending on the level of urbanization and the scope of the analysis, to also include the effects of underground stormwater drainage infrastructure, typically present in urban environments. Underground pipe flows may be regarded as an additional source of connectivity, which can alter and sometimes even reverse the connectivity induced by topography (e.g., when stormwaters are pumped against topographic gradients).

In this work, we derive a lumped metric of urbanization effects on hydrologic response, incorporating only topographically induced connectivity (i.e., neglecting any effects of underground storm sewer infrastructure), and test its performance as a predictor in regional peak-flow equations. Peak flows are among the hydrologic-response variables of greatest interest in urban flooding risk (Feng et al., 2021), and the most important for design purposes (Vogel and Castellarin, 2017). While considering the additional source of connectivity introduced by the underground drainage network would be straightforward, as explained in the Discussion, we could not account for it here, because it was impossible to obtain stormwater sewer data for the hundreds of watersheds involved in our regional scale analyses. However, for the scope of our investigation, which focuses on hydrologic response under severe flooding (peak flows with return periods from 2 to 500 years) considering only topographically induced connectivity should be acceptable. This approach allows us to capture the impacts of land development on the surface and near-surface phases of a basin's response, as well as the effects of streams and watercourses, including the artificial ditches and canals that make up the so-called major drainage system of stormwater infrastructure (i.e., excluding the underground network, also known as the minor system; Martins et al., 2017). During severe flooding, it is surface dynamics that predominantly govern hydrologic response, as the underground stormwater infrastructure's capacity is typically exceeded.

We benchmark our hydrologic-connectivity-based index of urbanization (HCIU) against the traditional fraction of TIA, by alternatively using one of these two metrics as a predictor in regional peak-flow equations for urbanized basins. Imperviousness descriptors expressed as a fraction of the total basin area (e.g., TIA, EIA, and DCIA) are still among the most

popular approaches to quantify the effects of land development in lumped hydrologic and regional models (Bell et al., 2016; Yang et al., 2023). Among these, we choose TIA as a benchmark because HCIU and TIA both condense distributed surface basin information (i.e., LULC and the topographic structure, and LULC only, respectively) into a lumped urbanization metric, making their comparison conceptually straightforward. On the other hand, EIA is an indirect estimate of the impacts of urbanization, based on retrospective analyses of concurrent historic flow and precipitation data for the case-study watersheds (Ebrahimian et al., 2016b). In preliminary tests, we found much uncertainty with EIA values, possibly due to the challenges involved in reliably estimating precipitation depths across basins with varying sizes and, for the same watershed, across distinct storm events (depending, e.g., on the areal footprint and location of the storm relative to basin extent). We also discarded DCIA, as its estimation would require knowing the configuration of the stormwater sewer network for each case-study basin, which was unfeasible, as mentioned above.

In the next section, we explain in detail our conceptual methodology for deriving HCIU. We then test this novel lumped urbanization metric against TIA as a predictor of peak flows in regional equations, for three case-study regions with urbanized watersheds, described in Section 3. We conclude by showing and discussing the performance of HCIU as compared to TIA (in Sections 4 and 5, respectively), highlighting strengths and weaknesses of the proposed conceptual framework, and outlining possible future research directions.

## 2 Methodology

Of the many types of connectivity indices proposed in the literature (see, e.g., Bracken et al., 2013, for a comprehensive review), we adopt a formulation (Eq. 1) first introduced by Borselli et al. (2008), which measures the potential hydrologic connectivity based on a weighted topographic analysis (Bracken et al., 2013). We consider two alternatives for the weights, as a function of either the Manning's surface roughness coefficient $n$ (Eq. 2) or else the Curve Number $CN$ (Eq. 3) of each basin pixel. Conceptually, both $n$ and $CN$ are distributed basin properties that consistently vary across LULC types with different surface runoff dynamics, e.g., due to their distinct water retention/detention capacity or infiltrability.

Below, we recall established formulations based on Eq. (1), including recommendations for the weighting factors. These methods provide a measure of the hydrologic connectivity at each basin cell, resulting in a connectivity map for the watershed. We then propose hydrologically driven criteria to obtain a lumped, hydrologic-connectivity-based index of urbanization (HCIU), able to summarize the effects of the spatial arrangement of the varied LULC patches in a watershed, in terms of their distributed impacts on connectivity. Two alternative indices, $HCIU(n)$ and $HCIU(CN)$ (depending on whether $n$ or $CN$ is chosen as weighting coefficient) are derived in this work and tested against the traditional TIA as explanatory variables in predictive peak-flow equations.

### 2.1 Connectivity-Index Formulations

Borselli et al. (2008) proposed a widely used GIS-based index of connectivity to assess sediment erosion and transport, which was then modified by Cavalli et al. (2013), Persichillo et al. (2018), Zanandrea et al. (2019), Hooke et al. (2021), and Husic & Michalek (2022), among others, to focus on other basin dynamics, such as runoff generation or landslide occurrence. In general, irrespective of the formulation, computing the index of connectivity requires assigning a flow direction (by either the D8 or D-infinity algorithm; Hooke et al., 2021) and slope value $S$ to each basin cell, from the DEM, as well as a weighting coefficient $W$ that varies across formulations depending on some additional hydrologic properties of interest (e.g., potential for erosion or runoff generation).

The index of connectivity ($IC_k$) is estimated for each raster cell $k$ of the basin hillslope component (i.e., excluding cells corresponding to the stream network) as the logarithm of the ratio of the upslope ($D_{up,k}$) and downslope ($D_{dn,k}$; Fig. 1a) components, as shown in Eq. (1) (Hooke et al., 2021).

$$IC_k = \log_{10}\left(\frac{D_{up,k}}{D_{dn,k}}\right) = \log_{10}\left(\frac{\bar{W}_k \, \bar{S}_k \, \sqrt{A_k}}{\sum_{i=k}^{n_k} \frac{d_i}{W_i \, S_i}}\right) \hspace{3cm} (1)$$

The upslope component $D_{up,k}$ relates to cell $k$'s upstream area (determined from the flow direction raster), and is proportional to its length scale $\sqrt{A_k}$ (where $A_k$ is the area draining to cell $k$), its average slope $\bar{S}_k$, and its average weighting coefficient $\bar{W}_k$. On the other hand, the downslope component $D_{dn,k}$ accounts for the effects along the topographically determined flow path between cell $k$ and the stream network, obtained as the summation of runoff travel distances $d_i$

(weighted by their respective coefficient $W_i$ and slope $S_i$) across cells $1, \dots, i, \dots, n_k$, moving from cell $k$ down to the pour point where the runoff pathway eventually meets the stream network.

In the literature, the cell weighting factor $W$ depends on the type of analysis. Some examples include: 1) the RUSLE C-factor (Renard et al., 1997), i.e., a measure of the potential for erosion, adopted in sediment transport studies (Borselli et al., 2008); 2) measures of topographic roughness, often used for morphologic characterization (Cavalli and Marchi, 2008) or

175 landslide risk assessment (Husic and Michalek, 2022), and 3) quantifications of the hydrologic characteristics of different LULC types, typically expressed as a function of Manning's surface roughness coefficient $n$, to study, e.g., anthropogenic effects on landscape and sediment transport changes (Persichillo et al., 2018), landslide occurrence (Zanandrea et al., 2019), or runoff generation (Hooke et al., 2021).

In general, independently of the choice of $W$, the index of connectivity $IC_k$ (Eq. 1) is higher at cells with a larger and/or

180 steeper contributing area, reflecting that such conditions are associated to the generation of potentially larger runoff volumes, that can concentrate faster at cell $k$. Indeed, $D_{up,k}$ shows some similarity with the well-known topographic wetness index (TWI, Beven & Kirkby, 1979; Riihimäki et al., 2021), often regarded as a proxy for soil moisture (Riihimäki et al., 2021); both are proportional to the size of the upstream contributing area, but TWI only considers the local (at-the-cell) slope, instead of the average slope of the area draining to the cell. In turn, longer travel distances to the stream network, as well as runoff

paths with milder slopes, both increase the value of the downslope component $D_{dn,k}$ (in the denominator of $IC_k$), resulting in reduced connectivity values, to reflect the lower potential of distant (upstream) hillslope locations to readily contribute the upland runoff (or sediment) to the stream network.

## 2.2 Recommended weighting coefficients for deriving HCIU

Among the options discussed above, when deriving HCIU we recommend choosing $W$ values that primarily depend on

the LULC type of each basin cell, considering both developed (urbanized) and more natural (e.g., barren, croplands, forested, etc.) categories, possibly differentiating across distinct intensities of land-development and dominant vegetation types, for the developed and vegetated categories, respectively. In this way, the effects of pixels with different surface characteristics can be differentially weighted depending on their potentials for either generating and quickly transmitting surface runoff (e.g., in the case of developed cells) or else retaining, detaining, or infiltrating water (e.g., in the case of cells with vegetated land cover),

depending on the distinct hydrologic dynamics associated with different LULC types. In turn, this incorporates sensitivity to the presence and spatial arrangement of the wide spectrum of LULC patches, from natural to fully developed. Specifically, land development will locally increase connectivity at those locations that receive runoff water from urbanized pixels upstream, as well as those sectors that contribute runoff volumes to the stream network through impervious water pathways, proportionally to the intensity of land development; in a similar way, cells with more natural LULC types, characterized by a

lower range for the weights (i.e., reflecting higher capacity for interception, detention, retention, infiltration, or some other runoff-mitigation mechanism), will locally decrease connectivity.

A candidate LULC-sensitive expression for the weighting coefficient $W$ is the $n$-dependent function given by Eq. (2) (Persichillo et al., 2018; Zanandrea et al., 2019; Hooke et al., 2021), which assigns larger weights to urbanized cells, with small

roughness coefficients, as compared to vegetated (undisturbed) cells, which typically display greater roughness (Liu & De
Smedt, 2004; Hooke et al., 2021).

$$W(n) = 1 - n \qquad (2)$$

Another candidate could be a weighting specified as a function of the Curve Number (CN; Rallison, 1980), which accounts
for LULC and soil characteristics with synergistic effects on surface runoff dynamics. In this work, we propose the expression
$W(CN)$ given by Eq. (3).

$$W(CN) = CN/100 \qquad (3)$$

In what follows, we assess both weighting approaches (i.e., based on either $n$ or $CN$) for deriving a hydrologic-
connectivity-based index of urbanization, indicating the resulting metrics as either $HCIU(n)$ or $HCIU(CN)$, respectively. A
table of the Manning's surface roughness coefficients associated with different LULC types, adapted from Hooke et al. (2021)
and Liu and De Smedt (2004), is reported in Appendix B. Similarly, CN values adapted from Wu et al. (2024) are tabulated in
Appendix C, for different combinations of LULC types and hydrologic soil groups (HSGs, Rallison, 1980; Ross et al., 2018).

**2.3 From distributed connectivity to a lumped, hydrologic-connectivity-based index of urbanization $HCIU$**

We seek to obtain a meaningful, lumped basin urbanization metric that conceptually encapsulates the impacts of the spatial
arrangement of land development and other land-use/land-cover types into a single number, starting from a basin's connectivity
map. To achieve this, we propose to first determine a relative measure of the effects of urbanization on the connectivity at each
basin cell, by normalizing with respect to a benchmark. This is done by computing the ratio between the connectivity of each
cell in the actual basin and the connectivity at the same cell but for a virtual, totally developed copy of the watershed (see
Fig. 1b and 1c), with the same shape, relief, and stream network, but fully urbanized conditions. It is clear that connectivity in
this virtual, "fully paved" basin takes the highest possible value at each cell, for a given watershed's shape, topography, and
stream network. To normalize consistently positive quantities, we suggest a change to the traditional connectivity given by
Eq. (1), by computing the hydrologic connectivity index $HCI_k$ of cell $k$ simply as the ratio of the upslope over the downslope
component (Eq. 4), without considering the logarithmic transform, so that connectivity is always positive. $HCI_k$ differs from
the traditional $IC_k$ (Eq. 1) only in the scale and sign but maintains all other properties of the original formulation, including
the sensitivity to topographic characteristics and to the spatial arrangement of patches with different LULC types.

$$HCI_k = \frac{D_{up,k}}{D_{dn,k}} = \frac{\overline{W}_k \, \bar{S}_k \, \sqrt{A_k}}{\sum_{i=k}^{n_k} \frac{d_i}{W_i S_i}} \qquad (4)$$

The imposed change in land cover in the totally developed benchmark basin involves forcing a Manning roughness value $n$ of
0.02 (or a $CN$ value of 99) and a consequent $W(n)$ of 0.98 (or 0.99 in the case of $W(CN)$), at each hillslope cell. To distinguish
the weighting coefficients of actual basin cells from the corresponding ones in its virtual, fully developed copy, we indicate
the latter as $W_{imp}$ in what follows (for "impervious"). The resulting connectivity map for the totally impervious basin (Fig. 1c)
is a benchmark for the localized effects of the varied land-use/land-cover characteristics in the actual basin. The connectivity
values for the totally developed watershed represent a theoretical upper limit for the level of connectivity that could be achieved
at each pixel in the study watershed, for its given, fixed shape, topography, and stream network. Using the proposed, non-
negative formulation of the connectivity index given by Eq. (4) ensures that the normalization is operated on non-negative
values, so that the resulting normalized variable is in the interval ]0; 1]. Eq. (5) provides the expression of the normalized
connectivity index $\widehat{HCI}_k$ for reference cell $k$ (see also Fig. 1d).

$$\widehat{HCI}_k = \frac{HCI_k}{HCI_{imp,k}} = \frac{\overline{W}_k}{\overline{W}_{k,imp}} \cdot \frac{\sum_{i=k}^{n_k} \frac{d_i}{W_{i,imp} S_i}}{\sum_{i=k}^{n_k} \frac{d_i}{W_i S_i}} = \frac{\overline{W}_k}{W_{imp}} \cdot \frac{\frac{1}{W_{imp}} \sum_{i=k}^{n_k} \frac{d_i}{S_i}}{\sum_{i=k}^{n_k} \frac{d_i}{W_i S_i}} \qquad (5)$$

In Eq. (5), $\overline{W}_{k,imp}$ and $W_{i,imp}$ both refer to the totally impervious benchmark basin, indicating the mean weighting
coefficient of the drainage area upstream of cell $k$, and the coefficient of the generic cell $i$ downstream of $k$, respectively.

Since all upstream cells will have a constant coefficient value $W_{imp}$ in the impervious basin, $\overline{W}_{k,imp}$ is equal to $W_{imp}$, for any $k$. Similarly, $W_{i,imp} = W_{imp}$, for any $i$.

The normalized connectivity index $\widehat{HCI}_k$ in Eq. (5) is equal to the product of the ratio of the average weighting coefficients from the upslope components, $\overline{W}_k / W_{imp}$, and the ratio of the weighted distances of reference cell $k$ to its pour point along the stream network, $\frac{1}{W_{imp}} \sum_{i=k}^{n_k} \frac{d_i}{S_i} / \sum_{i=k}^{n_k} \frac{d_i}{W_i S_i}$, measured in the totally impervious and the actual basin, respectively. For both factors, the numerator is always smaller than or equal to the denominator. In what follows, to emphasize the distinction between $HCI_k$ and $\widehat{HCI}_k$ whenever these two are compared, we will refer to the former as "absolute connectivity", instead of simply

"connectivity," while the latter will be termed "normalized connectivity."

        With the normalized connectivity map of a watershed, we can derive HCIU for the basin, as a weighted average of the normalized connectivity indices at each pixel. We consider a weighted average, instead of a straightforward arithmetic mean, because there is still one aspect that the connectivity index does not account for, by construction, i.e., the distance of each cell's pour point to the basin's outlet. As shown in Fig. 1a, the at-a-cell connectivity index considers the flow path from each

cell $k$ to its pour point along the stream network. However, different pour points along the network are located at varying distances from the basin outlet; hence, hillslope cells with similar levels of connectivity (e.g., because they have comparable drainage areas and characteristics, and are located at similar distances from some segment of the stream network) but at different locations with respect to the basin's main channel and its outlet, will display strong variability in their potential for quickly contributing runoff to the outlet, because of the different timing. To account for these differences, we consider, for

each hillslope cell $k$, its corresponding "along-the-stream-network" distance $d_{SN,k}$ to the outlet, measured starting from the pour point $SN_k$ of cell $k$ (see Fig. 1e). We then assign a weight $w_k$ to each hillslope cell $k$ in a way such that pixels whose pour point is closer to the outlet receive a larger weight than those which are further away from it. In this work, we propose the weighting function $f_{w,HCIU}(d_{SN,k})$ given by Eq. (6), which assigns weights in the interval [0.5; 1]. Those basin cells pouring at stream locations with the minimum distance $d_{min}$ to the outlet receive a weight of 1, while cells that drain to stream

locations at the maximum distance $d_{max}$ to the outlet get a weight of 0.5. All other basin cells with distances $d_{SN,k}$ between $d_{min}$ and $d_{max}$ receive intermediate weights that vary linearly with $d_{SN,k}$ in the interval ]0.5; 1[. It is worth noting that $d_{max}$ varies across basins, while $d_{min}$, the distance between that channel pixel adjacent to the outlet and the outlet, is clearly equal to one pixel, for any basin.

$$w_k = f_{w,HCIU}(d_{SN,k}) = 1 - 0.5 \frac{d_{SN,k} - d_{min}}{d_{max} - d_{min}} \qquad (6)$$

In Eq. (6), the weight assigned to the generic cell $k$ is indicated as $w_k$. Once a weight is assigned to each cell, depending on its "along-the-stream-network" distance $d_{SN,k}$ to the outlet, the lumped hydrologic-connectivity-based index of urbanization $HCIU$ is obtained as the weighted average of the normalized connectivity indices of each basin cell, using Eq. (7), where the normalized connectivity of the generic cell $k$ is indicated as $\widehat{HCI}_k$ (also see Fig. 1f).

$$HCIU = \frac{\sum_k w_k \widehat{HCI}_k}{\sum_k w_k} \qquad (7)$$

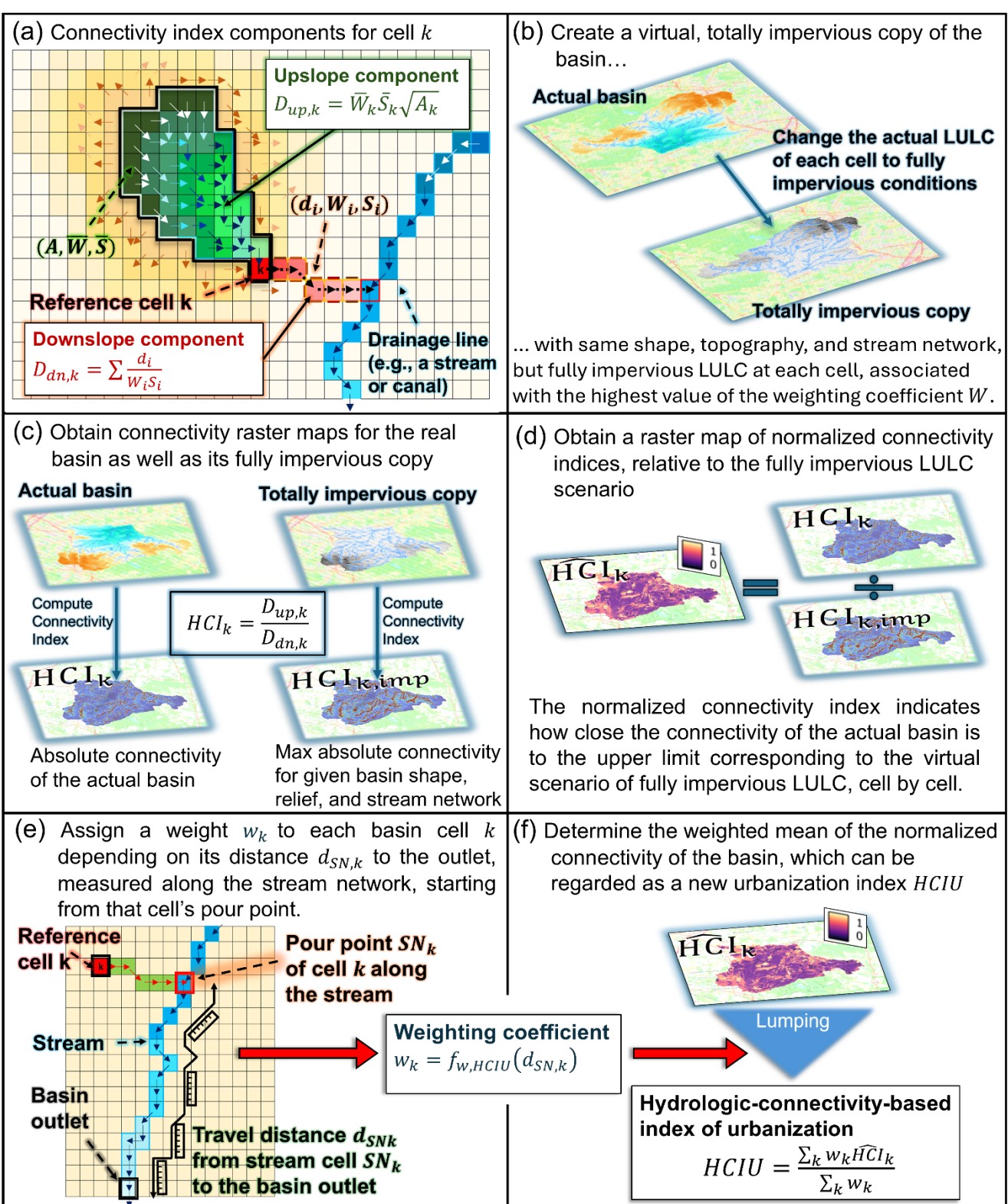

**Figure 1: Methodological steps for obtaining the hydrologic-connectivity-based index of urbanization ($HCIU$): a) scheme for calculating Borselli et al.'s (2008) connectivity index at generic cell $k$; b) create a virtual, totally impervious copy of the basin, with the same shape, topography, and stream network, but different LULC, i.e., fully developed at all cells; c) separately calculate the raster maps of connectivity for both the actual basin and its totally impervious copy; d) calculate the raster map of normalized connectivity for the basin by dividing the connectivity of the actual basin by the connectivity of the totally impervious copy, on a cell-by-cell basis; e) assign a weight $w_k$ to each basin cell $k$ depending on its distance to the outlet, as measured along the stream network, starting from the cell's pour point; f) calculate $HCIU$ as a weighted average of the normalized connectivities at each basin cell.**

In summary, the proposed methodology provides a lumped metric ($HCIU$) that is able to conceptually capture the varied hydrologic effects arising from the spatial arrangement of different LULC patches, both natural and developed, depending on their relative location with respect to each other, the stream network, and the basin outlet. First, hillslope-to-stream

connectivities, weighted depending on the hydrologic effects of distinct LULC types, are normalized with respect to a fully impervious benchmark (Fig. 1a, 1b, 1c, and 1d), which allows to compare the effects of heterogeneous levels of urbanization both across and within basins. Then, $HCIU$ is obtained as a weighted average of normalized connectivities across the entire watershed, assigning different weights to each pixel depending on the "along-the-stream-network" distance of that cell's pour point to the basin outlet (Fig. 1e and 1f).

The proposed two-step formulation – where the flow paths of hillslope cells to the pour points along the stream network and then the distances of those pour points to the basin outlet are considered separately – is different from other established, outlet-focused applications of the connectivity index, such as the $IC\_outlet$ distributed metric proposed by Cavalli et al. (2013). The latter is calculated following Borselli et al. (2008; with some adaptations to the weighting coefficient and the flow direction algorithm) but considering flow paths all the way to the outlet (hence, considering both overland flows and subsequent channelized flows within the same path), instead of flow paths to the closest stream link, following only hillslope surfaces. The two main components of a basin's hydrologic response, i.e., overland and channel flow, generally involve quite different temporal scales, because of the different orders of magnitude in roughness and water depths. The $IC\_outlet$ metric is able to capture these differences, as $IC\_outlet$ raster maps typically exhibit the highest connectivity values along the watershed stream network (comparable only to connectivities in the hillslope sectors closest to the outlet), followed by connectivities in zero-order valleys or hollows adjacent to channels (Cavalli et al., 2013). On the other hand, the focus of our methodology is on the hydrologic effects of land development, which mostly influences the surface and near-surface components of basin response by locally decreasing infiltration and increasing runoff speeds. Considering only the hillslope-to-stream connectivity in our first step allows us to enhance the method's sensitivity to the effects of land development on hydrologic response, by focusing on how runoff interacts with the distinct LULC patches encountered along the hillslope path, which control (i.e., enhance or mitigate) the connectivity. Once runoff reaches the stream network, the effects of travel distance along the stream network must still be accounted for, but this is performed in the separate, second step, considering a narrower range for the weights. This ensures that $HCIU$ displays adequate sensitivity to urbanized sectors that are adjacent to the stream network, but at reaches located far upstream from the outlet.

Breaking down the calculations for $HCIU$ in two parts (the hillslope-to-stream and then stream-to-outlet flow paths) also presents a practical advantage, particularly for large-scale implementation of the index. To ensure broad applicability of the proposed methodology, we need to be able to quickly compute $HCIU$ for any basin (in a region, country, province, state, etc.), as selected by the final user. If we were to use a "cell-to-outlet" scheme, such as the $IC\_outlet$ metric, we would need to recompute everything from scratch, every time a user chooses a different basin (i.e., a different outlet location along the stream network). Splitting the computations from cell to pour point, and then pour point to outlet, offers the opportunity to precompute "static" (i.e., independent of outlet location) raster maps of connectivity and normalized connectivity, for all the pixels over large areas. In this way, later, when a user selects a specific outlet location, the final computation of $HCIU$ only involves the much-quicker weighted averaging of the precomputed, at-a-cell, normalized connectivities, only considering those cells within the selected basin and their along-the-stream-network distances to its outlet.

## 3 Data and case studies for testing $HCIU$

### 3.1 Hydrologically homogenous regions

To test the proposed connectivity-based urbanization index $HCIU$ against the traditional TIA fraction as a predictive variable in regional peak flow equations, we considered three distinct hydrologically homogeneous regions, as determined by the U.S. Geological Survey (Southard, 2010; Austin, 2014; Feaster et al., 2014; see Fig. 2). One homogenous region encompasses all major metropolitan areas in Missouri (MO) and surroundings, including 34 urbanized watersheds (Southard, 2010). The second involves urban centers in Virginia (VA), with a total of 112 developed basins (Austin, 2014). The third

homogenous region (Feaster et al., 2014) is the largest, spanning parts of three states – Georgia, North, and South Carolina; it
includes 79 urbanized watersheds, encompassing the "Piedmont" and part of the "Ridge and Valley" ecoregions defined by
the U.S. Environmental Protection Agency (USEPA), resulting in a long band of land, moving from Georgia to North Carolina,
with consistent flood frequency characteristics (Feaster et al., 2014). For brevity, in what follows we will refer to this third
homogeneous region as EPA ecoregion (or EPAE). The VA and EPAE case studies only include basins with at least 10% of
TIA (Austin, 2014; Feaster et al., 2014), while the MO study considers a lower threshold, with all basins above 5% TIA, except
for one, with only 2.33% (Southard, 2010). TIA values for all case-study basins are reported in Appendix A.

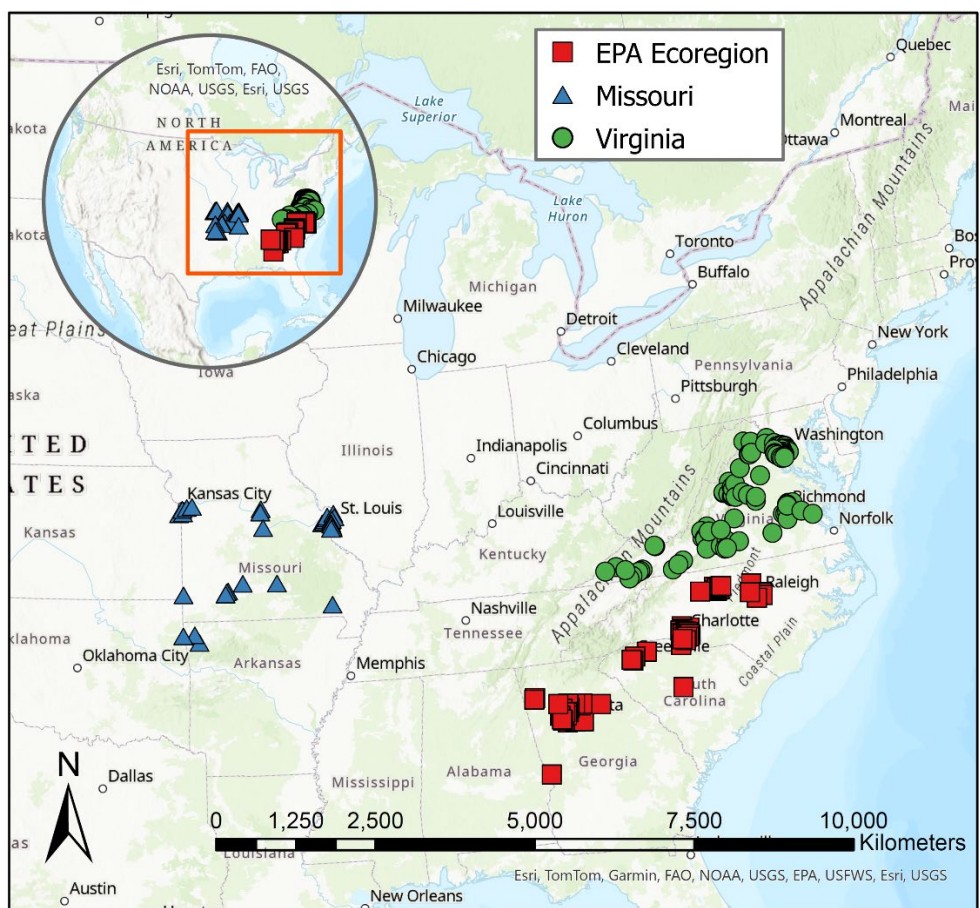

**Figure 2: Case-study regions (basin locations are reported in Appendix E; map created with ArcGIS Pro; ESRI, 2024).**

For each basin in their case-study regions, Southard (2010), Austin (2014), and Feaster et al. (2014) extracted annual
maxima series from instantaneous discharge records (typically with 15-minute or hourly temporal resolution) and performed
flood frequency analyses to estimate peak flow values for a range of return periods (see Appendix A), following the U.S.
national guidelines provided in Bulletin 17B (Interagency Advisory Committee on Water Data, 1982). Southard (2010) and
Austin (2014) obtained the fraction of TIA from the National Land Cover Dataset (NLCD) 2001 (Homer et al., 2020), for the
MO and VA case studies, respectively. On the other hand, for the EPA ecoregion Feaster et al. (2014) considered the 2006
version.

Regional peak flow equations proposed for these regions all include basin area $A$ and the percentage of TIA (simply
referred to as $TIA$ in what follows) as explanatory variables for predicting the flood $Q_T$ with return period $T$. Southard (2010)
and Feaster et al. (2014) adopted the functional form given by Eq. (8), with $TIA$ as the generic urbanization metric $U$, while
Austin (2014) considered a different form, where the peak flow per unit area ($Q_T/A$) is modeled as a function of $A$ and $TIA$.
For convenience and consistency, in this work we systematically consider the simple linear model given by Eq. (8) to test the
predictive power of the hydrologic-connectivity-based index of urbanization $HCIU$ against the traditional $TIA$. We

alternatively use $TIA$ or $HCIU$ as the generic urbanization metric $U$ in Eq. (8), each time fitting the regression model on $A$-$TIA$-$Q_T$ and $A$-$HCIU$-$Q_T$ data, respectively. The explanatory power of these competing variables would change if we considered other functional dependencies, but from a qualitative standpoint, the superiority of one variable over the other to explain peak flows should not be affected by this change.

$$\log Q_T \;=\; \beta_0 + \beta_1 \log A + \beta_2 U \tag{8}$$

Figure 3 plots all the $A$-$TIA$ pairs for the three case studies, considering basin areas in logarithmic scale. The datasets span four orders of magnitude of basin sizes, and a wide range of land-development conditions. A table with all the basins considered in the different case study regions, and related information, is provided in Appendix A.

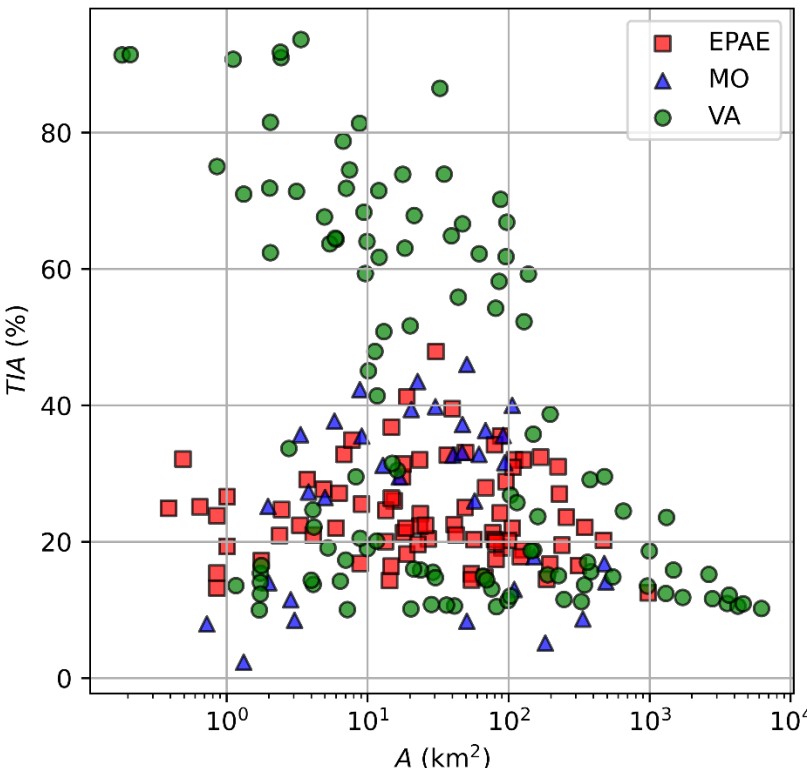

**Figure 3: Case-study basins from the EPAE, MO, and VA homogeneous regions, characterized by their area ($A$) and percentage of total impervious area ($TIA$). Areas are plotted in logarithmic scale.**

### 3.2 DEM, LULC, and HSG data

For consistency in the comparisons across the different case studies, we used the same DEM and LULC data across all homogenous regions. Because of its fine resolution and thorough coverage across the U.S. territory, we selected the 1/3rd arc-
365 second DEMs by the USGS (U.S. Geological Survey, 2023), while for LULC we adopted the same NLCD maps (Homer et al., 2020) to obtain $TIA$ as in the original studies, thus using the 2001 version for the MO (Southard, 2010) and VA (Austin, 2014) case studies, and the 2006 version for the EPAE case study (Feaster et al., 2014). The original NLCD maps do not distinguish between needleleaf and broadleaf dominant species within the forest categories, even though these two types of tree cover have very different runoff retention capabilities and should thus be modeled using different Manning roughness
coefficients (Liu and De Smedt, 2004). This information was obtained from the global, 300-m-resolution LULC maps produced by the European Space Agency (ESA, 2017), by overlapping information from the two sources. Making this distinction results in the expansion of the two original NLCD forested LULC categories "evergreen" and "deciduous" into four classes: "deciduous needleleaf", "deciduous broadleaf", "evergreen needleleaf", and "evergreen broadleaf" forest. To generate Curve-Number-based urbanization metrics $HCIU(CN)$ for our case studies, we also needed information about the

hydrologic soil group for each basin cell, for which we considered the HYSOGs250m global dataset (Ross et al., 2018), providing worldwide HSGs over a 250-meter grid.

Information from the DEM, (expanded) LULC, and HSG maps was extracted for each case-study basin using the watershed boundaries obtained from the USGS data repository by Krstolic (2006), for the VA case study, and from the StreamStats web application developed and maintained by the USGS (U.S. Geological Survey, 2019), for the MO and EPAE

case studies. Then, LULC types and LULC-HSG pairs at each basin pixel were mapped into values of Manning's roughness coefficient $n$ and Curve Number $CN$, respectively, following the tables given in Appendices B (adapted from Liu & De Smedt, 2004; Hooke et al., 2021) and C (adapted from Wu et al., 2024), respectively.

For each basin, the stream network was obtained from the flow accumulation raster (using the D8 algorithm), by setting a minimum threshold for the number of upstream cells. Instead of following the traditional procedure of considering a constant

threshold for the whole basin (Tarboton and Ames, 2001), we locally selected the threshold for each headwater to closely match its location as per the National Land Cover Dataset Plus High Resolution (NHDPlus HR; Moore et al., 2019), outlining a digital stream network as similar as possible to the official blue lines provided in that dataset. We found this approach a preferable alternative to DEM-burning procedures that would enforce the NHDPlus stream network onto the DEM by locally lowering the elevations and introducing preferential flow directions (Getirana et al., 2009), because our method is better at

preserving the actual connectivity patterns between the hillslope component and the stream network. DEM-burning algorithms, on the other hand, could have led to issues such as the occurrence of parallel streams arising from the non-alignment of the original DEM channels and the corresponding network links coming from the external source (Lindsay, 2016), with the consequent risk of diffused runoff pathway disruptions or alterations before reaching the stream.

Figure 4 shows the frequency distribution of the Manning's surface roughness coefficient $n$ (Fig. 4a, 4b, and 4c), the $CN$

(Fig. 4d, 4e, and 4f), and the slope (Fig. 4g, 4h, and 4i), observed both at the pixel and basin scales (considering basin averages in the latter case), for the three case-study regions. Based on the functional dependency of $HCI_k$ outlined by Eq. (4), the connectivity-based urbanization index $HCIU$, in its two versions $HCIU(n)$ and $HCIU(CN)$, is expected to display sensitivity to those distributed basin characteristics. We note that the VA case study presents a wider spectrum of $n$ values (Fig. 4c) as compared to the other two (Fig. 4a and 4b), which is imputable to differences in the vegetated areas, with VA more dominated

by the presence of broadleaf tree species, both deciduous and evergreen. While VA also presents a higher frequency of low CN-value pixels (Fig. 4f), as compared to the other two case studies (Fig. 4d and 4e), the difference in the spread of $CN$ values is not as noticeable as is the case with $n$. This is because, in contrast to our $n$ values, the $CN$ for different combinations of LULC types and hydrologic soil groups do not weigh broadleaf and needleleaf tree vegetated areas differently (Wu et al., 2024). The VA case study also differs from the other two in its relief characteristics, with a wider range of point and basin-

averaged slopes (Fig. 4i, as compared with Fig. 4g and 4h), associated with basins from the Appalachian region.

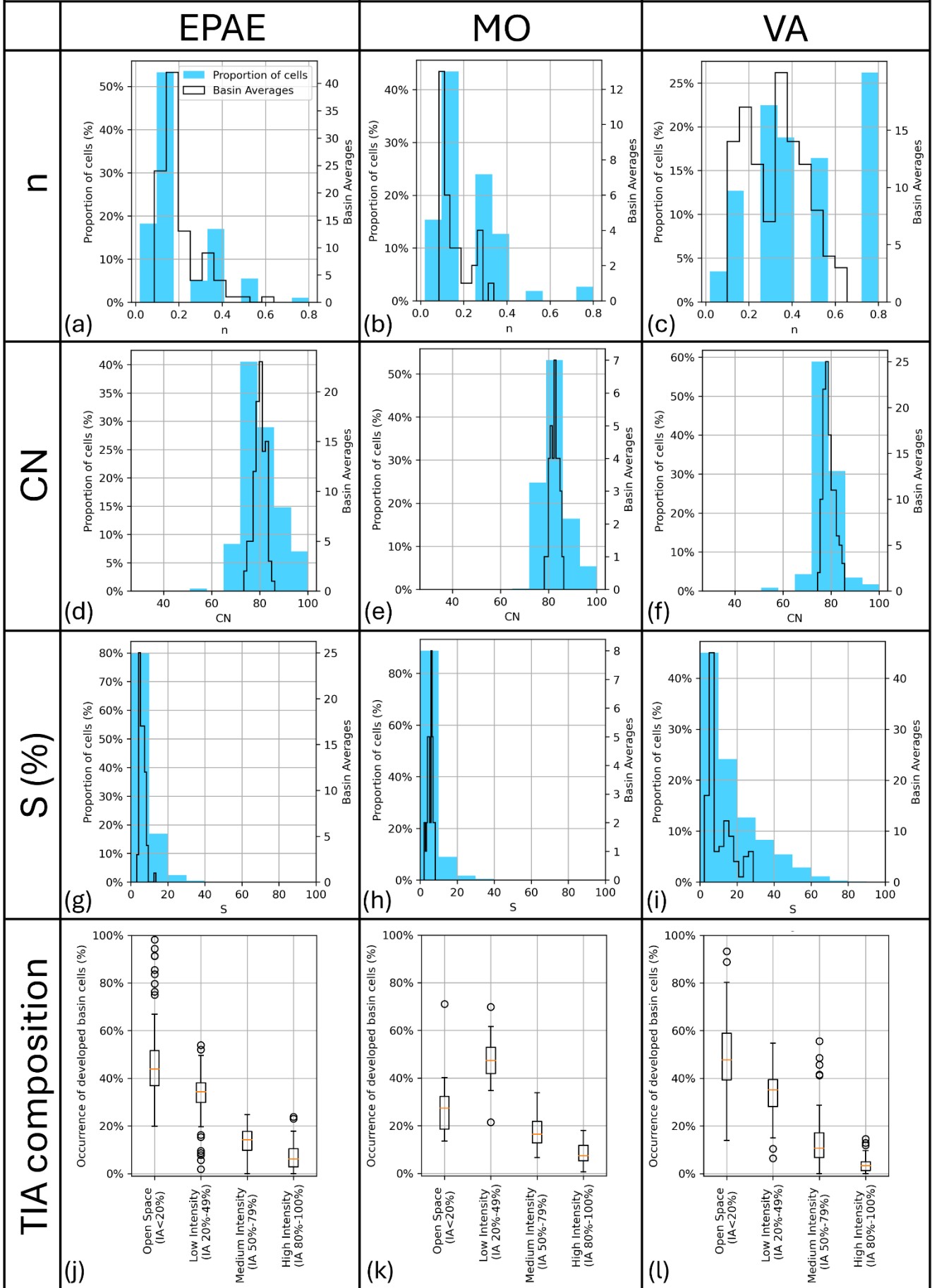

**Figure 4: Frequency distribution of the Manning surface roughness coefficient $n$ (a, b, and c), curve number $CN$ (d, e, and f), and slope $S$ (g, h, and i), for the three case studies, as observed at each basin pixel across all watersheds (filled bars) and as basin averages (empty bars); j, k, and l: mix of the four NLCD developed LULC types (associated with different amounts of impervious area IA) for all basins in the three case studies, respectively, expressed as percentages of the total extent of developed areas.**

Figure 4 also illustrates the mix of developed LULC types in the basins, by showing the distributions (boxplots) of the extents of the four developed NLCD categories in each watershed, for the three homogenous regions (Fig. 4j, 4k, and 4l, respectively). Those categories include "Developed, Open Space", "Developed, Low Intensity", "Developed, Medium Intensity", and "Developed, High Intensity", associated with ranges of impervious area of less than 20%, 20%-49%, 50%-79%, and 80% or more, respectively. For each watershed, Fig. 4 provides developed LULC extents as percentages of the number of basin pixels with given developed LULC category (say, "Low Intensity") with respect to the total number of developed LULC cells in that watershed (i.e., the sum of the numbers of "Developed, Open Space", "Low Intensity", "Medium Intensity", and "High Intensity" cells).

The four urbanized LULC categories are associated with distinct levels of imperviousness. For instance, "Open Space" is the least developed LULC category, with each pixel having less than 20% impervious area (IA), while highly urbanized cells, with impervious areas from 80% to 100%, fall in the "High Intensity" category. Because of these differences, moderately developed areas contribute less to the overall $TIA$ of a basin, as compared to highly impervious ones. In other words, larger areas of "Developed, Open Space" and "Developed, Low Intensity" patches are needed, as compared to "Developed, Medium Intensity" and "Developed, High Intensity" ones, to contribute the same proportion of $TIA$ in a watershed.

It is evident from Fig. 4j, 4k, and 4l that each region has its own characteristic mix of basin urbanized areas, even though with a clear dominance of moderately developed LULC categories (i.e., "Open Space" and "Low Intensity") over the more urbanized types, across all the case studies. VA (Fig. 4l) is the region with the smallest relative extents of highly urbanized areas (with respect to the total extent of land development), with most basins having less than 10% "High Intensity" pixels (i.e., with IA between 80% and 100%) and wider (relative) extents of "Open Space" areas (i.e., with IA less than 20%), as compared to basins from the other two case studies (Fig. 4j and 4k). MO, on the other hand (Fig. 4k), contains many watersheds with more concentrated urbanized areas, as indicated by the higher proportion of both "Medium" and "High Intensity" areas, and low-urbanization environments more dominated by "Low Intensity" (i.e., with IA between 20% and 50%), than "Open Space" areas (i.e., with IA below 20%). The EPAE region (Fig. 4j) displays intermediate conditions between those observed for VA (Fig. 4l) and MO (Fig. 4k), both in terms of the more (i.e., "Medium" and "High Intensity") and the less urbanized LULC categories (i.e., "Open Space" and "Low Intensity").

# 4 Results

## 4.1 Interpretation of the intermediate raster data products

To get a qualitative understanding of how the new connectivity-based urbanization index works and examine differences between the $n$-based and $CN$-based formulations, Fig. 5 shows the intermediate raster products generated by applying the proposed methodology at one of the MO watersheds. Basin 06894000 (Fig. 5a) is characterized by a quite heterogeneous mosaic of land-cover patches, with many parks and forested areas, as well as neighborhoods and parking lots. A zoom on a portion of the basin, in the circular window in Fig. 5a (obtained from OpenStreetMap; OpenStreetMap contributors, 2015), offers a more detailed glimpse into the variety of urbanized and vegetated areas.

Figures 5c and 5e show the $HCI$ raster maps for basin 06894000, obtained from the $n$- and $CN$-based formulations, respectively. The pairs of circular windows in Fig. 5c and 5e both depict the same enlarged portion of absolute connectivity maps obtained for the actual basin ($HCI$) and the totally developed virtual copy ($HCI_{imp}$), respectively. In all cases, absolute connectivity increases for cells located closer to the stream network. Slope also controls both formulations of $HCI$, with higher gradients of absolute connectivity observed in steeper valleys, as compared to flatter riverine zones (see Fig. 5b). This is clear when looking at the floodplain for the basin's main channel (the darker buffer area around the main channel in Fig. 5b), which is characterized by lower absolute connectivities than many of the riverine zones of its tributaries, with steeper slopes.

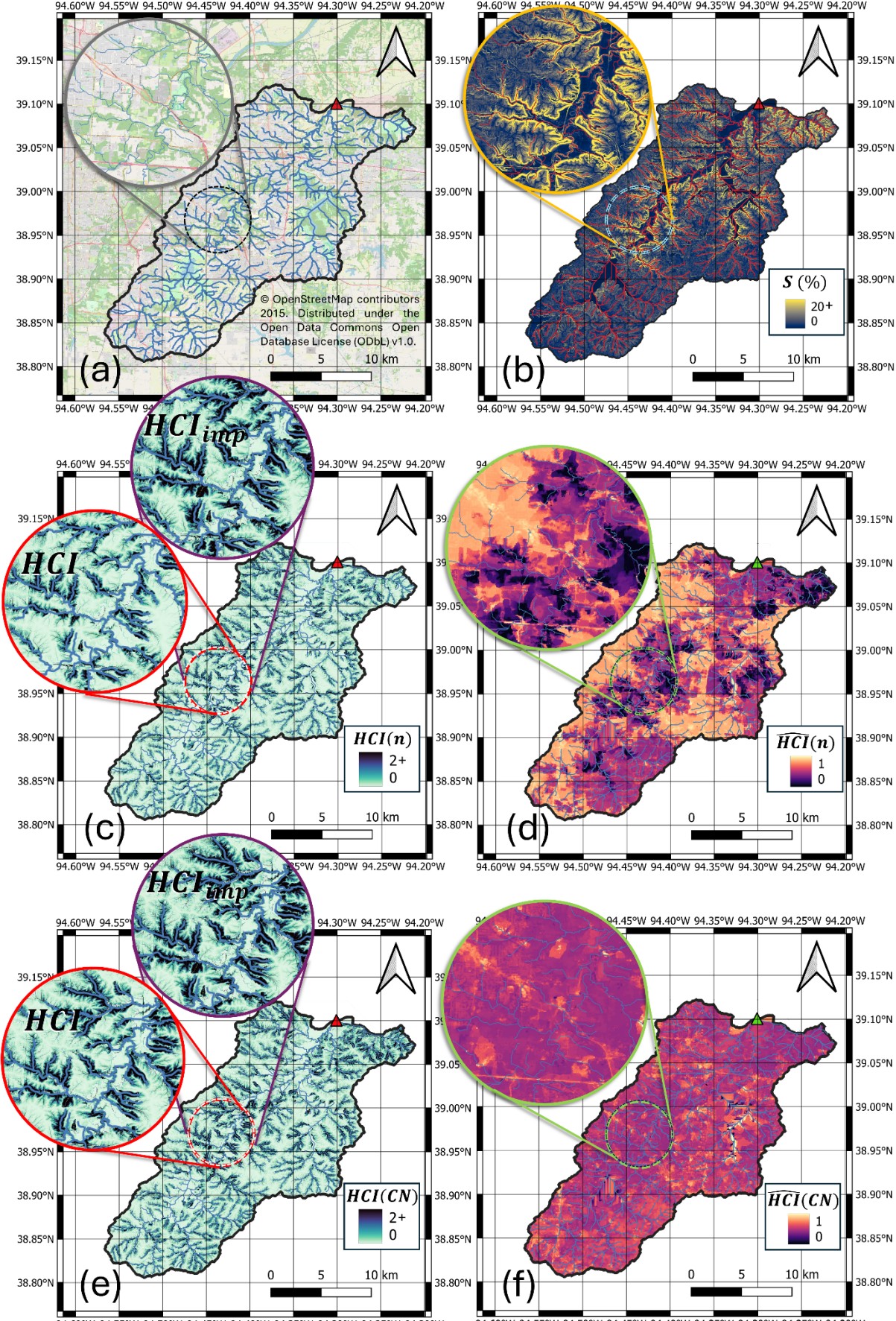

**Figure 5. a) Basin 06894000 (MO); b) raster map of slopes $S$; c) raster map of absolute hydrologic connectivity indices $HCI(n)$ obtained as a function of Manning roughness coefficient ($n$); d) raster map of normalized connectivity indices $\widehat{HCI}(n)$ obtained as a function of Manning roughness coefficient ($n$); e) raster map of absolute hydrologic connectivity indices $HCI(CN)$ obtained as a function of curve number ($CN$); f) raster map of normalized connectivity indices $\widehat{HCI}(CN)$ obtained as a function of curve number ($CN$). Background map in Fig. a) was retrieved from OpenStreetMap (OpenStreetMap contributors, 2015).**

The raster maps of normalized connectivity $\widehat{HCI}$ (Fig. 5d and 5f), on the other hand, display low sensitivity to topographic gradients, as there is no correlation with local spatial patterns in slopes (Fig. 5b). For instance, the floodplain of the main channel displays higher normalized connectivity $\widehat{HCI}(n)$ than many of the tributary valleys, even though the latter show higher absolute connectivity $HCI(n)$ (compare Fig. 5c and 5d). This is because the $n$-based normalized connectivity is most sensitive to differences in land-cover, as captured by $W(n)$ (also see Eq. 5). This is easy to notice from the $\widehat{HCI}(n)$ raster (Fig. 5d), where cells with lower normalized connectivities coarsely correspond to green patches in the map of Fig. 5a, while cells with high values of $\widehat{HCI}(n)$ are typically associated to developed areas of the basin.

While spatial patterns in $\widehat{HCI}(n)$ are easy to interpret retrospectively, because $n$ only reflects differences in land-cover types, spatial patterns in $\widehat{HCI}(CN)$ (Fig. 5f) correlate less with the spatial arrangement of land-cover types. This is expected, as $W(CN)$ reflects differences not only in LULC types, but also in hydrologic soil groups. Depending on the specific soil group, some pervious cells with lower soil infiltrability can get $CN$ values as high as those of urbanized cells (Wu et al., 2024), which may result in similar normalized connectivity values for some of the pervious patches, as compared to impervious sectors.

In summary, the two types of normalized connectivity raster maps, $\widehat{HCI}(n)$ and $\widehat{HCI}(CN)$ (Fig. 5d and 5f, respectively), may look very different from each other even though the absolute connectivity maps, $HCI(n)$ and $HCI(CN)$ (Fig. 5c and 5e, respectively), present similar spatial patterns. This is because local topography and the proximity to the stream network both are strong controls for the absolute connectivity, but not necessarily so for the normalized connectivity.

## 4.2 Conceptual differences between $HCIU$ and $TIA$

The new proposed urbanization metric expresses the relative degree of hydrologic connectivity – with respect to the fully developed benchmark – that arises from the spatial arrangement of the spectrum of developed and undisturbed LULC patches present in a basin, with different hydrologic characteristics, contingent upon the topographic structure of the hillslopes and stream network. As such, it is expected to index the effects of land development differently, as compared to $TIA$, which is simply the proportion of impervious surfaces. This means that different basins with similar percentages of $TIA$ will display variability in their associated $HCIU$ values, because of the heterogeneous effects on connectivity (as captured by $W$) of distinct spatial arrangements of the urban sectors (with different land-development intensities) and the undeveloped patches in the watershed. Figure 6 shows, in two separate subplots (6a and 6b), the variability in $HCIU$ obtained for our case studies for various levels of $TIA$, considering $HCIU(n)$ and $HCIU(CN)$, respectively. We include example basins with distinct spatial arrangements of natural and developed areas (Fig. 6c to 6i) to clearly illustrate the sensitivity of $HCIU$ to LULC configurations. For the sake of simplicity, for this visualization we adopt a simplified, four-category LULC classification: "Highly Impervious", "Moderately Impervious", "Moderately Pervious", and "Highly Pervious". We label developed pixels with impervious areas equal to or above 50% (i.e., the NLCD "Medium" and "High Intensity" classes), as "Highly Impervious", while those with imperviousness below 50% (i.e., the NLCD "Open Space" and "Low Intensity" classes) are classified as "Moderately Impervious". On the other hand, the "Moderately Pervious" category encompasses those undeveloped cells with moderate values of Manning's coefficient $n$, equal to or below 0.3 (see Appendix B), such as herbaceous, hay/pasture, and barren land LULC, while the "Highly Pervious" category covers the remaining undeveloped LULC types, including forested and other densely vegetated areas.

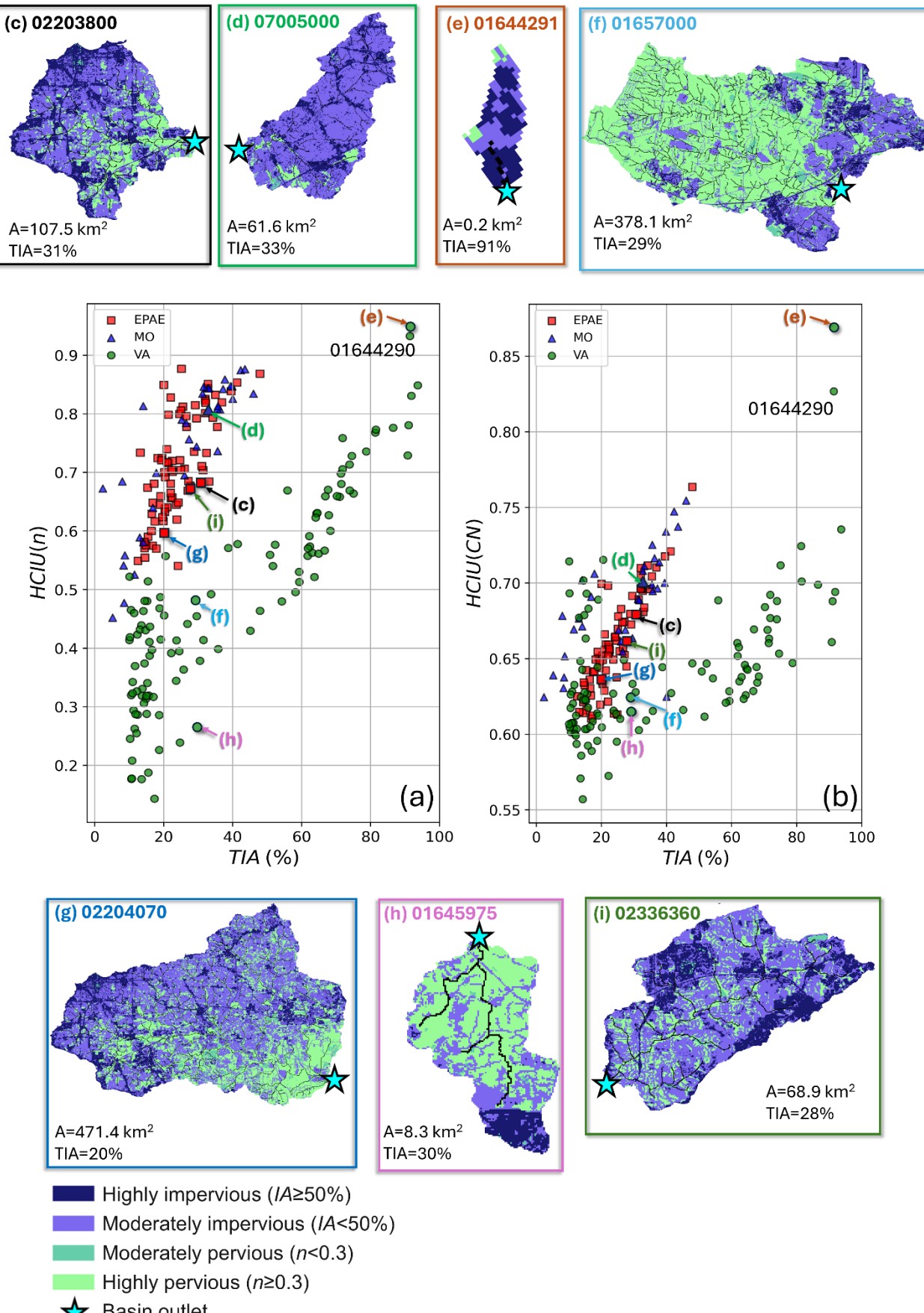

**Figure 6. Case-study basins from the EPAE, MO, and VA homogenous regions in the *TIA-HCIU* plane, considering a) *HCIU(n)* and b) *HCIU(CN)*, respectively. Seven specific basins (c–i) are used as examples to visualize the sensitivity of *HCIU* to different spatial arrangements of LULC patches, even for basins with similar *TIA*; a simplified, four-category LULC classification is adopted, as indicated in the legend.**

In general, a wide range of $HCIU$ values is associated to any given $TIA$, across all case studies; this is expected, since similar relative amounts of impervious areas can have different effects on hydrologic connectivity, depending on their spatial arrangement with respect to each other and also with respect to the less developed and undeveloped patches (Fig. 6a and 6b). However, we also note general positive trends, both within and across all the case studies considered, suggesting that basins
with increasingly larger $TIA$ are characterized by overall higher $HCIU$ values, as should also be expected.

Figures 6a and 6b also show that $HCIU(n)$ and $HCIU(CN)$ have different ranges of variability. Both have a lower bound greater than 0; this is intrinsic to the formulation of $HCIU$ itself as a (weighted) average of the relative degree of connectivity of the watershed. The absolute connectivity of a basin with totally pervious land-cover type, which can be regarded as a lower-bound, yet realistic scenario, is small (or very small), compared to the connectivity of the corresponding totally impervious
virtual basin, but not zero. Therefore, the fact that $HCIU$'s lower bound is greater than zero comes from physical (or at least conceptual) considerations, depending on the range of variability of the weighing coefficient $W$, specifically on its minimum value (see Eq. 5). Since $HCIU(n)$ and $HCIU(CN)$ use different weighting coefficients $W(n)$ and $W(CN)$, with different lower limits, their lower bounds will differ.

Figures 6a and 6b display another interesting characteristic of $HCIU$, i.e., its sensitivity to the heterogeneity in the
undeveloped LULC types in a basin. Compared to the other two case studies, VA watersheds got overall smaller $HCIU$ values for similar levels of $TIA$. This is because VA basins are characterized by pervious land-cover types with generally greater flood-mitigating capabilities (larger $n$ values and smaller $CN$ values), as indicated by the wider spread of Manning's coefficients in the range of small values (Fig. 4c), as well as the stronger positive skewness in the frequency distribution of $CN$ values (Fig. 4f), as compared to the other two case studies (Fig. 4a-4b, and 4d-4e, respectively). As a result, VA basins
have lower $HCIU$ values as compared to those from the MO and EPAE case studies with similar relative amounts of developed areas, because the pervious fraction in VA basins is more effective in reducing the hydrologic connectivity of the urbanized patches. This effect is more evident with the $n$-based formulation for $HCIU$, where a clear separation exists also for the smallest $TIA$ values. This is modulated by the diverse ways in which the effects on hydrologic connectivity of distinct undeveloped LULC types are differentially weighted by $W(n)$ and $W(CN)$.

Distinct spatial arrangements of pervious and impervious patches (see examples in Fig. 6c to 6i) get $HCIU(n)$ and $HCIU(CN)$ values (Fig. 6a and 6b, respectively) in agreement with our expectations, irrespective of the weighting approach (based on either $n$ or $CN$). For instance, focusing on watersheds with similar $TIA$ levels (around 30%), basin 01645975 (Fig. 6h), with predominant "Highly Pervious" land cover and most urbanized areas concentrated in a limited area far upstream from its outlet, gets significantly lower $HCIU$ values (both in the $HCIU(n)$ and $HCIU(CN)$ scales) than basins 01657000
(Fig. 6f) and 02203800 (Fig. 6c), both of which display spatial arrangements of impervious areas that would be expected to cause stronger hydrologic effects.

Basin 01657000 (Fig. 6f) has a proportion of pervious and impervious LULC patches similar to basin 01645975 (Fig. 6h), but its urbanized areas (including a considerable extent of "Highly Impervious" patches) are all concentrated downstream, where they can contribute impervious runoff more effectively to the outlet, hence having a stronger impact on the overall
hydrologic connectivity of the watershed. Basin 02203800 (Fig. 6c) shows even worse conditions, with a significantly larger proportion of "Moderately Impervious" areas than pervious patches, which impact a wider extent of the watershed, including areas near its outlet. These LULC conditions, even though they are associated with locally lower levels of imperviousness as compared with the highly impervious but spatially concentrated areas of basin 01657000 (Fig. 6f), systematically increase distributed connectivity, hence decreasing the overall response time of the watershed. Consistently, basin 02203800 (Fig. 6c)
gets higher $HCIU(n)$ and $HCIU(CN)$ values than basin 01657000 (Fig. 6f).

These comparisons indicate that large developed areas, even with locally low levels of imperviousness, can have stronger effects on $HCIU$, as compared to highly urbanized but spatially concentrated patches. Because of this, basins that get the highest $HCIU$ values are those where developed patches are dominant and uniformly spread. Basins 07005000 (Fig. 6d) and

01644291 (Fig. 6e), both in the higher end of the $TIA$-$HCIU$ plots (Fig. 6a and 6b), exemplify this case, with their surfaces entirely covered by a mix of "Moderately" and "Highly Impervious" land-cover, and negligible presence of "Moderately" or "Highly Pervious" patches.

The stronger effects on $HCIU$ of large extents of moderately impervious patches, as compared with highly urbanized but concentrated areas, hold true also when comparing watersheds with different $TIA$ levels. For instance, basin 02204070 (Fig. 6g) has $TIA$ of 20%, lower than the 30% $TIA$ of 01657000 (Fig. 6f) and 01645975 (Fig. 6h); yet the $HCIU$ value of the former is higher than the $HCIU$ values of the latter two, both for the $n$- and $CN$-based formulations. This is, again, because the spatial arrangement of its impervious patches covers a wider (even though less dense) spatial extent, resulting in a more widespread increase in connectivity.

The examples above indicate that $HCIU$ displays sensitivity to the spatial arrangement of the heterogeneous LULC patches in a watershed, irrespective of basin size and shape. However, $HCIU$ can also capture similarities in these spatial arrangements, across basins with different shapes and sizes. For instance, basins 02203800 (Fig. 6c) and 02336360 (Fig. 6i), with areas of 108 and 70 km$^2$, respectively, both display a preponderance of "Moderately Impervious" LULC types, with few, more intensely developed patches located far upstream with respect to the outlet, and a finely dispersed pervious fraction, that tends to condense into larger pervious patches moving towards the outlet. Because of these similarities in the LULC spatial configurations, these two basins get similar $HCIU$ values, both when using $HCIU(n)$ and $HCIU(CN)$, despite their differences in size and shape.

Two highly urbanized VA basins, 01644290 and 01644291 (Fig. 6e), have unusually higher $HCIU(CN)$ values, as compared to all others (Fig. 6b), because of their peculiar LULC conditions. Both are very small in size (< 0.20 km$^2$) and are almost totally characterized by highly impervious LULC ("Developed, High Intensity" NLCD type). Furthermore, the few, small undeveloped patches are unable to mitigate the connectivity of the developed areas, since they are located far upstream, close to the water divide (see Fig. 6e for basin 01644291). Both aspects lead to LULC conditions almost identical to those for the virtual, totally impervious scenario considered for normalizing absolute connectivity, hence justifying the large $HCIU(CN)$ values for these two watersheds, which are the highest among all basins and case studies. These two watersheds are outliers on the set of $HCIU(CN)$ values for VA (Fig. 6b), while they are the highest in the $TIA$-$HCIU(n)$ plane (Fig. 6a), but still clustered with other VA basins with high $TIA$.

## 4.3 Performance of $HCIU$ in regional peak-flow equations

We tested the predictive power of $HCIU$ as an alternative to $TIA$, to be used as urbanization metric in the development of regional peak flow equations for the three case studies. Figure 7 shows the performance of the regional model (Eq. 8) fitted on basin area $A$, $HCIU$, and a range of flood quantiles $Q_T$ (with return periods $T$ of 2, 5, 10, 25, 50, 100, and 500 years; see Appendix A), comparing it to the benchmark model with the same functional dependency, but with $TIA$ as the urbanization metric. The adjusted R$^2$ is considered as error metric.

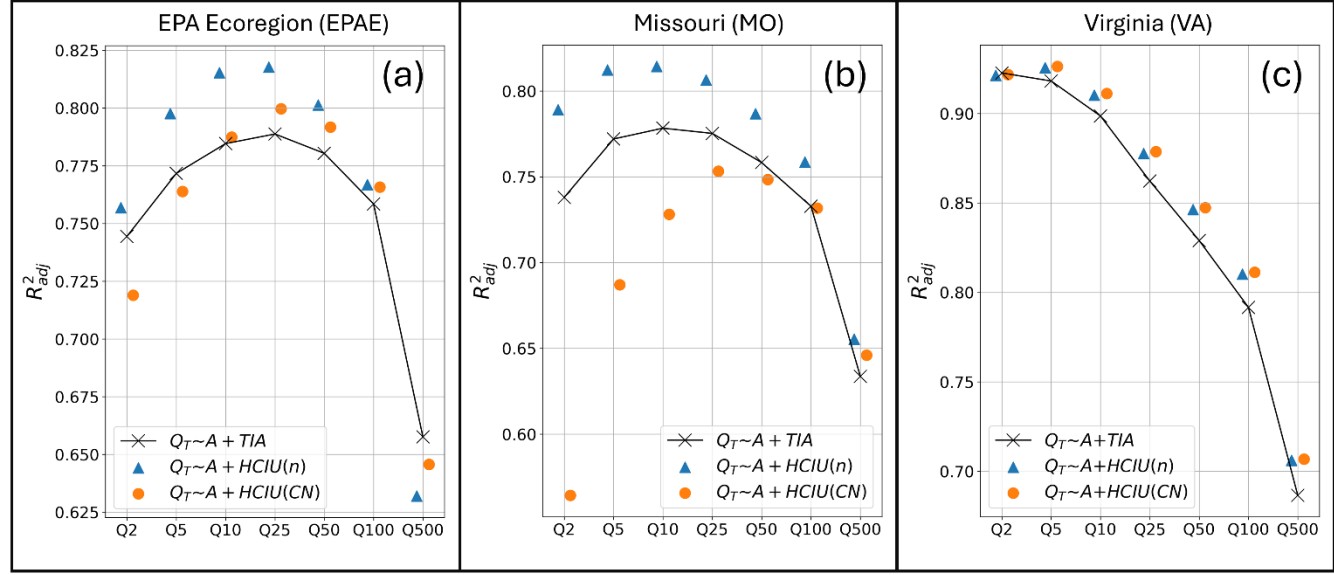

**Figure 7. Comparison of the performance of regional peak-flow equations calibrated on 1) $A$, $TIA$, and $Q_T$ data (i.e., $Q_T \sim A + TIA$ benchmark model), 2) $A$, $HCIU(n)$, and $Q_T$ data ($Q_T \sim A + HCIU(n)$ model), and 3) $A$, $HCIU(CN)$, and $Q_T$ data ($Q_T \sim A + HCIU(CN)$ model), for the a) EPAE, b) MO, and c) VA case studies. Quantiles $Q_T$ associated with return periods $T$ of 2, 5, 10, 25, 50, 100, and 500 years are considered (data in Appendix A).**

Our results indicate that not only is $HCIU$ a strong peak flow predictor in combination with $A$, it also systematically outperforms $TIA$ when the $n$-based formulation is considered. Improvement is strongest for the EPAE and MO cases studies, while for VA, where the performance of the benchmark model (with $A$ and $TIA$) was already the highest among case studies, only a marginal gain is obtained. On the other hand, the $CN$-based formulation for $HCIU$ seems to be overall less robust, with varying behaviors depending on the specific case study. For the EPAE homogeneous region, $HCIU(CN)$ outperforms the benchmark (but not its $n$-based counterpart) when fitting extreme flood values with return periods between 10 and 100 years, while it slightly underperforms for other flood quantiles. On the other hand, for the MO case study $HCIU(CN)$ displays a noticeably worse performance as compared to the benchmark, except for the two largest flood quantiles ($Q_{100}$ and $Q_{500}$). VA is the only region where the $CN$-based $HCIU$ performs similarly to the $n$-based $HCIU$, systematically outperforming the benchmark, even though only marginally, as all models perform well. In the Discussion section we hypothesize about what might explain the lower performance of $HCIU(CN)$, as compared to $HCIU(n)$.

Figure 7 also shows that, regardless of the urbanization metric considered, model performance decreases with increasing return period $T$ of the flood quantile. This may be due to a combination of two aspects. First, for such extreme events, differences across distinct LULC patches in the basin become increasingly negligible from a hydrologic perspective (Ogden et al., 2011), as wetter pervious patches infiltrate a smaller fraction of precipitation, causing any land-cover descriptor to lose predictive power. The second reason lies in the inevitable uncertainties associated with the estimation of such extreme quantiles from short flow records (Klemeš, 2000), which means that the models are fitted on highly uncertain data points. Because of these reasons, we suggest that improvements in prediction accuracy of urbanization metrics should be only pursued in the range of more frequent floods (say, below the 100-year return period). A generalized decrease in model performance is also observed when moving from intermediate to smaller quantiles, except for $TIA$ in VA. This trend may be due to the increasing influence of the minor drainage system on hydrologic response during smaller events, overshadowing surface runoff dynamics. However, both $TIA$ and $HCIU$ primarily focus on aspects related to surface runoff.

The main hydrologic application of regional peak-flow equations fitted on data from gauged basins is to extrapolate the relationships to other, ungauged basins. Therefore, a more informative way of testing the two competing urbanization metrics is to consider a k-fold validation framework, where the full dataset is split into a training and a test set, for fitting and evaluating the model, respectively. Following an approach similar to Dell'Aira et al. (2022), we produced a distribution of test errors (shown in Fig. 8) by repeatedly fitting the peak-flow equation (Eq. 8) and evaluating model performance on separate subsets

of the full dataset. For each case study, we alternately considered a 66-33%, 75-25%, and 80%-20% proportion for the training
and test-set sizes, respectively, associated with 3, 4, and 5 alternative training-test subset splits, respectively, in a way that all
610 the obtained test sets together span the full dataset without having duplicate basins. For each of these proportions, we repeated
the k-fold procedure 10 times, considering 10 different random samplings to populate the folds, to avoid any potential bias
associated with a single sampling. This procedure resulted in 120 blind assessments of the performance of the peak-flow
equations, each time fitted and tested on different subsets, for each case study.

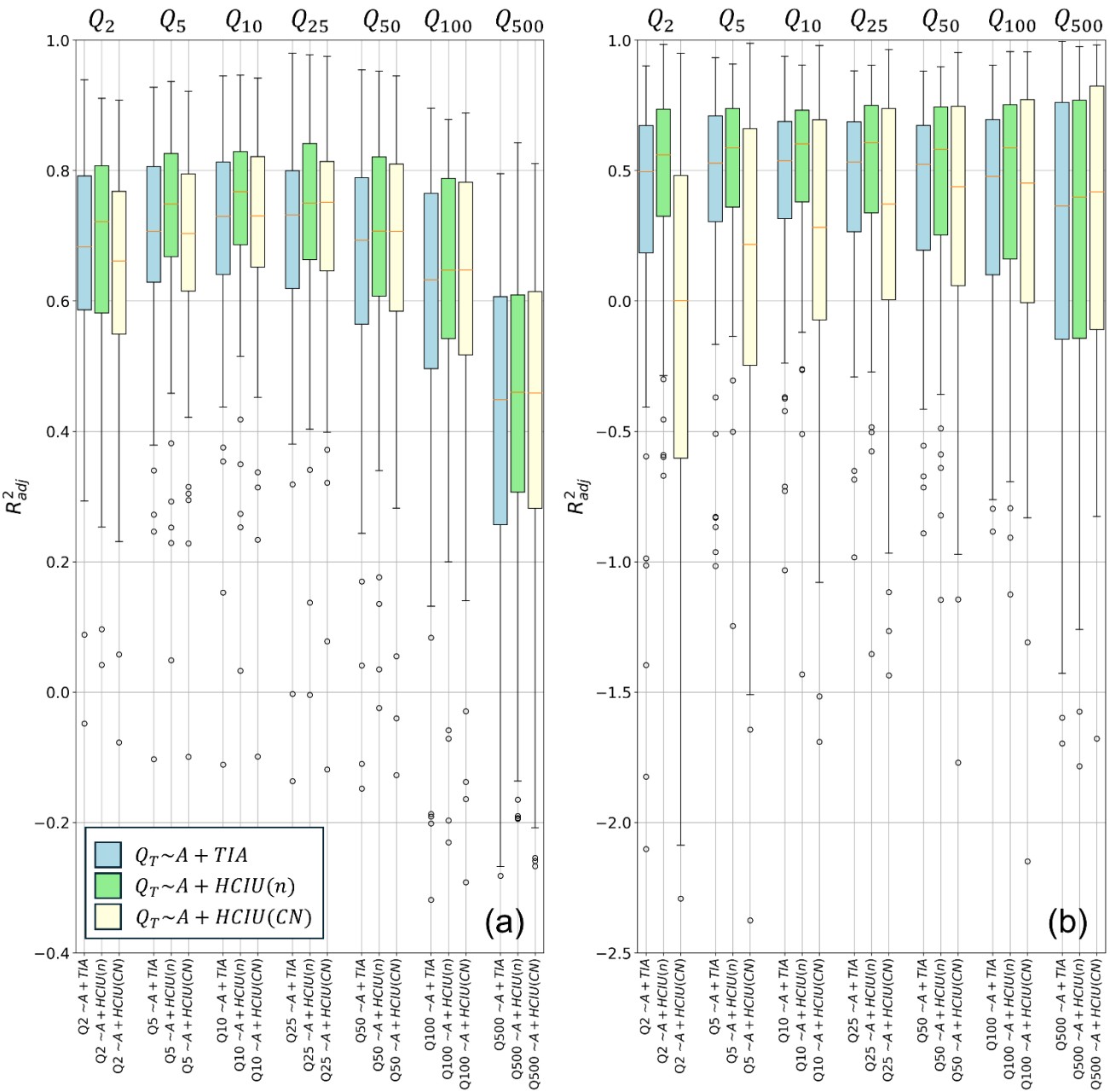

**Figure 8. Boxplots of test errors for 1) the benchmark $Q_T{\sim}A + TIA$, 2) $Q_T{\sim}A + HCIU(n)$, and 3) $Q_T{\sim}A + HCIU(CN)$ models, when**
**they are fitted and blind-tested on distinct basin subsets, for the a) EPAE and b) MO homogenous regions, respectively (see this**
**same plots for VA in Appendix D). For each boxplot, filled bars represent error values between the first and the third quartiles, the**
**upper (lower) whisker extends from the third (first) quartile by adding (subtracting) 1.5 times the interquartile range, and any**
**outliers beyond the whiskers are marked as circles.**

The boxplots of errors in Fig. 8 confirm that $HCIU(n)$ is the more robust urbanization metric for peak flow regression
equations. For all flood quantiles, $HCIU(n)$ improves model performance (as compared to the $Q_T{\sim}A + TIA$ benchmark) both
in average terms (as indicated by the systematically higher medians, as well as the higher location of the boxplots), as well as
with respect to the error spread, which is narrower for all EPAE (Fig. 8a) and MO (Fig. 8b) quantiles, with the only exception
of $Q_2$ for EPAE (Fig. 8a).

On the other hand, $HCIU(CN)$ exhibits again a more heterogeneous performance. In the EPAE case study (Fig. 8a), the error boxplots for $Q_T \sim A + HCIU(CN)$ are marginally better than those for the benchmark $Q_T \sim A + TIA$ across all quantiles except the two smallest (i.e., $Q_2$, and $Q_5$), with slightly lower distributions of adjusted $R^2$ (in agreement with what was observed in Fig. 7). A different picture is obtained for the MO case (Fig. 8b), where $HCIU(CN)$ significantly underperforms the $Q_T \sim A + TIA$ benchmark, for most quantiles, indicating that $HCIU(CN)$ is not a robust urbanization metric for the MO region.

The same boxplots for VA, moved to Appendix D since they do not add any relevant additional information, indicate comparable results as those already observed when testing the new urbanization metric (both with the $n$- and $CN$-based weighting approaches) on the full dataset (Fig. 7c). Specifically, both $HCIU(n)$ and $HCIU(CN)$ lead to improvements in model performance, as their related error boxplots are shifted upward, as compared to the benchmark.

## 5 Discussion

Our results indicate that the proposed conceptual framework for deriving hydrologic-connectivity-based urbanization metrics does produce lumped basin descriptors that successfully encapsulate information about the flood-enhancing impacts of urban sectors with different land-development intensities (and thus, with different imperviousness levels), considering their spatial arrangement, but is also sensitive to the spectrum of mitigation effects afforded by undeveloped (pervious) patches, depending on their relative location within the watershed. This is obtained by differentially weighing the effects of different LULC types through widely used and accepted conceptual criteria such as Manning's surface roughness coefficients and Curve Number values, hence capturing heterogeneity in the capacity of LULC patches to both generate and retain/detain runoff.

The resulting numerical value of $HCIU$ for a single watershed represents a measure of the proportion of hydrologic connectivity arising from the specific mosaic of land-cover patches in that watershed, relative to the maximum theoretical connectivity of that basin if its surface were completely paved. While this definition may not be as straightforward as that for $TIA$, which is simply a proportion of impervious areas, it results in a conceptually more comprehensive, and hydrologically driven representation of the distributed impacts of urbanization on surface runoff dynamics, in a lumped basin descriptor. Like $TIA$, $HCIU$ can be used to characterize and compare different basins, either to simply determine which basin is more impacted by land development, or else to develop regional models. $HCIU$ can also be utilized for planning, to compare the expected hydrologic effects of different scenarios for land-development in a given watershed.

We suggest that $HCIU$ should also increase our explanatory power when predicting other event-related variables such as lag times and times of concentration. $HCIU$ is indeed sensitive not only to the presence and spatial arrangement of LULC patches with different hydrologic characteristics but also to those locations where flows tend to concentrate, locally decreasing surface runoff travel times, as conceptually reflected in the upslope component $D_{up,k}$ (Eq. 4). $HCIU$ also considers the distance of these surface runoff "hotspots," where stormwater tends to concentrate and travel faster, to the stream network, as reflected by the downslope component $D_{dn,k}$. This in turn determines how easily those locations with accumulating flows will contribute to the overall basin response. Ultimately, $HCIU$ conceptually summarizes in a single number the effects of all potential runoff travel paths occurring on the basin surface, moving towards the stream, including interactions among converging surface flow paths, following a hydrologically driven approach. Because other response variables, such as lag time and time of concentration, are emergent basin properties arising from the interactions of all individual travel paths, their correlations with $HCIU$ or other connectivity-based descriptors should be investigated in future research. The $HCIU$ approach could also be further tested against $TIA$ for predicting other hydrologic variables that are affected by land-development, such as water quality indicators. Depending on the specific application, it may be necessary to make some adaptations to account for additional sources of connectivity, induced by, e.g., the underground stormwater sewer infrastructure.

In the next subsection, we discuss in more detail the possible reasons for the poor performance of the $CN$-based $HCIU$.
We then highlight advantages and disadvantages of the proposed methodology and point out challenges and possible future research directions to deploy the connectivity-based analysis framework to a range of different hydrologic applications.

**5.1 Considerations on the low performance of $HCIU(CN)$**

As noted in Section 4.1, the $CN$-based formulation of the normalized connectivity index and the resulting $HCIU(CN)$ (obtained as the weighted average of the former) are more complex than the $n$-based formulation, as they attempt to account for the combined effects of different coexisting land-use/land-cover and soil types. Because of this, the choice of the $CN$-based weighting coefficients as well as the interpretation of the resulting raster maps of normalized connectivity require particular attention.

For our analysis, we adapted $CN$ values associated with different land-use/land-cover types and hydrologic soil groups from the work by Wu et al. (2024), based on hydrologic modeling. According to their classification, urban cells with varying degrees of land development may be assigned $CN$ values over a wide range, depending on the associated soil characteristics. In some cases, moderately urbanized cells may have weights $W$ not too dissimilar from those of some cells with natural land-cover types, but low soil infiltrability, meaning that both types of cells are expected to generate similar amounts of runoff. However, it is clear that undeveloped patches would still mitigate hydrologic impacts better than urban areas, mostly through enhanced detention due to their higher roughness, delaying runoff contributions.

Interpretations for the overall low performance of $HCIU(CN)$ may hence include the variability in stormwater hydrology dynamics due to storm intensity and soil moisture conditions. Depending on these aspects, dominant control mechanisms on hydrologic connectivity and disconnection, such as soil infiltrability, rainfall interception by vegetation and litter, and surface roughness, as well as their interplay, may significantly change (Saffarpour et al., 2016; Zölch et al., 2017). For instance, the potential for generating runoff in moderately urbanized cells may be mainly governed by the type of soil, when regular events are considered – like in the case of the validation of the $CN$ classification proposed by Wu et al. (2024). On the other hand, when the scope of the analysis focuses on more extreme events (as in our case, with peak flow equations), the effects of land development may become preponderant (e.g., by decreasing response lag-times), since soils are more likely to reach saturated conditions. All of this in turn suggests the need to suitably adjust the $CN$ values of urbanized cells to better differentiate them from pervious pixels, even when the latter display low infiltrability.

Another likely explanation is that $CN$ also depends on antecedent soil moisture conditions (ASMCs), with different sets of values associated to distinct ASMC categories (i.e., dry, average, and wet; Wu et al., 2024). In our work, we considered the average scenario. However, because of the interactions between event intensity and soil moisture conditions, different sets of $CN$ values, associated with different ASMCs, should probably be considered instead, depending on the predicted event's return period. In the case of small, more frequent events (where we found the worst performances for EPAE and MO; Fig. 7a and 7b), $HCIU(CN)$ might benefit from using lower $CN$ values, to reflect drier antecedent conditions. On the other hand, for greater return-period events, soils are saturated faster, and higher $CN$ values, related to wet ASMCs, might improve the predictive power of $HCIU(CN)$.

In summary, based on the overall low performance observed with $HCIU(CN)$, the adopted weighting approach based on average-ASMCs Curve Numbers may not be the best for our proposed methodological framework, at least when the analysis aims at predicting extreme peak flows. A poor tuning of the $CN$-based weighting coefficients, which do not clearly distinguish undeveloped patches with low infiltrability from developed areas with highly permeable soils, may also explain why the $HCIU(CN)$ seems less robust, compared to the simpler $n$-based version. Further research is needed to study the sensitivity of $HCIU$ to different $CN$-based weighting approaches and fine tune the coefficients to maximize its predictive power.

## 5.2 Advantages and limitations of the proposed approach and future research directions

The proposed conceptual framework builds on and takes advantage of our qualitative understanding of some of the varied and complex dynamics affecting a basin's hydrologic response. The main strength of the proposed *HCIU* is that, in contrast with the traditional percentage of total impervious area or other metrics that index the impervious fraction of a basin, it explicitly takes into account the location and spatial arrangement of all types of LULC patches, in relation with basin topography, the stream network, and the outlet, in a continuum from the highly developed (which tend to generate more runoff, 710   faster) to the undisturbed, forested patches (with the strongest mitigating effects on hydrologic response). This represents an advantage over other advanced urbanization metrics in the literature. For instance, while the methodologies adopted by Yang et al. (2011) and Beck et al. (2016) both frame the hydrologic interconnections across some main LULC categories simply based on geometric aspects such as the density, spatial adjacency, and granularity of distinct LULC patches, *HCIU* captures the connectivity across a spectrum of surface patches as driven by basin topography. Or, in contrast with the two metrics 715   proposed by Zhang and Shuster (2014), that also consider the effects of basin relief but adopt a binary, pervious-or-impervious LULC representation, *HCIU* considers the continuous heterogeneity of hydrologic characteristics in the mosaic of LULC patches over a watershed.

      This is achieved by alternatively using Manning's surface roughness coefficients or Curve Numbers, which are both well-established conceptualizations that quantify different LULC types' abilities to either facilitate the generation and transmission 720   of runoff, or else retain, detain, or dampen water volumes, dependent on their surface roughness, extent (and type) of vegetated areas, as well as infiltrability. Highly impervious patches with smooth surfaces and limited losses represent one extreme in the continuous spectrum of LULC potentials for generating runoff, resulting in large peak flows; in the other are forested areas and other LULC types with natural soils, presence of litter, and high vegetation densities, because of their ability to diminish runoff volumes and travel speeds through interception (in the canopy and litter), temporary canopy storage, high surface 725   roughness, and variable infiltration capacity.

      Besides its comprehensive formulation, that conceptually considers the synergistic effects of both topography and LULC, another advantage of *HCIU* is that it can be virtually derived for any basin in the U.S. or the world, from widely available data, i.e., the DEM and LULC (or *CN*) maps. As a result, the proposed urbanization metric may easily find systematic application in peak flow prediction models and, depending upon preliminary testing, also in a range of other water resources management 730   fields that require quantifying the level of urbanization in a basin, such as water quality assessments and stormwater infrastructure design. We considered two alternative approaches to weigh the effects of different LULC types (and HSGs, in the case of *CN*-based weights) on hydrologic connectivity. However, modelers may also try other weighting criteria, depending on their specific needs. To foster the dissemination of our new metric, a link to a Python program to calculate the *HCIU* of any basin is provided in the Code Availability Statement ([https://github.com/dllaira/HCIU-urbanization-metric](https://github.com/dllaira/HCIU-urbanization-metric)).

A further advantage of the proposed approach is that it yields simple and easy-to-interpret, yet conceptually comprehensive assessments of the hydrologic impacts of urbanization across different basins. It can also be used to compare different land-development scenarios for the same watershed, aiding stakeholders in making urban planning decisions, or evaluating possible future LULC changes due to, e.g., the distributed implementation of candidate stormwater control measures.

      A current limitation of *HCIU* is that it only frames connectivity patterns driven by basin relief (topography), even though 740   the potential presence of stormwater drainage infrastructure would be another important source of connectivity in any urbanizing basin. Specifically, in its current version, the methodology does not capture the effects of underground stormwater sewer networks (also referred to as the minor system of stormwater infrastructure; Martins et al., 2017), although these are typically present in urban environments, especially in highly developed areas. However, stormwater drainage infrastructure usually includes not only underground pipe networks but also surface flow pathways and canals, which make up the so-called 745   major system. The major system is critical for handling larger, less-frequent storm events. When calculating *HCIU*, major drainage system sections connected to natural channels are treated as part of the stream network (assuming that excess flow

from the major system is poured directly into the stream network). This means that the connectivity of hillslope cells draining to the major system is calculated referring to the pour points along the major system. The contributions of these hillslope cells are then weighted based on the "along-the-stream-network" distance to the outlet, measured starting from the major-system pour point and following both the major-system and any subsequent natural-stream-network links downstream, when averaging the (normalized) connectivities to compute HCIU. This approach captures the effects of the stream network and major drainage systems, which have a stronger influence on the hydrologic response to extreme rainfall events compared to the minor system, whose capacity is typically overwhelmed by large runoff volumes. Consequently, the proposed HCIU should be a more reliable predictor of hydrologic-response variables under severe flooding conditions, as also suggested by the increase in model performance moving from small to intermediate peak quantiles, observed in Fig. 7.

On the other hand, when the analysis focuses on basin response to regular storms (e.g., in water-quality studies), the effects of the minor system should not be neglected, as the underground network may be able to handle most of the (smaller) runoff volumes. Another scenario where it is highly recommended to explicitly consider underground connectivity is when dealing with heavily urbanized watersheds, typically characterized by the presence of extensive drainage infrastructure. In these basins, detention tanks and sections of the minor system pumping stormwater against topographic gradients may completely change the connectivity determined by topographically driven surface runoff pathways. Our results are for basins with heterogeneous LULC characteristics, where urbanized sectors with varying development rates are mixed with natural LULC patches, typically displaying a distribution of land-development intensities more skewed towards lower values (Fig. 4j, 4k, and 4l), as is common for residential areas. Among the three studied regions, only VA included watersheds with TIA above 50%, but all of those were of small size. Because our dataset may not be representative of large, highly urbanized basins, for these cases (e.g., in countries where cities present generally higher land-development intensities, as compared to the U.S.) we recommend considering the effects of the minor, underground stormwater drainage infrastructure as well, when deriving HCIU. If stormwater sewer data are not available for the study region, and HCIU is estimated only considering topographically induced connectivity, some preliminary testing of its predictive power on gauged basins should be required (e.g., using the validation approach depicted in Fig. 8), before using the index for systematically generating peak-flows in ungauged, highly urbanized watersheds.

In principle, the proposed methodology allows for considering the effects of underground stormwater infrastructure as well; this would be achieved by first identifying pixels that are connected to each other and the stream network through stormwater sewer links, and then computing the connectivity index in a way such that connectivity patterns due to the stormwater sewer infrastructure override those from topography, whenever applicable. As we have found, however, information about underground stormwater drainage networks is not easily available, which may impede considering this additional source of connectivity, particularly in the case of multi-basin studies. Depending on data availability, an expanded version of *HCIU* that also accounts for the effects of stormwater sewer infrastructure as an additional source of connectivity should be another topic for future research.

**6 Conclusion**

We proposed and tested a new, hydrologic-connectivity-based index of urbanization *HCIU* that can be obtained in a GIS framework from the digital elevation model and land-use/land-cover (or Curve Number) map of a basin. We showed that, compared to the traditional fraction of total impervious area (*TIA*), *HCIU* helps capture more information about the impacts of land development on hydrologic response. *TIA* only indexes the proportion of impervious patches in a basin, while our new metric explicitly accounts for the spatial arrangement of the different land patches found in a watershed, both natural and developed. This is obtained by considering the spectrum of localized effects of distinct land-use/land-cover types on the hydrologic connectivity of surface runoff pathways.

The methodology builds on the well-established connectivity index, which has already found wide application in several hydrologic and geomorphic problems. Our specific interpretation of the connectivity index, in the framework of our approach, considers it as that distributed property explaining the ability of any hillslope location to quickly receive and transfer runoff to the stream network, depending on how topographically induced runoff pathways interact with urbanized sectors (with possible local differences in land-development intensity) and the undeveloped, typically more pervious patches in a watershed. We considered two alternative, widely used conceptual descriptors for quantifying the potential for runoff generation of different land-use/land-cover types, i.e., Manning's surface roughness coefficients and Curve Numbers. Depending on these metrics, the contributions on hydrologic connectivity of basin pixels with distinct surface characteristics are weighted differently. We found that weighting factors specified as a function of Manning's surface roughness coefficients result in more robust *HCIU* metrics, as compared to Curve-Number-based weighting methods, when explaining urban peak flows. However, we do not exclude that a fine tuning of the latter might improve model performance.

Irrespective of the weighting criterion, the procedure for obtaining a lumped metric for the effects of urbanization on hydrologic response, starting from a connectivity map of the basin of interest, involves the following steps: first, we define a normalized connectivity map, with respect to the maximum connectivity scenario associated with a virtual, fully developed copy of the original watershed; then, we calculate *HCIU* as a weighted average of the normalized connectivities of all basin pixels that are not part of the stream network, depending on the distance from their pour point to the watershed outlet, following the stream network.

We have shown that our new urbanization metric improves the predictive power of existing peak-flow regional equations, for three comprehensive case studies. Further research is required to test *HCIU*'s explanatory power for other hydrologic-response (i.e., flood-related) variables, such as lag-time and time of concentration, as well as other hydrologic variables of interest that have traditionally displayed correlation with *TIA*, such as water quality indicators. Depending on the scope of the analysis, an expanded version of the current formulation to account for the additional source of connectivity introduced by underground storm sewer infrastructure may be necessary; in highly urbanized watersheds, the latter may be a stronger control of basin response than topographically induced connectivity, especially in the case of less intense, more frequent events.

Besides its direct application as a metric of urbanization effects on basin response, *HCIU*'s sensitivity to the spatial arrangement of more developed and less developed (or undeveloped) sectors may provide a novel framework to facilitate comparisons of the hydrologic impacts caused by basin changes (e.g., due to urbanization or the introduction of stormwater runoff control measures), offering a valuable tool to stakeholders for informed urban-planning decisions. More research is needed to study the benefits and the range of applicability of the hydrologic connectivity-based index of urbanization.

## Appendix A

**Table A1. Case study basins. Area $A$, fraction of total impervious area $TIA$, and flood quantiles $Q_2$ to $Q_{500}$ were retrieved from Southard (2010), Austin (2014), and Feaster et al. (2014), for the MO, VA, and EPAE case study, respectively; we calculated $HCIU(n)$ and $HCIU(CN)$ following the methodology proposed in this work.**

| | Gauge ID | A (km²) | TIA (%) | Case Study | $Q_2$ (m³/s) | $Q_5$ (m³/s) | $Q_{10}$ (m³/s) | $Q_{25}$ (m³/s) | $Q_{50}$ (m³/s) | $Q_{100}$ (m³/s) | $Q_{500}$ (m³/s) | $HCIU(n)$ | $HCIU(CN)$ |
|---|---|---|---|---|---|---|---|---|---|---|---|---|---|
| 1 | 01613900 | 41.3 | 10.59 | VA | 24.3 | 46.5 | 64.6 | 90.7 | 112.4 | 135.9 | 197.7 | 0.177 | 0.601 |
| 2 | 01615000 | 150.6 | 18.78 | VA | 67.3 | 125.1 | 174.7 | 251.3 | 319.1 | 397.0 | 622.4 | 0.468 | 0.645 |
| 3 | 01616000 | 44.0 | 55.88 | VA | 15.3 | 26.4 | 35.3 | 48.3 | 59.2 | 71.1 | 103.5 | 0.669 | 0.689 |
| 4 | 01621450 | 1.7 | 15.41 | VA | 1.3 | 2.4 | 3.6 | 5.7 | 7.9 | 10.7 | 20.7 | 0.514 | 0.693 |
| 5 | 01623000 | 1.7 | 14.05 | VA | 0.3 | 2.0 | 6.9 | 27.3 | 70.3 | 170.3 | 1133.8 | 0.464 | 0.643 |
| 6 | 01623500 | 10.0 | 18.99 | VA | 1.1 | 4.4 | 9.8 | 25.1 | 48.0 | 88.6 | 334.1 | 0.487 | 0.689 |
| 7 | 01624800 | 189.1 | 15.15 | VA | 71.5 | 121.9 | 162.2 | 221.2 | 271.2 | 326.2 | 477.4 | 0.480 | 0.663 |
| 8 | 01625000 | 965.4 | 13.51 | VA | 165.0 | 318.8 | 451.4 | 655.3 | 834.8 | 1038.9 | 1622.0 | 0.439 | 0.648 |
| 9 | 01626000 | 328.7 | 11.22 | VA | 76.1 | 165.0 | 255.5 | 417.4 | 581.3 | 790.6 | 1514.1 | 0.294 | 0.620 |
| 10 | 01626500 | 345.7 | 13.66 | VA | 92.9 | 199.3 | 299.6 | 466.1 | 622.4 | 809.3 | 1387.0 | 0.319 | 0.625 |
| 11 | 01626850 | 382.7 | 15.6 | VA | 113.5 | 239.5 | 361.6 | 571.2 | 774.7 | 1025.4 | 1843.1 | 0.319 | 0.627 |
| 12 | 01627500 | 548.2 | 14.84 | VA | 152.9 | 318.0 | 464.7 | 694.6 | 899.6 | 1133.8 | 1807.2 | 0.314 | 0.628 |
| 13 | 01628500 | 2795.4 | 11.65 | VA | 466.9 | 902.2 | 1303.1 | 1964.3 | 2586.7 | 3335.7 | 5697.3 | 0.370 | 0.633 |
| 14 | 01629500 | 3554.2 | 10.97 | VA | 552.2 | 1078.0 | 1575.0 | 2414.0 | 3222.5 | 4213.5 | 7438.8 | 0.342 | 0.628 |
| 15 | 01631000 | 4232.8 | 10.55 | VA | 590.4 | 1153.3 | 1675.2 | 2539.5 | 3355.5 | 4341.0 | 7455.8 | 0.320 | 0.626 |
| 16 | 01636210 | 36.3 | 10.71 | VA | 21.0 | 34.9 | 46.6 | 64.4 | 80.1 | 98.1 | 150.4 | 0.178 | 0.607 |
| 17 | 0163626650 | 29.1 | 15.52 | VA | 15.9 | 20.1 | 22.8 | 26.2 | 28.7 | 31.1 | 37.0 | 0.188 | 0.603 |
| 18 | 01638350 | 81.9 | 10.48 | VA | 48.9 | 118.7 | 194.3 | 336.1 | 485.4 | 680.7 | 1148.8 | 0.465 | 0.617 |
| 19 | 01643805 | 98.7 | 11.39 | VA | 128.5 | 296.5 | 469.2 | 778.4 | 1089.9 | 1484.7 | 2827.7 | 0.460 | 0.608 |
| 20 | 01644280 | 197.2 | 38.7 | VA | 148.0 | 186.2 | 212.2 | 245.9 | 271.8 | 298.2 | 363.3 | 0.571 | 0.644 |
| 21 | 01644290 | 0.2 | 91.4 | VA | 0.2 | 0.8 | 1.9 | 5.0 | 9.7 | 18.2 | 71.0 | 0.933 | 0.827 |
| 22 | 01644291 | 0.2 | 91.43 | VA | 2.9 | 4.7 | 6.2 | 8.6 | 10.8 | 13.4 | 21.3 | 0.948 | 0.869 |
| 23 | 01644295 | 0.9 | 75.03 | VA | 2.3 | 4.8 | 7.2 | 11.3 | 15.2 | 20.1 | 35.9 | 0.664 | 0.712 |
| 24 | 01644300 | 8.8 | 81.35 | VA | 10.6 | 14.7 | 17.8 | 22.0 | 25.5 | 29.2 | 39.0 | 0.768 | 0.724 |
| 25 | 01645700 | 11.3 | 47.91 | VA | 13.8 | 21.1 | 26.4 | 33.7 | 39.6 | 45.8 | 61.7 | 0.464 | 0.647 |
| 26 | 01645750 | 4.1 | 24.68 | VA | 2.9 | 5.0 | 6.7 | 9.6 | 12.2 | 15.3 | 25.0 | 0.239 | 0.595 |
| 27 | 01645784 | 2.0 | 62.38 | VA | 11.9 | 17.6 | 21.7 | 27.3 | 31.7 | 36.2 | 47.7 | 0.574 | 0.635 |
| 28 | 01645900 | 13.1 | 50.81 | VA | 10.3 | 16.8 | 22.2 | 30.4 | 37.6 | 46.0 | 70.4 | 0.559 | 0.642 |
| 29 | 01645975 | 8.3 | 29.52 | VA | 34.5 | 37.4 | 39.2 | 41.5 | 43.2 | 44.8 | 48.6 | 0.267 | 0.615 |
| 30 | 01646000 | 149.8 | 35.77 | VA | 50.1 | 95.7 | 143.7 | 234.0 | 330.7 | 460.7 | 960.2 | 0.399 | 0.615 |
| 31 | 01646200 | 12.1 | 61.73 | VA | 31.3 | 62.0 | 91.4 | 141.6 | 190.4 | 250.9 | 450.5 | 0.557 | 0.660 |
| 32 | 01646600 | 7.5 | 74.53 | VA | 17.6 | 26.4 | 32.9 | 42.0 | 49.4 | 57.3 | 78.1 | 0.728 | 0.653 |
| 33 | 01646700 | 21.4 | 67.85 | VA | 35.6 | 59.9 | 80.7 | 112.8 | 141.5 | 174.8 | 273.7 | 0.627 | 0.634 |
| 34 | 01646750 | 1.1 | 90.74 | VA | 8.1 | 14.1 | 19.3 | 27.6 | 35.1 | 43.9 | 70.5 | 0.729 | 0.661 |
| 35 | 01646800 | 6.0 | 64.33 | VA | 27.1 | 44.9 | 60.7 | 85.9 | 109.3 | 137.2 | 224.3 | 0.630 | 0.626 |
| 36 | 01652400 | 2.4 | 91 | VA | 19.3 | 25.7 | 30.4 | 36.8 | 41.9 | 47.4 | 61.7 | 0.781 | 0.688 |
| 37 | 01652430 | 2.4 | 91.8 | VA | 18.8 | 29.5 | 38.6 | 52.6 | 65.2 | 79.9 | 124.0 | 0.831 | 0.694 |
| 38 | 01652470 | 3.4 | 93.64 | VA | 21.1 | 39.4 | 56.3 | 84.5 | 111.5 | 144.3 | 250.1 | 0.849 | 0.736 |
| 39 | 01652500 | 32.6 | 86.5 | VA | 84.3 | 156.3 | 217.6 | 311.5 | 394.2 | 488.2 | 758.0 | 0.776 | 0.699 |
| 40 | 01652600 | 7.1 | 71.83 | VA | 15.5 | 32.5 | 49.2 | 78.0 | 106.3 | 141.4 | 257.9 | 0.759 | 0.677 |
| 41 | 01652610 | 18.4 | 63.05 | VA | 19.9 | 34.4 | 47.9 | 70.3 | 91.8 | 118.2 | 204.8 | 0.563 | 0.638 |

| | | | | | | | | | | | | | |
|---|---|---|---|---|---|---|---|---|---|---|---|---|---|
| 42 | 01652620 | 4.9 | 67.64 | VA | 19.0 | 31.4 | 41.5 | 56.2 | 68.7 | 82.6 | 121.3 | 0.670 | 0.645 |
| 43 | 01652645 | 1.3 | 70.99 | VA | 8.4 | 8.9 | 9.2 | 9.5 | 9.7 | 10.0 | 10.4 | 0.658 | 0.669 |
| 44 | 01652650 | 12.0 | 71.48 | VA | 29.4 | 56.8 | 82.9 | 127.4 | 170.6 | 224.0 | 400.4 | 0.680 | 0.663 |
| 45 | 01652710 | 5.4 | 63.68 | VA | 15.4 | 24.7 | 31.8 | 42.1 | 50.6 | 60.0 | 85.2 | 0.522 | 0.623 |
| 46 | 01652810 | 5.9 | 64.51 | VA | 10.3 | 15.8 | 20.4 | 27.4 | 33.5 | 40.6 | 61.5 | 0.631 | 0.643 |
| 47 | 01652910 | 34.9 | 73.91 | VA | 59.7 | 104.9 | 142.2 | 198.2 | 246.7 | 301.3 | 455.1 | 0.669 | 0.688 |
| 48 | 01653000 | 87.7 | 70.21 | VA | 118.1 | 197.5 | 260.8 | 353.1 | 431.3 | 517.3 | 753.2 | 0.660 | 0.684 |
| 49 | 01653210 | 6.7 | 78.74 | VA | 16.2 | 23.8 | 29.6 | 37.9 | 44.9 | 52.5 | 73.3 | 0.736 | 0.647 |
| 50 | 01653447 | 2.0 | 81.53 | VA | 6.1 | 8.5 | 10.3 | 12.8 | 14.8 | 17.0 | 22.6 | 0.773 | 0.701 |
| 51 | 01653900 | 17.8 | 73.88 | VA | 36.0 | 74.3 | 114.4 | 189.2 | 268.1 | 372.9 | 762.0 | 0.714 | 0.676 |
| 52 | 01653950 | 3.1 | 71.38 | VA | 27.4 | 43.8 | 56.8 | 75.8 | 91.9 | 109.6 | 158.9 | 0.698 | 0.660 |
| 53 | 01654000 | 61.8 | 62.22 | VA | 66.8 | 122.3 | 170.0 | 244.0 | 310.1 | 386.0 | 608.0 | 0.591 | 0.658 |
| 54 | 01654500 | 9.6 | 59.32 | VA | 12.1 | 28.0 | 46.3 | 82.9 | 124.3 | 182.1 | 417.1 | 0.513 | 0.622 |
| 55 | 01655000 | 96.0 | 61.81 | VA | 40.1 | 59.8 | 75.8 | 99.9 | 120.9 | 144.8 | 213.9 | 0.570 | 0.652 |
| 56 | 01655310 | 9.9 | 64.03 | VA | 12.5 | 26.4 | 41.8 | 72.0 | 105.4 | 151.4 | 335.3 | 0.623 | 0.646 |
| 57 | 01655350 | 39.3 | 64.87 | VA | 24.7 | 40.6 | 55.4 | 80.2 | 104.2 | 133.9 | 232.7 | 0.561 | 0.643 |
| 58 | 01655370 | 9.4 | 68.32 | VA | 14.1 | 26.8 | 38.3 | 57.1 | 74.6 | 95.5 | 160.7 | 0.600 | 0.638 |
| 59 | 01655380 | 16.3 | 30.47 | VA | 12.9 | 24.6 | 35.8 | 54.8 | 73.3 | 96.2 | 172.4 | 0.378 | 0.628 |
| 60 | 01655390 | 81.0 | 54.24 | VA | 44.1 | 77.5 | 109.1 | 163.0 | 215.7 | 281.5 | 503.2 | 0.480 | 0.637 |
| 61 | 01656800 | 20.1 | 51.67 | VA | 9.8 | 11.2 | 12.0 | 13.0 | 13.7 | 14.4 | 16.1 | 0.540 | 0.612 |
| 62 | 01656960 | 128.8 | 52.27 | VA | 88.3 | 169.9 | 240.1 | 348.3 | 443.7 | 552.5 | 863.1 | 0.576 | 0.647 |
| 63 | 01657000 | 378.1 | 29.1 | VA | 193.3 | 354.0 | 510.3 | 784.1 | 1058.2 | 1406.5 | 2617.9 | 0.481 | 0.625 |
| 64 | 01657415 | 478.4 | 29.55 | VA | 304.1 | 564.4 | 831.4 | 1322.7 | 1838.0 | 2520.5 | 5065.9 | 0.455 | 0.633 |
| 65 | 01657500 | 1476.7 | 15.85 | VA | 312.1 | 458.4 | 566.3 | 715.3 | 835.3 | 963.6 | 1297.5 | 0.387 | 0.618 |
| 66 | 01657655 | 10.2 | 45.08 | VA | 15.8 | 27.3 | 37.3 | 53.0 | 67.4 | 84.2 | 135.4 | 0.430 | 0.616 |
| 67 | 01657800 | 11.7 | 41.39 | VA | 14.7 | 21.5 | 27.0 | 35.2 | 42.2 | 50.2 | 72.8 | 0.578 | 0.627 |
| 68 | 01667600 | 1.7 | 12.35 | VA | 2.2 | 3.0 | 3.5 | 4.3 | 4.8 | 5.4 | 6.9 | 0.423 | 0.594 |
| 69 | 01673500 | 14.9 | 31.5 | VA | 3.1 | 5.6 | 7.9 | 11.9 | 15.8 | 20.6 | 36.6 | 0.414 | 0.603 |
| 70 | 01673550 | 66.1 | 14.95 | VA | 10.7 | 24.3 | 41.1 | 77.5 | 121.8 | 188.5 | 497.2 | 0.319 | 0.603 |
| 71 | 02019400 | 75.9 | 13.09 | VA | 60.7 | 110.3 | 154.2 | 224.4 | 288.8 | 364.7 | 596.4 | 0.432 | 0.677 |
| 72 | 02027700 | 1.2 | 13.56 | VA | 1.0 | 2.0 | 3.1 | 5.0 | 6.9 | 9.3 | 17.9 | 0.176 | 0.571 |
| 73 | 02030800 | 7.0 | 17.31 | VA | 6.9 | 11.3 | 14.8 | 20.0 | 24.4 | 29.3 | 42.7 | 0.143 | 0.608 |
| 74 | 02031000 | 246.7 | 11.54 | VA | 130.3 | 252.2 | 352.5 | 500.1 | 624.1 | 759.7 | 1121.9 | 0.263 | 0.604 |
| 75 | 02033500 | 1303.5 | 12.41 | VA | 253.3 | 346.0 | 417.7 | 520.5 | 606.8 | 702.0 | 963.6 | 0.287 | 0.604 |
| 76 | 02034000 | 1716.6 | 11.83 | VA | 445.4 | 821.5 | 1147.1 | 1656.3 | 2112.7 | 2640.3 | 4196.6 | 0.287 | 0.600 |
| 77 | 02034050 | 4.1 | 13.74 | VA | 5.6 | 18.0 | 36.2 | 81.6 | 143.3 | 244.0 | 773.6 | 0.331 | 0.586 |
| 78 | 02037800 | 47.0 | 66.61 | VA | 13.5 | 29.4 | 47.1 | 82.1 | 121.1 | 175.4 | 393.9 | 0.567 | 0.629 |
| 79 | 02038000 | 85.7 | 58.19 | VA | 21.9 | 52.4 | 87.6 | 158.3 | 238.0 | 349.4 | 799.1 | 0.496 | 0.621 |
| 80 | 02038500 | 138.2 | 59.25 | VA | 30.8 | 63.6 | 98.3 | 163.7 | 233.5 | 327.1 | 680.5 | 0.530 | 0.629 |
| 81 | 02042000 | 363.0 | 16.99 | VA | 56.3 | 131.8 | 223.1 | 417.1 | 647.9 | 987.4 | 2494.4 | 0.317 | 0.589 |
| 82 | 02042287 | 161.0 | 23.7 | VA | 45.4 | 86.5 | 125.7 | 192.6 | 258.0 | 339.2 | 610.2 | 0.413 | 0.616 |
| 83 | 02042426 | 97.0 | 66.86 | VA | 55.7 | 64.4 | 70.2 | 77.6 | 83.1 | 88.7 | 102.1 | 0.659 | 0.669 |
| 84 | 02042500 | 651.3 | 24.49 | VA | 45.7 | 86.0 | 124.9 | 192.2 | 258.8 | 342.6 | 627.8 | 0.395 | 0.614 |
| 85 | 02042780 | 6.4 | 14.22 | VA | 2.6 | 3.6 | 4.5 | 5.8 | 7.0 | 8.3 | 12.2 | 0.306 | 0.557 |
| 86 | 02044400 | 4.2 | 22.16 | VA | 5.5 | 16.1 | 30.6 | 64.7 | 108.8 | 178.1 | 517.9 | 0.392 | 0.573 |
| 87 | 02055000 | 994.3 | 18.65 | VA | 216.1 | 363.3 | 472.6 | 621.3 | 738.8 | 860.8 | 1165.8 | 0.286 | 0.614 |
| 88 | 02055100 | 30.3 | 14.73 | VA | 22.0 | 49.8 | 77.6 | 126.1 | 173.8 | 233.2 | 428.4 | 0.431 | 0.679 |
| 89 | 02056000 | 1319.5 | 23.56 | VA | 311.2 | 496.7 | 640.2 | 845.3 | 1015.2 | 1200.6 | 1698.7 | 0.345 | 0.627 |

| | | | | | | | | | | | | | |
|---|---|---|---|---|---|---|---|---|---|---|---|---|---|
| 90 | 02056650 | 144.5 | 18.67 | VA | 75.7 | 160.6 | 240.7 | 373.5 | 498.1 | 647.3 | 1110.6 | 0.226 | 0.596 |
| 91 | 02057500 | 2634.0 | 15.21 | VA | 451.4 | 735.7 | 964.8 | 1303.7 | 1594.5 | 1919.9 | 2834.5 | 0.341 | 0.622 |
| 92 | 02057700 | 2.0 | 71.86 | VA | 4.0 | 5.4 | 6.4 | 7.8 | 8.8 | 9.9 | 12.7 | 0.706 | 0.692 |
| 93 | 02059000 | 3673.5 | 12.18 | VA | 587.6 | 908.1 | 1152.8 | 1499.9 | 1786.2 | 2096.9 | 2928.0 | 0.327 | 0.612 |
| 94 | 02059450 | 28.4 | 10.81 | VA | 12.7 | 24.6 | 35.9 | 55.5 | 74.8 | 98.9 | 179.7 | 0.208 | 0.607 |
| 95 | 02060500 | 4615.3 | 10.91 | VA | 844.7 | 1294.1 | 1640.1 | 2134.0 | 2544.6 | 2993.1 | 4205.1 | 0.323 | 0.607 |
| 96 | 02061150 | 4.0 | 14.37 | VA | 4.0 | 10.8 | 19.1 | 36.3 | 56.0 | 84.1 | 199.2 | 0.374 | 0.623 |
| 97 | 02062500 | 6225.6 | 10.22 | VA | 887.2 | 1383.0 | 1792.2 | 2413.4 | 2959.1 | 3584.9 | 5414.2 | 0.245 | 0.605 |
| 98 | 02076400 | 5.2 | 19.11 | VA | 5.1 | 8.2 | 10.7 | 14.6 | 18.1 | 22.1 | 34.1 | 0.415 | 0.612 |
| 99 | 02076500 | 23.8 | 15.84 | VA | 14.1 | 26.4 | 36.8 | 52.5 | 66.2 | 81.5 | 124.6 | 0.411 | 0.592 |
| 100 | 02086849 | 56.7 | 20.3 | EPAE | 56.1 | 67.7 | 74.2 | 81.3 | 86.1 | 90.3 | 99.4 | 0.638 | 0.637 |
| 101 | 0208726005 | 196.8 | 16.7 | EPAE | 64.3 | 111.3 | 146.7 | 195.1 | 233.3 | 273.0 | 371.0 | 0.575 | 0.626 |
| 102 | 02087324 | 313.4 | 16.5 | EPAE | 97.4 | 154.3 | 206.4 | 294.5 | 376.6 | 481.4 | 818.4 | 0.629 | 0.631 |
| 103 | 0208732885 | 17.7 | 29.5 | EPAE | 26.0 | 47.9 | 67.1 | 98.0 | 126.3 | 159.7 | 261.9 | 0.792 | 0.692 |
| 104 | 02087359 | 77.2 | 21.3 | EPAE | 40.8 | 75.0 | 109.6 | 172.7 | 237.6 | 322.8 | 634.3 | 0.716 | 0.660 |
| 105 | 02087580 | 54.4 | 15.3 | EPAE | 52.1 | 78.2 | 101.7 | 139.3 | 174.7 | 217.8 | 354.0 | 0.674 | 0.642 |
| 106 | 0209399200 | 41.2 | 22.5 | EPAE | 31.4 | 57.5 | 77.6 | 106.2 | 129.4 | 154.3 | 217.2 | 0.701 | 0.656 |
| 107 | 02094659 | 19.0 | 41.2 | EPAE | 52.7 | 76.5 | 92.3 | 113.0 | 128.6 | 144.1 | 181.5 | 0.853 | 0.721 |
| 108 | 02094770 | 39.9 | 39.5 | EPAE | 47.3 | 63.4 | 73.9 | 87.5 | 97.4 | 107.3 | 131.1 | 0.839 | 0.718 |
| 109 | 02095000 | 88.1 | 35.5 | EPAE | 79.6 | 90.6 | 97.4 | 105.3 | 110.7 | 116.1 | 127.7 | 0.778 | 0.704 |
| 110 | 02095271 | 36.8 | 32.7 | EPAE | 58.0 | 78.4 | 90.9 | 105.3 | 115.2 | 124.6 | 144.7 | 0.844 | 0.695 |
| 111 | 02095500 | 96.1 | 28.8 | EPAE | 80.7 | 127.1 | 161.4 | 208.1 | 244.9 | 283.2 | 382.3 | 0.735 | 0.680 |
| 112 | 0209553650 | 229.2 | 27 | EPAE | 143.3 | 166.2 | 178.7 | 192.3 | 201.0 | 209.5 | 225.4 | 0.669 | 0.674 |
| 113 | 0209741955 | 54.6 | 14.4 | EPAE | 143.3 | 166.2 | 178.7 | 192.3 | 201.0 | 209.5 | 225.4 | 0.571 | 0.621 |
| 114 | 02115845 | 13.4 | 20 | EPAE | 45.9 | 56.9 | 64.0 | 72.2 | 78.4 | 84.4 | 98.3 | 0.850 | 0.699 |
| 115 | 0212414900 | 89.6 | 19.1 | EPAE | 85.0 | 141.6 | 194.0 | 281.8 | 368.1 | 470.1 | 818.4 | 0.660 | 0.651 |
| 116 | 0214266000 | 68.1 | 14.8 | EPAE | 27.7 | 50.1 | 71.6 | 108.7 | 145.0 | 190.6 | 345.5 | 0.576 | 0.633 |
| 117 | 02142900 | 42.5 | 21 | EPAE | 43.6 | 65.1 | 81.3 | 103.9 | 122.0 | 141.9 | 193.7 | 0.641 | 0.656 |
| 118 | 0214291555 | 81.6 | 17.4 | EPAE | 56.4 | 87.8 | 109.6 | 138.2 | 159.7 | 181.2 | 232.8 | 0.616 | 0.641 |
| 119 | 0214295600 | 26.9 | 20.4 | EPAE | 33.1 | 53.2 | 67.7 | 86.9 | 101.7 | 117.2 | 154.9 | 0.692 | 0.649 |
| 120 | 02145940 | 9.1 | 25.5 | EPAE | 24.7 | 28.6 | 30.9 | 33.1 | 34.8 | 36.2 | 39.1 | 0.812 | 0.655 |
| 121 | 02146211 | 15.5 | 26 | EPAE | 25.3 | 39.1 | 52.1 | 74.5 | 97.1 | 125.4 | 224.0 | 0.706 | 0.683 |
| 122 | 0214627970 | 23.5 | 32 | EPAE | 70.5 | 105.3 | 128.0 | 156.0 | 176.1 | 196.2 | 240.4 | 0.801 | 0.710 |
| 123 | 02146300 | 79.5 | 34.2 | EPAE | 116.4 | 167.1 | 206.1 | 261.4 | 305.8 | 356.8 | 492.7 | 0.793 | 0.712 |
| 124 | 02146315 | 14.8 | 36.8 | EPAE | 48.4 | 71.6 | 87.2 | 106.5 | 120.6 | 134.2 | 165.9 | 0.820 | 0.710 |
| 125 | 02146348 | 23.7 | 24.1 | EPAE | 22.4 | 33.4 | 40.5 | 49.0 | 54.7 | 60.3 | 71.9 | 0.541 | 0.664 |
| 126 | 02146381 | 169.1 | 32.4 | EPAE | 99.4 | 146.1 | 184.9 | 243.8 | 297.3 | 356.8 | 532.4 | 0.733 | 0.696 |
| 127 | 02146409 | 30.6 | 47.9 | EPAE | 94.0 | 120.6 | 133.9 | 147.5 | 155.7 | 162.5 | 174.7 | 0.869 | 0.764 |
| 128 | 0214642825 | 13.5 | 24.6 | EPAE | 41.9 | 59.7 | 73.9 | 94.6 | 112.1 | 132.0 | 188.0 | 0.800 | 0.662 |
| 129 | 0214645022 | 49.2 | 25 | EPAE | 75.3 | 111.6 | 138.8 | 176.1 | 206.4 | 239.3 | 325.6 | 0.806 | 0.670 |
| 130 | 02146470 | 6.8 | 32.8 | EPAE | 32.0 | 48.1 | 59.2 | 73.3 | 84.1 | 95.1 | 120.9 | 0.851 | 0.681 |
| 131 | 02146500 | 106.2 | 22 | EPAE | 110.2 | 151.5 | 178.7 | 213.2 | 238.7 | 264.2 | 325.6 | 0.828 | 0.698 |
| 132 | 02146507 | 110.3 | 32 | EPAE | 192.8 | 266.7 | 317.1 | 385.1 | 438.9 | 492.7 | 628.6 | 0.826 | 0.697 |
| 133 | 02146530 | 127.4 | 32 | EPAE | 137.3 | 194.0 | 236.7 | 294.5 | 342.6 | 393.6 | 529.5 | 0.823 | 0.696 |
| 134 | 0214655255 | 19.0 | 18.2 | EPAE | 33.7 | 77.0 | 114.7 | 171.0 | 217.8 | 268.4 | 399.3 | 0.725 | 0.650 |
| 135 | 02146562 | 14.8 | 26.4 | EPAE | 25.7 | 54.4 | 82.1 | 129.1 | 174.7 | 229.6 | 407.8 | 0.796 | 0.674 |
| 136 | 02146600 | 100.0 | 20.2 | EPAE | 105.3 | 145.3 | 170.8 | 202.2 | 224.8 | 247.8 | 300.2 | 0.700 | 0.652 |
| 137 | 02146700 | 18.0 | 21.3 | EPAE | 48.1 | 72.2 | 88.3 | 109.0 | 124.6 | 140.2 | 176.7 | 0.798 | 0.642 |

| | | | | | | | | | | | | | |
|---|---|---|---|---|---|---|---|---|---|---|---|---|---|
| 138 | 02146750 | 239.3 | 19.5 | EPAE | 145.0 | 201.3 | 235.0 | 274.1 | 300.2 | 325.6 | 376.6 | 0.725 | 0.648 |
| 139 | 0214678175 | 17.9 | 31.4 | EPAE | 27.0 | 40.2 | 50.1 | 64.0 | 75.3 | 87.8 | 120.1 | 0.704 | 0.689 |
| 140 | 02159785 | 1.0 | 19.3 | EPAE | 4.1 | 5.9 | 6.9 | 8.1 | 8.9 | 9.7 | 11.4 | 0.721 | 0.644 |
| 141 | 02160325 | 23.4 | 22.3 | EPAE | 21.3 | 34.3 | 49.8 | 82.4 | 121.2 | 178.7 | 441.7 | 0.704 | 0.658 |
| 142 | 02164000 | 125.9 | 18.8 | EPAE | 68.0 | 98.5 | 120.6 | 150.4 | 173.9 | 198.5 | 261.9 | 0.645 | 0.655 |
| 143 | 02164011 | 7.8 | 34.9 | EPAE | 26.7 | 33.1 | 36.5 | 40.2 | 42.8 | 44.7 | 48.7 | 0.832 | 0.696 |
| 144 | 02168845 | 1.0 | 26.6 | EPAE | 4.4 | 5.4 | 6.1 | 7.1 | 7.8 | 8.6 | 10.4 | 0.778 | 0.657 |
| 145 | 02203800 | 107.5 | 30.9 | EPAE | 107.6 | 159.7 | 189.2 | 221.7 | 242.7 | 261.4 | 297.3 | 0.680 | 0.679 |
| 146 | 02203835 | 8.9 | 16.8 | EPAE | 20.2 | 28.3 | 34.5 | 43.3 | 50.4 | 58.0 | 78.4 | 0.648 | 0.610 |
| 147 | 02203845 | 2.5 | 24.7 | EPAE | 12.1 | 16.7 | 19.0 | 21.2 | 22.5 | 23.5 | 25.2 | 0.721 | 0.638 |
| 148 | 02203884 | 4.9 | 27.7 | EPAE | 19.0 | 26.3 | 31.1 | 36.8 | 41.3 | 45.6 | 55.8 | 0.678 | 0.645 |
| 149 | 02203900 | 256.4 | 23.6 | EPAE | 148.9 | 215.2 | 256.6 | 305.8 | 339.8 | 371.0 | 438.9 | 0.645 | 0.650 |
| 150 | 02204070 | 471.4 | 20.2 | EPAE | 186.3 | 266.7 | 325.6 | 402.1 | 464.4 | 526.7 | 690.9 | 0.596 | 0.635 |
| 151 | 02205000 | 3.3 | 22.4 | EPAE | 8.5 | 18.1 | 26.2 | 37.9 | 47.6 | 58.0 | 84.4 | 0.720 | 0.655 |
| 152 | 02205230 | 0.9 | 15.4 | EPAE | 4.0 | 5.6 | 6.5 | 7.7 | 8.5 | 9.3 | 11.0 | 0.590 | 0.616 |
| 153 | 02205500 | 6.3 | 27.1 | EPAE | 15.3 | 31.7 | 48.7 | 80.7 | 114.7 | 159.7 | 328.5 | 0.685 | 0.652 |
| 154 | 02205596 | 18.7 | 22 | EPAE | 20.8 | 31.1 | 37.9 | 46.4 | 52.7 | 58.6 | 72.5 | 0.681 | 0.652 |
| 155 | 02206105 | 0.4 | 24.9 | EPAE | 2.4 | 3.6 | 4.4 | 5.6 | 6.5 | 7.4 | 9.6 | 0.704 | 0.613 |
| 156 | 02206136 | 0.9 | 23.8 | EPAE | 3.7 | 4.6 | 5.3 | 6.0 | 6.5 | 7.1 | 8.3 | 0.620 | 0.614 |
| 157 | 02206500 | 347.1 | 22.1 | EPAE | 102.8 | 161.7 | 207.6 | 271.6 | 325.6 | 382.3 | 535.2 | 0.656 | 0.659 |
| 158 | 02207000 | 14.3 | 14.3 | EPAE | 24.1 | 37.7 | 46.7 | 58.0 | 66.0 | 73.9 | 91.5 | 0.581 | 0.614 |
| 159 | 02207500 | 979.0 | 12.5 | EPAE | 156.0 | 270.4 | 393.6 | 625.8 | 877.8 | 1223.3 | 2593.8 | 0.549 | 0.625 |
| 160 | 02208050 | 25.8 | 22.4 | EPAE | 19.7 | 32.6 | 43.3 | 59.7 | 74.2 | 91.2 | 140.2 | 0.657 | 0.656 |
| 161 | 02217505 | 3.7 | 29.1 | EPAE | 14.3 | 19.0 | 22.5 | 27.2 | 31.1 | 35.4 | 46.7 | 0.815 | 0.673 |
| 162 | 02218565 | 14.7 | 16.4 | EPAE | 14.0 | 23.9 | 30.9 | 40.2 | 47.3 | 54.4 | 70.8 | 0.600 | 0.636 |
| 163 | 02334885 | 121.7 | 17.8 | EPAE | 50.4 | 86.9 | 115.5 | 155.7 | 188.3 | 224.0 | 317.1 | 0.570 | 0.642 |
| 164 | 02335347 | 0.5 | 32.1 | EPAE | 4.2 | 5.9 | 6.9 | 8.0 | 8.7 | 9.4 | 10.8 | 0.819 | 0.705 |
| 165 | 02335700 | 186.5 | 14.5 | EPAE | 57.8 | 100.5 | 132.2 | 175.0 | 208.4 | 243.2 | 328.5 | 0.555 | 0.634 |
| 166 | 02335870 | 79.5 | 20.3 | EPAE | 105.9 | 162.3 | 199.9 | 247.8 | 282.6 | 317.1 | 396.4 | 0.632 | 0.633 |
| 167 | 02336080 | 49.5 | 33.1 | EPAE | 61.7 | 68.0 | 72.5 | 78.4 | 83.0 | 87.8 | 100.0 | 0.684 | 0.684 |
| 168 | 02336102 | 6.0 | 22 | EPAE | 20.5 | 25.6 | 28.6 | 32.0 | 34.3 | 36.5 | 41.3 | 0.665 | 0.622 |
| 169 | 02336238 | 2.4 | 20.9 | EPAE | 16.6 | 20.9 | 23.9 | 27.9 | 30.9 | 34.0 | 41.9 | 0.740 | 0.629 |
| 170 | 02336300 | 224.8 | 31 | EPAE | 182.9 | 231.3 | 261.9 | 300.2 | 328.5 | 354.0 | 419.1 | 0.711 | 0.679 |
| 171 | 02336360 | 68.9 | 27.9 | EPAE | 69.1 | 88.9 | 101.9 | 117.8 | 129.7 | 141.6 | 169.1 | 0.674 | 0.662 |
| 172 | 02336635 | 81.6 | 19.9 | EPAE | 86.1 | 150.4 | 203.3 | 281.2 | 348.3 | 424.8 | 640.0 | 0.617 | 0.636 |
| 173 | 02336700 | 1.8 | 17.2 | EPAE | 8.5 | 10.8 | 12.2 | 13.8 | 15.0 | 16.1 | 18.5 | 0.681 | 0.612 |
| 174 | 02336705 | 22.8 | 19.6 | EPAE | 76.2 | 98.8 | 113.0 | 130.5 | 143.0 | 155.7 | 183.8 | 0.624 | 0.620 |
| 175 | 02341548 | 4.1 | 21 | EPAE | 11.6 | 16.3 | 19.3 | 23.1 | 26.0 | 28.9 | 35.7 | 0.718 | 0.666 |
| 176 | 02392975 | 87.0 | 24.2 | EPAE | 52.1 | 85.5 | 116.1 | 166.8 | 215.2 | 274.1 | 467.2 | 0.649 | 0.665 |
| 177 | 02395990 | 0.9 | 13.2 | EPAE | 3.1 | 4.6 | 5.5 | 6.6 | 7.3 | 7.9 | 9.3 | 0.734 | 0.615 |
| 178 | 02396550 | 0.6 | 25.1 | EPAE | 4.0 | 4.8 | 5.4 | 5.9 | 6.3 | 6.7 | 7.4 | 0.877 | 0.678 |
| 179 | 03165200 | 2.8 | 33.69 | VA | 2.6 | 4.9 | 6.7 | 9.4 | 11.8 | 14.3 | 21.2 | 0.492 | 0.609 |
| 180 | 03167300 | 1.7 | 10.01 | VA | 1.8 | 3.2 | 4.3 | 6.0 | 7.4 | 8.9 | 13.0 | 0.414 | 0.607 |
| 181 | 03167700 | 11.6 | 20.1 | VA | 9.1 | 14.3 | 18.4 | 24.5 | 29.7 | 35.5 | 51.7 | 0.456 | 0.605 |
| 182 | 03177700 | 103.0 | 26.83 | VA | 19.9 | 26.6 | 31.1 | 36.8 | 41.2 | 45.6 | 56.4 | 0.441 | 0.649 |
| 183 | 03177710 | 114.7 | 25.75 | VA | 21.0 | 29.1 | 34.6 | 41.5 | 46.7 | 52.0 | 64.4 | 0.364 | 0.610 |
| 184 | 03474700 | 21.2 | 16 | VA | 6.7 | 12.6 | 18.1 | 27.4 | 36.4 | 47.6 | 84.6 | 0.437 | 0.683 |
| 185 | 03474800 | 20.3 | 10.15 | VA | 9.6 | 21.2 | 31.2 | 46.3 | 59.0 | 72.9 | 109.8 | 0.419 | 0.693 |

| | | | | | | | | | | | | | |
|---|---|---|---|---|---|---|---|---|---|---|---|---|---|
| 186 | 03475600 | 8.9 | 20.51 | VA | 1.2 | 1.5 | 1.8 | 2.2 | 2.5 | 2.8 | 3.7 | 0.557 | 0.715 |
| 187 | 03475700 | 7.2 | 10.06 | VA | 3.8 | 6.4 | 8.4 | 11.0 | 13.2 | 15.4 | 20.9 | 0.522 | 0.714 |
| 188 | 03478400 | 69.8 | 14.42 | VA | 12.5 | 18.7 | 23.3 | 29.7 | 34.9 | 40.5 | 55.1 | 0.481 | 0.702 |
| 189 | 03524500 | 225.7 | 15.03 | VA | 78.6 | 125.4 | 166.4 | 232.1 | 292.8 | 365.3 | 591.8 | 0.288 | 0.623 |
| 190 | 03525800 | 1.8 | 16.53 | VA | 2.6 | 3.6 | 4.2 | 5.1 | 5.8 | 6.5 | 8.2 | 0.397 | 0.605 |
| 191 | 03530000 | 103.0 | 12.06 | VA | 65.0 | 91.5 | 109.3 | 131.8 | 148.6 | 165.4 | 205.3 | 0.255 | 0.599 |
| 192 | 06893300 | 68.6 | 36.3 | MO | 120.1 | 177.3 | 219.5 | 277.5 | 322.8 | 373.8 | 504.0 | 0.807 | 0.716 |
| 193 | 06893500 | 476.6 | 16.8 | MO | 286.0 | 447.4 | 574.8 | 761.7 | 920.3 | 1095.9 | 1588.6 | 0.640 | 0.691 |
| 194 | 06893560 | 40.4 | 32.7 | MO | 89.2 | 165.7 | 238.7 | 362.5 | 487.0 | 640.0 | 1152.5 | 0.844 | 0.708 |
| 195 | 06893562 | 46.9 | 33.1 | MO | 122.3 | 192.3 | 261.4 | 382.3 | 506.9 | 665.4 | 1237.4 | 0.844 | 0.711 |
| 196 | 06893600 | 12.9 | 31.2 | MO | 42.2 | 62.6 | 75.0 | 89.5 | 99.1 | 107.9 | 125.7 | 0.834 | 0.690 |
| 197 | 06894000 | 489.5 | 14.1 | MO | 140.7 | 245.8 | 334.1 | 475.7 | 597.5 | 741.9 | 1163.8 | 0.581 | 0.671 |
| 198 | 06910200 | 3.0 | 8.48 | MO | 9.4 | 15.4 | 19.5 | 24.8 | 28.6 | 32.6 | 41.3 | 0.477 | 0.638 |
| 199 | 06910230 | 182.1 | 5.15 | MO | 122.0 | 189.4 | 238.4 | 305.8 | 356.8 | 410.6 | 546.5 | 0.452 | 0.639 |
| 200 | 06910430 | 1.3 | 2.33 | MO | 2.6 | 5.5 | 8.4 | 13.6 | 18.9 | 25.7 | 49.0 | 0.672 | 0.625 |
| 201 | 06923000 | 2.0 | 14 | MO | 5.4 | 8.7 | 11.2 | 14.8 | 17.8 | 20.9 | 29.4 | 0.813 | 0.702 |
| 202 | 06929000 | 2.8 | 11.5 | MO | 5.6 | 11.8 | 17.3 | 25.9 | 33.4 | 41.9 | 66.3 | 0.526 | 0.669 |
| 203 | 06935800 | 2.0 | 25.2 | MO | 11.4 | 18.3 | 22.9 | 28.6 | 32.6 | 36.5 | 45.0 | 0.790 | 0.669 |
| 204 | 06935850 | 16.8 | 29.5 | MO | 39.4 | 54.1 | 64.6 | 78.4 | 89.2 | 100.5 | 129.1 | 0.744 | 0.664 |
| 205 | 06935890 | 57.5 | 26 | MO | 71.1 | 120.3 | 160.6 | 220.0 | 271.3 | 328.5 | 487.0 | 0.695 | 0.661 |
| 206 | 06935955 | 30.3 | 39.8 | MO | 62.3 | 91.2 | 110.7 | 135.4 | 153.8 | 172.2 | 215.5 | 0.847 | 0.734 |
| 207 | 06935980 | 8.8 | 42.3 | MO | 49.6 | 74.5 | 90.6 | 109.9 | 123.7 | 136.8 | 165.4 | 0.874 | 0.747 |
| 208 | 06936475 | 106.2 | 40 | MO | 149.5 | 205.9 | 239.6 | 278.6 | 305.8 | 331.3 | 382.3 | 0.826 | 0.625 |
| 209 | 07005000 | 61.6 | 32.8 | MO | 167.1 | 237.6 | 288.8 | 354.0 | 407.8 | 461.6 | 600.3 | 0.806 | 0.700 |
| 210 | 07010022 | 22.6 | 43.5 | MO | 92.3 | 116.4 | 130.8 | 147.5 | 159.1 | 170.2 | 194.0 | 0.876 | 0.737 |
| 211 | 07010030 | 5.0 | 26.5 | MO | 22.1 | 34.3 | 43.3 | 55.5 | 65.1 | 75.3 | 101.4 | 0.784 | 0.655 |
| 212 | 07010035 | 3.8 | 27.3 | MO | 18.1 | 30.6 | 38.8 | 48.7 | 55.2 | 61.4 | 74.2 | 0.757 | 0.669 |
| 213 | 07010086 | 94.0 | 31.6 | MO | 123.2 | 168.8 | 204.2 | 255.7 | 300.2 | 348.3 | 481.4 | 0.846 | 0.689 |
| 214 | 07010090 | 9.1 | 35.5 | MO | 35.7 | 46.4 | 52.4 | 58.0 | 61.7 | 64.6 | 70.2 | 0.808 | 0.699 |
| 215 | 07010180 | 47.1 | 37.2 | MO | 94.6 | 120.1 | 137.9 | 160.8 | 179.0 | 197.4 | 243.2 | 0.842 | 0.696 |
| 216 | 07010208 | 5.8 | 37.7 | MO | 30.0 | 44.2 | 53.8 | 66.3 | 75.9 | 85.2 | 108.2 | 0.858 | 0.714 |
| 217 | 07019317 | 20.4 | 39.4 | MO | 114.1 | 165.4 | 201.9 | 251.5 | 288.8 | 331.3 | 436.1 | 0.847 | 0.700 |
| 218 | 07048490 | 3.3 | 35.7 | MO | 18.1 | 25.8 | 31.4 | 39.4 | 45.6 | 52.4 | 69.9 | 0.813 | 0.695 |
| 219 | 07052000 | 50.5 | 46 | MO | 77.9 | 124.0 | 162.5 | 221.4 | 274.1 | 334.1 | 512.5 | 0.834 | 0.755 |
| 220 | 07052100 | 91.4 | 35.6 | MO | 69.4 | 111.9 | 150.1 | 212.9 | 271.8 | 342.6 | 572.0 | 0.736 | 0.725 |
| 221 | 07052160 | 151.3 | 17.8 | MO | 76.5 | 119.5 | 155.2 | 209.5 | 257.1 | 311.5 | 470.1 | 0.699 | 0.706 |
| 222 | 07063200 | 0.7 | 7.94 | MO | 3.5 | 5.9 | 7.6 | 9.6 | 11.1 | 12.5 | 15.6 | 0.685 | 0.678 |
| 223 | 07186600 | 109.6 | 13 | MO | 56.1 | 82.4 | 99.4 | 120.6 | 135.9 | 150.6 | 184.1 | 0.589 | 0.677 |
| 224 | 07195000 | 335.9 | 8.67 | MO | 156.3 | 297.3 | 407.8 | 560.7 | 682.4 | 812.7 | 1135.5 | 0.559 | 0.652 |
| 225 | 07195865 | 50.5 | 8.35 | MO | 48.1 | 81.3 | 107.0 | 143.3 | 173.0 | 205.3 | 288.8 | 0.541 | 0.631 |

 **Appendix B**

**Table B1. Manning's roughness coefficients associated to the different NLCD land-use/land-cover types (adapted from Liu & De Smedt, 2004; Hooke et al., 2021).**

| Land-use/Land-cover | n |
|---|---|
| Developed, Open Space | 0.12 |
| Developed, Low Intensity | 0.1 |
| Developed, Medium Intensity | 0.07 |
| Developed, High Intensity | 0.02 |
| Barren Land | 0.1 |
| Deciduous Needleleaf Forest | 0.4 |
| Evergreen Needleleaf Forest | 0.4 |
| Mixed Forest | 0.55 |
| Evergreen Broadleaf Forest | 0.6 |
| Deciduous Broadleaf Forest | 0.8 |
| Shrubs/Scrubs | 0.4 |
| Herbaceous | 0.3 |
| Hay/Pasture | 0.3 |
| Cultivated Crops | 0.35 |
| Woody Wetlands | 0.5 |
| Emergent Herbaceous Wetlands | 0.5 |

**Appendix C**

**Table C1. Curve Numbers associated to NLCD land-use/land-cover types and hydrologic soil groups (adapted from Wu et al., 2024).**

| Land-use/Land-cover | Hydrologic Soil Group | | | |
|---|---|---|---|---|
| | A | B | C | D |
| Developed, Open Space | 45 | 65 | 76 | 82 |
| Developed, Low Intensity | 60 | 74 | 82 | 86 |
| Developed, Medium Intensity | 77 | 85 | 90 | 92 |
| Developed, High Intensity | 92 | 94 | 96 | 96 |
| Barren Land | 77 | 86 | 91 | 94 |
| Deciduous Needleleaf Forest | 45 | 66 | 77 | 83 |
| Evergreen Needleleaf Forest | 30 | 55 | 70 | 77 |
| Mixed Forest | 36 | 60 | 73 | 79 |
| Evergreen Broadleaf Forest | 30 | 55 | 70 | 77 |
| Deciduous Broadleaf Forest | 45 | 66 | 77 | 83 |
| Shrubs/Scrubs | 33 | 42 | 55 | 62 |
| Herbaceous | 30 | 58 | 71 | 78 |
| Hay/Pasture | 49 | 69 | 79 | 84 |
| Cultivated Crops | 62 | 75 | 83 | 87 |
| Woody Wetlands | 78 | 78 | 78 | 78 |
| Emergent Herbaceous Wetlands | 85 | 85 | 85 | 85 |

**Appendix D**

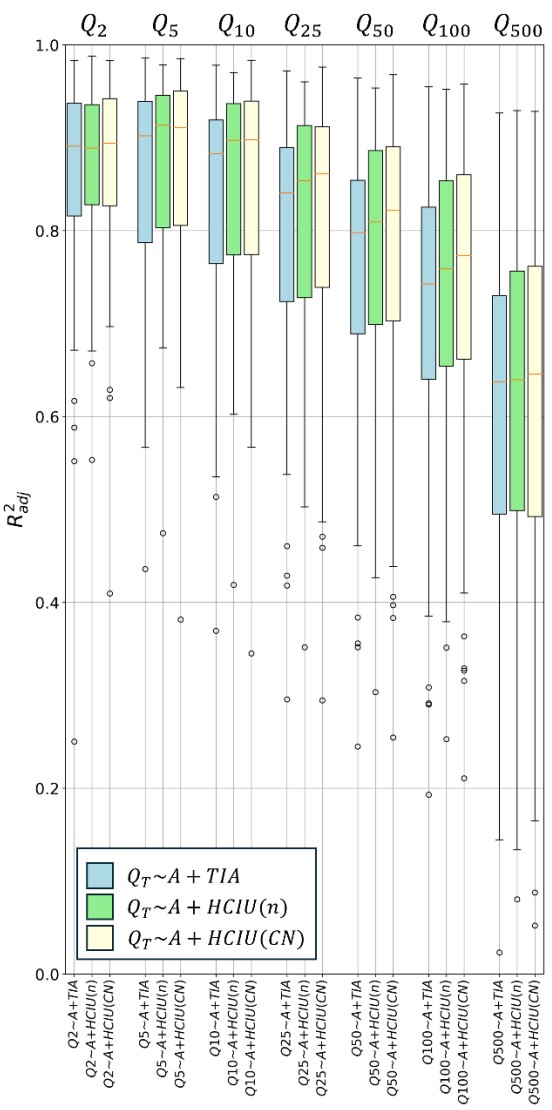

**Figure D1. Boxplots of test errors for 1) the benchmark $Q_T \sim A + TIA$, 2) $Q_T \sim A + HCIU(n)$, and 3) $Q_T \sim A + HCIU(CN)$ models, when they are fitted and blind-tested on distinct basin subsets, for the VA homogenous region. For each boxplot, filled bars represent**
**error values between the first and the third quartiles, the upper (lower) whisker extends from the third (first) quartile by adding (subtracting) 1.5 times the interquartile range, and any outliers beyond the whiskers are marked as circles.**

**Appendix E**

**Table E1. Outlet coordinates, referred to the World Geodetic System (WGS84).**

|   | Gauge ID | Case Study | Longitude | Latitude |
|---|----------|------------|-----------|----------|
| 1 | 01613900 | VA | -78.288059 | 39.214548 |
| 2 | 01615000 | VA | -78.078333 | 39.174722 |
| 3 | 01616000 | VA | -78.085833 | 39.177882 |
| 4 | 01621450 | VA | -78.917806 | 38.391793 |
| 5 | 01623000 | VA | -79.126142 | 38.166797 |
| 6 | 01623500 | VA | -79.117809 | 38.183463 |
| 7 | 01624800 | VA | -78.994471 | 38.128467 |
| 8 | 01625000 | VA | -78.861970 | 38.261796 |
| 9 | 01626000 | VA | -78.908079 | 38.057636 |
| 10 | 01626500 | VA | -78.896968 | 38.061247 |
| 11 | 01626850 | VA | -78.876968 | 38.088746 |

| 12 | 01627500 | VA | -78.836692 | 38.218742 |
|---|---|---|---|---|
| 13 | 01628500 | VA | -78.754746 | 38.322628 |
| 14 | 01629500 | VA | -78.534733 | 38.646231 |
| 15 | 01631000 | VA | -78.210834 | 38.914001 |
| 16 | 01636210 | VA | -78.185833 | 38.905667 |
| 17 | 0163626650 | VA | -78.128056 | 38.934167 |
| 18 | 01638350 | VA | -77.615444 | 39.191111 |
| 19 | 01643805 | VA | -77.683944 | 39.072306 |
| 20 | 01644280 | VA | -77.432389 | 39.046417 |
| 21 | 01644290 | VA | -77.371375 | 38.949277 |
| 22 | 01644291 | VA | -77.373041 | 38.949833 |
| 23 | 01644295 | VA | -77.367486 | 38.952888 |
| 24 | 01644300 | VA | -77.371097 | 38.966777 |
| 25 | 01645700 | VA | -77.338041 | 38.874834 |
| 26 | 01645750 | VA | -77.353041 | 38.897889 |
| 27 | 01645784 | VA | -77.344985 | 38.930111 |
| 28 | 01645900 | VA | -77.309706 | 38.965666 |
| 29 | 01645975 | VA | -77.246648 | 38.971221 |
| 30 | 01646000 | VA | -77.245814 | 38.975943 |
| 31 | 01646200 | VA | -77.205536 | 38.958999 |
| 32 | 01646600 | VA | -77.184425 | 38.911500 |
| 33 | 01646700 | VA | -77.139146 | 38.936222 |
| 34 | 01646750 | VA | -77.137757 | 38.905111 |
| 35 | 01646800 | VA | -77.144979 | 38.922889 |
| 36 | 01652400 | VA | -77.126646 | 38.858723 |
| 37 | 01652430 | VA | -77.102201 | 38.861779 |
| 38 | 01652470 | VA | -77.104145 | 38.842613 |
| 39 | 01652500 | VA | -77.085861 | 38.843333 |
| 40 | 01652600 | VA | -77.212204 | 38.865945 |
| 41 | 01652610 | VA | -77.174147 | 38.846501 |
| 42 | 01652620 | VA | -77.178869 | 38.879556 |
| 43 | 01652645 | VA | -77.170814 | 38.865112 |
| 44 | 01652650 | VA | -77.165536 | 38.860390 |
| 45 | 01652710 | VA | -77.186926 | 38.801502 |
| 46 | 01652810 | VA | -77.151369 | 38.810113 |
| 47 | 01652910 | VA | -77.127757 | 38.803169 |
| 48 | 01653000 | VA | -77.105590 | 38.804447 |
| 49 | 01653210 | VA | -77.083589 | 38.793169 |
| 50 | 01653447 | VA | -77.064700 | 38.788725 |
| 51 | 01653900 | VA | -77.271094 | 38.860945 |
| 52 | 01653950 | VA | -77.242482 | 38.873167 |
| 53 | 01654000 | VA | -77.228316 | 38.812891 |
| 54 | 01654500 | VA | -77.236455 | 38.813585 |
| 55 | 01655000 | VA | -77.202204 | 38.754281 |
| 56 | 01655310 | VA | -77.288318 | 38.801780 |
| 57 | 01655350 | VA | -77.226650 | 38.757336 |
| 58 | 01655370 | VA | -77.233872 | 38.750392 |
| 59 | 01655380 | VA | -77.252484 | 38.736503 |
| 60 | 01655390 | VA | -77.214149 | 38.704004 |
| 61 | 01656800 | VA | -77.466656 | 38.908445 |

| 62 | 01656960 | VA | -77.465544 | 38.821225 |
|---|---|---|---|---|
| 63 | 01657000 | VA | -77.457489 | 38.797892 |
| 64 | 01657415 | VA | -77.414154 | 38.766504 |
| 65 | 01657500 | VA | -77.326096 | 38.705671 |
| 66 | 01657655 | VA | -77.289984 | 38.680116 |
| 67 | 01657800 | VA | -77.226371 | 38.680116 |
| 68 | 01667600 | VA | -78.006664 | 38.397351 |
| 69 | 01673500 | VA | -77.382481 | 37.669311 |
| 70 | 01673550 | VA | -77.257755 | 37.662643 |
| 71 | 02019400 | VA | -79.757540 | 37.496801 |
| 72 | 02027700 | VA | -78.959466 | 37.562642 |
| 73 | 02030800 | VA | -78.808077 | 38.030138 |
| 74 | 02031000 | VA | -78.592794 | 38.102636 |
| 75 | 02033500 | VA | -78.453344 | 38.019306 |
| 76 | 02034000 | VA | -78.265837 | 37.857919 |
| 77 | 02034050 | VA | -78.241393 | 37.946807 |
| 78 | 02037800 | VA | -77.588600 | 37.454315 |
| 79 | 02038000 | VA | -77.522209 | 37.443759 |
| 80 | 02038500 | VA | -77.466373 | 37.461259 |
| 81 | 02042000 | VA | -77.494153 | 37.315428 |
| 82 | 02042287 | VA | -77.421649 | 37.641811 |
| 83 | 02042426 | VA | -77.424149 | 37.613201 |
| 84 | 02042500 | VA | -77.060803 | 37.436258 |
| 85 | 02042780 | VA | -76.766904 | 37.314036 |
| 86 | 02044400 | VA | -77.981670 | 37.079872 |
| 87 | 02055000 | VA | -79.938648 | 37.258471 |
| 88 | 02055100 | VA | -79.935319 | 37.417633 |
| 89 | 02056000 | VA | -79.871425 | 37.255138 |
| 90 | 02056650 | VA | -79.868091 | 37.227639 |
| 91 | 02057500 | VA | -79.521418 | 37.034312 |
| 92 | 02057700 | VA | -79.873366 | 37.007363 |
| 93 | 02059000 | VA | -79.473083 | 37.012923 |
| 94 | 02059450 | VA | -79.730315 | 37.379859 |
| 95 | 02060500 | VA | -79.285639 | 37.105694 |
| 96 | 02061150 | VA | -79.387530 | 37.369586 |
| 97 | 02062500 | VA | -78.945722 | 37.039444 |
| 98 | 02076400 | VA | -79.369191 | 36.933474 |
| 99 | 02076500 | VA | -79.311412 | 36.936529 |
| 100 | 02086849 | EPAE | -78.832296 | 36.059583 |
| 101 | 0208726005 | EPAE | -78.724530 | 35.845440 |
| 102 | 02087324 | EPAE | -78.611423 | 35.810929 |
| 103 | 0208732885 | EPAE | -78.5593078 | 35.816968 |
| 104 | 02087359 | EPAE | -78.583059 | 35.758416 |
| 105 | 02087580 | EPAE | -78.752249 | 35.718821 |
| 106 | 0209399200 | EPAE | -79.860069 | 36.137849 |
| 107 | 02094659 | EPAE | -79.855270 | 36.049536 |
| 108 | 02094770 | EPAE | -79.799742 | 36.037715 |
| 109 | 02095000 | EPAE | -79.725462 | 36.059935 |
| 110 | 02095271 | EPAE | -79.782466 | 36.097823 |
| 111 | 02095500 | EPAE | -79.708534 | 36.120195 |

| 112 | 0209553650 | EPAE | -79.661672 | 36.128122 |
|-----|------------|------|------------|-----------|
| 113 | 0209741955 | EPAE | -78.912984 | 35.872341 |
| 114 | 02115845 | EPAE | -80.257807 | 36.084298 |
| 115 | 0212414900 | EPAE | -80.715874 | 35.332301 |
| 116 | 0214266000 | EPAE | -80.921152 | 35.389568 |
| 117 | 02142900 | EPAE | -80.909624 | 35.328629 |
| 118 | 0214291555 | EPAE | -80.973052 | 35.300436 |
| 119 | 0214295600 | EPAE | -80.974610 | 35.240307 |
| 120 | 02145940 | EPAE | -81.016238 | 34.974710 |
| 121 | 02146211 | EPAE | -80.836955 | 35.262057 |
| 122 | 0214627970 | EPAE | -80.868234 | 35.240339 |
| 123 | 02146300 | EPAE | -80.904579 | 35.197899 |
| 124 | 02146315 | EPAE | -80.921902 | 35.206679 |
| 125 | 02146348 | EPAE | -80.927050 | 35.145767 |
| 126 | 02146381 | EPAE | -80.899248 | 35.090795 |
| 127 | 02146409 | EPAE | -80.837113 | 35.203642 |
| 128 | 0214642825 | EPAE | -80.770919 | 35.235958 |
| 129 | 0214645022 | EPAE | -80.831099 | 35.175358 |
| 130 | 02146470 | EPAE | -80.853095 | 35.164402 |
| 131 | 02146500 | EPAE | -80.854723 | 35.153631 |
| 132 | 02146507 | EPAE | -80.857844 | 35.148087 |
| 133 | 02146530 | EPAE | -80.882211 | 35.085094 |
| 134 | 0214655255 | EPAE | -80.719311 | 35.176025 |
| 135 | 02146562 | EPAE | -80.736609 | 35.186742 |
| 136 | 02146600 | EPAE | -80.767469 | 35.137760 |
| 137 | 02146700 | EPAE | -80.820040 | 35.140830 |
| 138 | 02146750 | EPAE | -80.869807 | 35.066373 |
| 139 | 0214678175 | EPAE | -80.953677 | 35.105022 |
| 140 | 02159785 | EPAE | -81.965937 | 34.952698 |
| 141 | 02160325 | EPAE | -82.301249 | 34.883480 |
| 142 | 02164000 | EPAE | -82.364644 | 34.800787 |
| 143 | 02164011 | EPAE | -82.407097 | 34.823811 |
| 144 | 02168845 | EPAE | -81.141053 | 34.040544 |
| 145 | 02203800 | EPAE | -84.308194 | 33.679573 |
| 146 | 02203835 | EPAE | -84.280408 | 33.746991 |
| 147 | 02203845 | EPAE | -84.262407 | 33.718109 |
| 148 | 02203884 | EPAE | -84.343674 | 33.635721 |
| 149 | 02203900 | EPAE | -84.223998 | 33.665809 |
| 150 | 02204070 | EPAE | -84.128472 | 33.630024 |
| 151 | 02205000 | EPAE | -84.004956 | 34.001973 |
| 152 | 02205230 | EPAE | -84.049231 | 34.001324 |
| 153 | 02205500 | EPAE | -84.016399 | 33.934720 |
| 154 | 02205596 | EPAE | -84.045993 | 33.912660 |
| 155 | 02206105 | EPAE | -84.211234 | 33.886614 |
| 156 | 02206136 | EPAE | -84.182610 | 33.888577 |
| 157 | 02206500 | EPAE | -84.078344 | 33.853347 |
| 158 | 02207000 | EPAE | -84.097350 | 33.861842 |
| 159 | 02207500 | EPAE | -83.914991 | 33.614607 |
| 160 | 02208050 | EPAE | -83.939169 | 33.978529 |
| 161 | 02217505 | EPAE | -83.401920 | 33.942363 |

| 162 | 02218565 | EPAE | -83.894082 | 34.010278 |
|---|---|---|---|---|
| 163 | 02334885 | EPAE | -84.088839 | 34.032626 |
| 164 | 02335347 | EPAE | -84.245228 | 33.956740 |
| 165 | 02335700 | EPAE | -84.269479 | 34.050537 |
| 166 | 02335870 | EPAE | -84.443359 | 33.953863 |
| 167 | 02336080 | EPAE | -84.286783 | 33.862050 |
| 168 | 02336102 | EPAE | -84.321663 | 33.855632 |
| 169 | 02336238 | EPAE | -84.343977 | 33.794878 |
| 170 | 02336300 | EPAE | -84.407689 | 33.820352 |
| 171 | 02336360 | EPAE | -84.378859 | 33.869173 |
| 172 | 02336635 | EPAE | -84.521394 | 33.803291 |
| 173 | 02336700 | EPAE | -84.467892 | 33.690876 |
| 174 | 02336705 | EPAE | -84.486349 | 33.715874 |
| 175 | 02341548 | EPAE | -84.938960 | 32.526251 |
| 176 | 02392975 | EPAE | -84.535676 | 34.068328 |
| 177 | 02395990 | EPAE | -85.138415 | 34.267345 |
| 178 | 02396550 | EPAE | -85.162184 | 34.232384 |
| 179 | 03165200 | VA | -80.900355 | 36.677350 |
| 180 | 03167300 | VA | -80.578677 | 36.837909 |
| 181 | 03167700 | VA | -80.725628 | 36.768184 |
| 182 | 03177700 | VA | -81.281766 | 37.255950 |
| 183 | 03177710 | VA | -81.304823 | 37.271506 |
| 184 | 03474700 | VA | -81.734565 | 36.783448 |
| 185 | 03474800 | VA | -81.804011 | 36.763169 |
| 186 | 03475600 | VA | -81.855402 | 36.747335 |
| 187 | 03475700 | VA | -82.041239 | 36.678721 |
| 188 | 03478400 | VA | -82.133743 | 36.631774 |
| 189 | 03524500 | VA | -82.456262 | 36.929269 |
| 190 | 03525800 | VA | -82.210970 | 36.830661 |
| 191 | 03530000 | VA | -82.770994 | 36.865097 |
| 192 | 06893300 | MO | -94.671300 | 38.940800 |
| 193 | 06893500 | MO | -94.559225 | 38.957112 |
| 194 | 06893560 | MO | -94.585223 | 39.039949 |
| 195 | 06893562 | MO | -94.578711 | 39.038983 |
| 196 | 06893600 | MO | -94.451000 | 39.076900 |
| 197 | 06894000 | MO | -94.300753 | 39.100543 |
| 198 | 06910200 | MO | -92.323627 | 39.002679 |
| 199 | 06910230 | MO | -92.340000 | 38.927900 |
| 200 | 06910430 | MO | -92.278945 | 38.578933 |
| 201 | 06923000 | MO | -92.913113 | 37.347462 |
| 202 | 06929000 | MO | -91.953400 | 37.323700 |
| 203 | 06935800 | MO | -90.583937 | 38.618272 |
| 204 | 06935850 | MO | -90.526652 | 38.646307 |
| 205 | 06935890 | MO | -90.488982 | 38.682701 |
| 206 | 06935955 | MO | -90.447450 | 38.728081 |
| 207 | 06935980 | MO | -90.432829 | 38.764208 |
| 208 | 06936475 | MO | -90.251215 | 38.818156 |
| 209 | 07005000 | MO | -90.226277 | 38.736631 |
| 210 | 07010022 | MO | -90.323740 | 38.668242 |
| 211 | 07010030 | MO | -90.314768 | 38.676892 |

| 212 | 07010035 | MO | -90.302848 | 38.682617 |
|-----|----------|----|-----------|-----------|
| 213 | 07010086 | MO | -90.326161 | 38.601214 |
| 214 | 07010090 | MO | -90.323566 | 38.576776 |
| 215 | 07010180 | MO | -90.299632 | 38.526898 |
| 216 | 07010208 | MO | -90.292979 | 38.490848 |
| 217 | 07019317 | MO | -90.341065 | 38.483307 |
| 218 | 07048490 | MO | -94.162300 | 36.048400 |
| 219 | 07052000 | MO | -93.331146 | 37.186689 |
| 220 | 07052100 | MO | -93.370200 | 37.168500 |
| 221 | 07052160 | MO | -93.404186 | 37.117840 |
| 222 | 07063200 | MO | -90.430922 | 36.784024 |
| 223 | 07186600 | MO | -94.582200 | 37.121100 |
| 224 | 07195000 | MO | -94.288400 | 36.222000 |
| 225 | 07195865 | MO | -94.605200 | 36.201800 |

**Code and data availability**

An open-source, Python-based version of the code used in this paper is available from https://github.com/dllaira/HCIU-urbanization-metric. Basin boundaries for the MO and EPAE cases studies were obtained from https://streamstats.usgs.gov/ss/?BP=submitBatch (U.S. Geological Survey, 2019), using the outlet coordinates given in Appendix E, while those for the VA case study from https://www.sciencebase.gov/catalog/item/631405e3d34e36012efa34bf (Krstolic, 2006). Digital elevation model maps were retrieved from https://www.sciencebase.gov/catalog/item/4f70aa9fe4b058caae3f8de5 (U.S. Geological Survey, 2023). The Land-use/Land-cover maps are available from https://www.mrlc.gov/data (Homer et al., 2020) and https://www.esa-landcover-cci.org/ (ESA, 2017). The map of hydrologic soil groups can be found at https://daac.ornl.gov/SOILS/guides/Global_Hydrologic_Soil_Group.html (Ross et al., 2018). Stream network data and headwater locations were retrieved from https://www.usgs.gov/national-hydrography/nhdplus-high-resolution (Moore et al., 2019).

**Author contributions**

CM proposed the idea to derive a new urbanization metric that is sensitive to the spatial arrangement of surface patches with different characteristics. FD suggested considering hydrologic connectivity as the basis for deriving a descriptor with this property and formulated the definition of *HCIU*. FD prepared the case studies and software used in this work, and wrote the manuscript, which was revised and edited by CM. CM supervised and secured the funding to support this work.

**Competing interests**

The authors declare that they have no conflict of interest.

**Acknowledgements**

This work was supported by the Tennessee Department of Transportation (TDOT) Long Range Planning Research Office and the U.S. Federal Highway Administration through the Project "Updating Equations for Peak Flow Estimation in Urban Creeks and Streams of Tennessee" (Project Number: RES 2020-23). The Authors thank Wesley Peck at TDOT for his continuous

support throughout the research project. The Authors also express gratitude to Pete McCarthy and Katharine Kolb at the USGS for their help in assembling the data for the VA and EPAE case studies, to Dr. Cristián Chadwick, who served as an external reviewer for the manuscript, and to the two anonymous reviewers, who provided insightful comments that helped better place the proposed methodology in the context of existing literature on the connectivity index, as well as more exhaustively characterize its strengths and limitations. Finally, the Authors acknowledge the technical support of Eric J. Spangler and Kristian Skjervold for the use of the High-Performance Computing resources at University of Memphis.

The work was performed using GIS software and a range of Python libraries (Van Rossum, 1995), including ArcGIS Pro and ArcPy (ESRI, 2024) , QGIS (QGIS Development Team, 2024), Rasterio (Gillies, 2024), Numpy (Van Der Walt et al., 2011), Pandas (Mckinney, 2011), Scikit-learn (Pedregosa et al., 2011), Geopandas (Jordahl, 2017), Shapely (Gillies, 2021), Whitebox (Lindsay, 2022), and Matplotlib (Hunter, 2007). The Flow Accumulation Algorithm proposed by Zhou et al. (2019) was adapted and implemented in the open-source version of the program.

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
