# Peer review of "Beyond Total Impervious Area: A New Lumped Descriptor of Basin-Wide Hydrologic Connectivity for Characterizing Urban Watersheds"

_EGUsphere, 2024_

## Author Comment (AC1)

**Reply to Anonymous Referee #1**

We thank the reviewer for their in-depth analysis of our work, which is clear from the comprehensiveness of their review. We are grateful for the constructive comments.

In what follows, we reply to each comment, explaining how we plan to address it in the revised manuscript.

1. **It would be beneficial if the authors made it more clear why they opted to compare their approach to TIA, compared to other more robust metrics. For example, the authors mention the use of the effective impervious area (EIA) and the directly connected impervious area (DCIA) however do not compare to these approaches. I would recommend adding some additional justification as to why the authors compared the HCIU index to TIA only.**

   *We are curious to compare HCIU with other established, more advanced urbanization metrics, and we did indeed look into EIA for this work, but ended up not including it (see the reasons below). Conversely, within the current benchmarking framework consisting of several hundred basins, we had to discard DCIA from the beginning, as reliable estimates would have required us to know the stormwater drainage infrastructure at all those watersheds, and this was impossible.*

   *Considering EIA seemed more feasible, as it only requires flow and precipitation data for each case-study basin, even though we would not be able to validate EIA estimates against DCIA. Thus, we attempted to generate EIA results, but these were ambiguous, forcing us to conclude that much additional work was required; for example, some of the issues that we ran into included:*

   *1) different literature methods for deriving EIA provide different estimates, depending on, e.g., the criteria to distinguish those smaller storm events where only the effective impervious area of a basin contributes to the overall hydrologic response (as compared to larger events, where other portions of the basin have an influence as well);*
   *2) results also vary depending on the linear regression method used to fit a model to precipitation and flow data – e.g., using either ordinary or weighted least squares; in the latter case, sources of uncertainty in precipitation and runoff observations must be considered when assigning the weights, which explicitly leads us to the other, more general problem that:*
   *3) EIA estimates are heavily influenced by our ability to accurately measure rainfall depths. Irrespective of whether such information comes from gauges or radar products, we generally have to assume uniform precipitation over the watershed (or over portions*

*of it, if the watershed is large enough to include more than one rain gauge, or more than one quadrant of the precipitation radar grid). Such assumption is not always accurate and can lead to errors in the estimation of precipitation volumes; furthermore, the error may vary widely with watershed size, as the uniformity assumption is generally more acceptable for the smaller basins. How acceptable the uniform-precipitation assumption is (and the errors incurred) for the same watershed may vary across distinct events, as it depends on the areal footprint of each storm, relative to basin size. These considerations may be affected by the type of storm (e.g., frontal vs. convective) or else by the relative location of the storm with respect to the basin.*

*We believe that these uncertainties in quantifying precipitation explain the ambiguous results in our EIA estimates, which displayed large variability depending on the selected method, sometimes resulting in EIA values greater than TIA, which is conceptually meaningless. Perhaps, due to issues similar to those we just described, many studies still use TIA as a descriptor of urbanization level, when synthesizing regional peak flow equations.*

*In contrast with EIA, TIA and HCIU only require a spatial, GIS-based characterization of the watershed, and do not depend on precipitation information. As such, comparing these two variables is conceptually more straightforward, as they represent two alternative methods to characterize the impacts of urbanization on hydrologic response starting from spatial information available for all studied basins. However, an analysis of the uncertainties in EIA estimates, investigating the above-mentioned issues, and a posterior comparison with HCIU, is a topic that we plan to address in future research.*

*In the manuscript, we will explain in the Introduction why we opted for considering TIA as a benchmark for HCIU, highlighting that these two metrics represent alternative ways to characterize the impacts of urbanization on hydrologic response starting from a spatial representation of the basin, making their comparison conceptually straightforward. We will also briefly mention the greater uncertainties with EIA estimates (especially if they cannot be validated against DCIA values obtained for the same watershed), and the need to have stormwater drainage information for each basin to be able to determine DCIA.*

2. **Although the paper is technically sound, I think that it would be better if the authors did not wait until the end of the paper to address the limitations of the proposed methodology. I think it is very important to address that this approach does not currently account for the urban drainage system. The urban drainage systems in heavily urbanized watersheds are a major runoff routing component**

**and are important for accurately evaluating hydrologic connectivity. I think it is important to address that for heavily urbanized watersheds differences in connectivity may be controlled by the presence and capacity of storm and combined sewers. This component is missing from the methodology based on the limitations addressed in the discussion. However, based on the methodological approach could be added relatively easily, which is excellent. I agree with the authors that it may be difficult to acquire sewer attribute information, however this is not this case in all Countries. In Canada for example much of the storm sewer attribute information is now being made publicly accessible via OpenData government portals. I think making this clear in the introduction may make the authors approach more accessible to individuals who may have access to this information.**

*We agree with the Reviewer and thank them for bringing up that data for storm sewer infrastructure may be available for Canadian regions. We are currently working on a more comprehensive version of HCIU that also accounts for the effects of storm sewer, therefore it is good for us to know about the opportunity to consider case-study areas from that country. To address their comment, we will include in the introduction a paragraph highlighting that the proposed methodology currently considers topography as the only driver of hydrologic connectivity, and does not yet include the effects of underground stormwater drainage infrastructure, due to the lack of stormwater pipe network data for the case-study basins. We will specify that this may be a limitation for heavily urbanized basins, but adaptations to the current methodology to incorporate the effects of the stormwater sewer network are straightforward, as explained in the Discussion section.*

3. **The proposed HCIU index is presently heavily dependent on topography, in some heavily urbanized regions (impervious cover above 80%) stormwater pumping across topographic gradients, and stormwater detention tanks may impact outcomes. In the case study component, the authors have a few catchments with high levels of imperviousness but these catchments seem to be relatively small and were only part of the VA case-study. I would suggest addressing this as a limitation in the paper.**

*We will elaborate on these limitations in the same additional paragraph addressing the previous comment. Specifically, we will highlight that the proposed methodology currently considers topography as the only driver of connectivity, which will be a limitation for highly urbanized basins, typically characterized by the presence of a dense stormwater sewer system, possibly including detention tanks and sections where stormwater may be pumped against topographic gradients. We will therefore*

*note that, for highly urbanized basins, it should be necessary to consider these additional sources of connectivity, to obtain reliable estimates of HCIU. As stated above, we will mention that adaptations to the current methodology to also incorporate the effects of the stormwater sewer network are straightforward, as explained in more detail in the Discussion section.*
*We are currently working on a more comprehensive version of HCIU, that addresses this issue.*

4. **I found the explanation and use of figure 1b, for creating a totally impervious copy to be complicated. I think it would be nice if the authors could simplify this text or provide a different graphic to support this process.**
*We thank the Reviewer for bringing this to our attention. Creating the totally impervious copy of each (actual) basin simply consists of considering a virtual copy of it, with the same shape, digital elevation model (i.e., same topography), and stream network, but replacing the actual land-use/land-cover (LULC) conditions with totally developed (i.e., 100% impervious) LULC. As a result, when calculating the raster map of the connectivity index, this 100%-impervious copy of the basin will have the maximum possible weight W at each basin cell, associated with the fully impervious conditions, and therefore the highest possible connectivity, for the given basin shape, topography, and stream network. To enhance clarity, we will make the following changes to Fig. 1b: 1) we will change the "Pave each single cell" text to "Change the actual LULC of each cell to fully impervious conditions"; 2) We will add the sub-caption "Virtual, totally impervious copy of the actual basin, with same shape, topography, and stream network, but fully impervious LULC at each cell". Figure 1b will also benefit from the expanded caption for Fig. 1, that we plan to introduce in the revised version to address the technical correction #3 (in a separate list below).*

5. **I am curious to why the authors evaluate the predictive power of their approach (HCIU) using peak flow only. Why did the authors not consider other important metrics of hydrologic response?**
*We considered peak flow because it is probably the most relevant variable as regards to urban flooding risk, and also because of the scope of the funding project, centered on improving regional equations for estimating peak flows. However, we are working on expanding the applications of HCIU, including the prediction of other hydrologic-response metrics (specifically, basin lag-time and time of concentration).*

6. **With regards to the case studies could you provide what type of flow data was used to generate the flow statistics? Daily, or sub-daily?**

*The flow data that the USGS used to generate peak-flow statistics are instantaneous flow time series, typically at a 15-min (or at most hourly) time resolution, depending on the specific USGS gaging station. Please note that we did not perform flood frequency analyses for our work: we used the flow quantiles as reported in the referenced USGS regional studies, as stated at lines 257-259.*

7. **The authors mention in the acknowledgments the need for computing power, is this a limitation of the approach? It would be nice to know what technical equipment they had access to for completing the analysis. Is this approach feasible for local governments or conservation groups to do?**
*Throughout the duration of the project, we generated increasingly faster code to perform the proposed methodology. For our analyses, we ran the computations in parallel, for multiple watersheds, instead of sequentially processing one basin at a time. To do so, we used the High-Performance Computer at the University of Memphis, allocating different basins to different processors, simultaneously. At the time of manuscript preparation, the computational time per basin varied from coffee-break duration to days, depending on size (with a few days required for basins of thousands of km2). We are developing a newer version of our code, such that computing power will no longer be a limitation and the approach will be feasible for local governments. We expect to cut down the processing times required for the largest basins (in the order of thousands of square kilometers) to a few minutes. Because newer versions of the code will be publicly shared with our future articles, we did not mention information on computational times in this manuscript, as these are expected to change with future versions of the code.*

8. **I think it could be beneficial for the authors to discuss the role of major vs minor system with regards to event size and hydrologic connectivity, especially for the peak flows represented by the extreme flood values.**
*We believe this a great suggestion, thank you.*
*In urban drainage networks, the minor system typically encompasses the underground infrastructure, including pipes and manholes (Martins et al., 2017). These systems are designed to handle frequent, smaller rainfall events and convey water efficiently to prevent localized flooding. On the other hand, the major drainage system refers to surface flow pathways and watercourses, which are critical during larger, less-frequent storm events, when the minor system's capacity may be easily exceeded (Martins et al., 2017).*
*We will stress in our manuscript that the proposed methodology, considering topography as the only driver of connectivity, does not currently account for the effects*

*of minor, underground drainage systems. However, major drainage system sections connected to the stream network are treated as part of that network, when assessing hillslope-to-stream connectivity. Our formulation for HCIU therefore captures the effects of major drainage systems on hydrologic response. Given their crucial role in mitigating larger, more extreme flood events in urban basins, HCIU is expected to be a reliable predictor for hydrologic-response variables under severe flooding conditions. We will include the distinction between minor and major drainage systems also in Section 5.2, where we discuss methodological adaptations to also incorporate the effects of the stormwater sewer network on connectivity. Specifically, we will note that effects from the minor system, mostly underground, should override the potential connectivity arising from topographic gradients. On the other hand, major systems can be regarded as part of the stream network (assuming that excess flow from the major system is poured directly at some section along the stream network), without the need for adaptations to the current methodology, to incorporate their effects. This means that the connectivity of hillslope cells draining to the major system should be calculated referring to the pour points along the major system. Then, the contributions of those hillslope cells should be weighted based on the "along-stream-network" distance to the outlet, measured starting from the major-system pour point and along both the major-system and subsequent natural-stream-network links downstream, when calculating HCIU as a weighted average of (normalized) connectivities.*

*Reference: Martins, R., Leandro, J., Chen, A. S., & Djordjević, S. (2017). A comparison of three dual drainage models: shallow water vs local inertial vs diffusive wave. Journal of Hydroinformatics, 19(3), 331-348.*

9. **Do you think the poor overall performance of the HCIU(*CN*) could be due to the quality of the raster product you are using?**
   *While the map of hydrologic soil groups that we considered has the coarsest resolution (250 m, versus 10 m for the DEM and 30 m for the LULC map), it is also the hydrologic variable with the smallest expected spatial variability, therefore we do not consider this to be the main explanation. We suspect that the main issue is that, based on existing CN tables, some developed (i.e. urbanized) cells can get CN values that are similar to those for cells with natural land covers, depending on the type of soil, which does not help in discriminating some of the flood-mitigating effects of such natural conditions. However, we will consider this comment in future work about improving HCIU(CN)'s predictive power for peak flows. We thank the Reviewer for providing this additional idea.*

**10. Line 560: This is interesting. Please elaborate on this point.**

*In line 560, we wrote* "We suggest that HCIU should also increase our explanatory power when predicting other event-related variables such as lag times and times of concentration". *We will further elaborate our statement by including the following additional considerations:* "*HCIU* is indeed sensitive not only to the presence and spatial arrangement of LULC patches with different hydrologic characteristics but also to the locations where flows tend to concentrate, locally decreasing surface runoff travel times. This is conceptually reflected in the upslope component $D_{up,k}$ (Eq. 1). *HCIU* also considers the distance of these surface runoff 'hotspots,' where stormwater tends to concentrate and travel faster, to the stream network, as reflected by the downslope component $D_{dn,k}$. This in turn determines how easily those locations with accumulating flows will contribute to the overall basin response. Ultimately, *HCIU* conceptually summarizes in a single number the effects of all potential runoff travel paths occurring on the basin surface and moving towards the stream, including interactions among converging surface flow paths, following a hydrologically driven approach. Because other response variables, such as lag time and time of concentration, are emergent basin properties arising from the interactions of all individual travel paths, we will investigate their correlations with *HCIU* and other connectivity-based descriptors in further research."

**Technical corrections:**

**1. Line 159: I would like to see some examples of LULC types.**

*To provide some examples of LULC types, we will change the text as follows:* "Among the options discussed above, when deriving HCIU we recommend choosing $W$ values that primarily depend on the LULC type of each basin cell, considering both developed and natural LULC categories (e.g., urbanized, barren, croplands, forested, etc.), possibly differentiating across distinct intensities of land-development and dominant vegetation types, for the developed and vegetated categories, respectively. In this way, the effects of pixels with different surface characteristics can be differentially weighted depending on their potentials for either generating and quickly transmitting surface runoff (e.g., in the case of developed cells) or else retaining, detaining, or infiltrating water (e.g., in the case of cells with vegetated land cover), depending on the distinct hydrologic dynamics associated with different LULC types.*"

**2. Line 124: The references are repeated multiple times.**

*To avoid reference repetitions, we will change the text as follows:* "Borselli et al. (2008) proposed a widely used GIS-based index of connectivity to assess sediment erosion and transport, which was then modified by Cavalli et al. (2013), Persichillo et al. (2018),

Zanandrea et al. (2019), Hooke et al. (2021), and Husic & Michalek (2022), among others, to focus on other basin dynamics, such as runoff generation or landslide occurrence.*"*

3. **Figure 1: Suggest expanding on your figure caption to help explain the process in more detail.**

   *Our expanded caption will be as follows: "*Figure 1: Methodological steps for obtaining the hydrologic-connectivity-based index of urbanization ($HCIU$): a) scheme for calculating Borselli et al.'s (2008) connectivity index at generic cell $k$; b) create a virtual, totally impervious copy of the basin, with the same shape, topography, and stream network, but different LULC, i.e., fully developed at all cells; c) separately calculate the raster maps of connectivity for both the actual basin and its totally impervious copy; d) calculate the raster map of normalized connectivity for the basin by dividing the connectivity of the actual basin by the connectivity of the totally impervious copy, on a cell-by cell basis; e) assign a weight $w_k$ to each basin cell $k$ depending on its distance to the outlet, as measured along the stream network, starting from the cell's pour point; f) calculate $HCIU$ as a weighted average of the normalized connectivities at each basin cell."

4. **Figure 2: What Ecoregion is this?**

   *At lines 247-252, we explain that "EPA Ecoregion" is a short name, used in our work, to refer to the hydrologically homogeneous region corresponding to the "Piedmont" and a small part of "Ridge and Valley" EPA ecoregions, consistent with the USGS case-study region from Feaster et al. (2014). In Figure 2, we used that short name for space-related reasons, as well as to be consistent with the rest of the manuscript.*

5. **Line 252-254: Suggest adding a reference to Appendix Table A1.**

   *We will add a reference to Appendix Table A1, for enhancing clarity.*

6. **Figure 4: add legend item for shaded blue bars as you have one for the basin averages.**

   *We will include the legend item for the shaded blue bars.*

7. **Line 329-330: Suggest defining the LULC ranges for the different percentages in the text as you have done in the figure.**

   *We will include that information in the text by introducing the following change: "*Figure 4 also illustrates the mix of developed LULC types in the basins, by showing the distributions (boxplots) of the extents of the four developed NLCD categories in each watershed, for the three homogenous regions (Fig. 4j, 4k, and 4l, respectively). Those categories include "Developed, Open Space", "Developed, Low Intensity", "Developed, Medium Intensity",

and "Developed, High Intensity", associated with ranges of impervious area of less than 20%, 20%-49%, 50%-79%, and 80% or more, respectively.*"

8. **Figure 6: with all the reference lines I found this confusing. Perhaps you could just color code the points you are highlighting based on the outline of the box color.**
   *We will remove the arrows and use instead letters to connect basins to points. We prefer avoiding using colors, to prevent any visualization issues in case of grayscale prints of the article.*

---

## Author Comment (AC2)

**Reply to Anonymous Referee #2**

We thank the reviewer for their valuable comments, which help better place the proposed methodology in the context of the existing literature on the connectivity index.

In what follows, we reply to each comment, explaining how we plan to address it in the revised manuscript.

1. **Lines 217-230—It is unclear how implementing an "along-the-stream network" differs from the well-known IC_outlet approach from Cavalli et al. (2013) and several other researchers/papers. It is necessary to explain why to choose this new approach over IC_outlet.**

    *In the past literature on the connectivity index, the IC_outlet metric has been proposed to directly characterize connectivity between hillslopes and catchment outlet. It is calculated using the traditional formulation by Borselli et al. (2008) (with some adaptations to the weighting coefficient and/or the flow direction algorithm adopted, depending on the specific application; see, e.g., Cavalli et al., 2013), but considering flow paths directed from each hillslope cell all the way to the outlet (hence, including paths along the stream network), instead of shorter flow paths along hillslope surfaces only, until reaching the stream network at the nearest pour point.*

    *In our methodology, we consider separately hillslope-to-stream and stream-to-outlet flow paths. This allows us to focus more on hillslope-to-stream connectivity, which is crucial when assessing the impacts of urbanization on hydrologic response. Land development primarily affects overland flow, occurring over the hillslope component of a basin. We aim to frame the connectivity between any hillslope patch (including urbanized sectors) and its nearest stream, to effectively analyze the hydrologic impacts of developed pixels, depending on their location relative to the stream network and all other pixels with different LULC types along the path to the pour point. Once runoff reaches the stream network, the effects of travel distance along the stream network must still be accounted for, but this is performed in the separate, second step, considering a narrower range for the weights. This ensures that HCIU displays adequate sensitivity to urbanized sectors that are adjacent to the stream network, but at reaches located far upstream from the outlet.*

    *Maybe more importantly, from an application perspective, we need to be able to quickly compute HCIU for any basin (in a region, country, province, state, etc.), as selected by the final user. If we were to use a "cell-to-outlet" scheme, such as the IC_outlet metric, we would need to recompute everything from scratch, every time a user chooses a different basin (i.e., a different outlet location along a stream of the river network). By splitting the*

*HCIU computations from cell to pour point, and then pour point to outlet, we can precompute all connectivities and normalized connectivities for all the pixels over large areas, once for all, irrespective of any basin and its outlet. Then, the final computation of HCIU, for any desired basin of interest (i.e., given a specific outlet along the stream network) only involves a much-quicker lumping via a weighted average of the precomputed at-a-cell normalized connectivities, only considering those cells within the basin and their along-the-stream-network distances to the desired outlet.*

*In the manuscript, we will clarify these differences by introducing the following additional considerations after Eq. 7 (i.e., after line 238 in the first version of the manuscript).*

"In summary, the proposed methodology provides a lumped metric ($HCIU$) that is able to conceptually capture the varied hydrologic effects arising from the spatial arrangement of different LULC patches, both natural and developed, depending on their relative location with respect to each other, the stream network, and the basin outlet. First, hillslope-to-stream connectivities, weighted depending on the hydrologic effects of distinct LULC types, are normalized with respect to a fully impervious benchmark (Fig. 1a, 1b, 1c, and 1d), which allows to compare the effects of heterogeneous levels of urbanization both across and within basins. Then, $HCIU$ is obtained as a weighted average of normalized connectivities across the entire watershed, assigning different weights to each pixel depending on the "along-the-stream-network" distance of that cell's pour point to the basin outlet (Fig. 1e and 1f).

The proposed two-step formulation – where the flow paths of hillslope cells to the pour points along the stream network and then the distances of those pour points to the basin outlet are considered separately – is different from other established, outlet-focused applications of the connectivity index, such as the $IC\_outlet$ distributed metric proposed by Cavalli et al. (2013). The latter is calculated following Borselli et al. (2008; with some adaptations to the weighting coefficient and the flow direction algorithm) but considering flow paths all the way to the outlet (hence, considering both overland flows and subsequent channelized flows within the same path), instead of flow paths to the closest stream link, following only hillslope surfaces. The two main components of a basin's hydrologic response, i.e., overland and channel flow, generally involve quite different temporal scales, because of the different orders of magnitude in roughness and water depths. The $IC\_outlet$ metric is able to capture these differences, as $IC\_outlet$ raster maps typically exhibit the highest connectivity values along the watershed stream network (comparable only to connectivities in the hillslope sectors closest to the outlet), followed by connectivities in zero-order valleys or hollows adjacent to channels (Cavalli et al., 2013). On the other hand, our methodology focuses on the hydrologic effects of land development, which mostly influences the overland-flow component by locally decreasing infiltration and increasing runoff speeds. Considering only the hillslope-to-stream connectivity in our first step allows

us to enhance the method's sensitivity to the effects of land development on hydrologic response, by focusing on how runoff interacts with the distinct LULC patches encountered along the hillslope path, which control (i.e., enhance or mitigate) the connectivity. Once runoff reaches the stream network, the effects of travel distance along the stream network must still be accounted for, but this is performed in the separate, second step, considering a narrower range for the weights. This ensures that $HCIU$ displays adequate sensitivity to urbanized sectors that are adjacent to the stream network, but at reaches located far upstream from the outlet.

Breaking down the calculations for $HCIU$ in two parts (the hillslope-to-stream and then stream-to-outlet flow paths) also presents a practical advantage, particularly for large-scale implementation of the index. To ensure broad applicability of the proposed methodology, we need to be able to quickly compute $HCIU$ for any basin (in a region, country, province, state, etc.), as selected by the final user. If we were to use a "cell-to-outlet" scheme, such as the $IC\_outlet$ metric, we would need to recompute everything from scratch, every time a user chooses a different basin (i.e., a different outlet location along the stream network). Splitting the computations from cell to pour point, and then pour point to outlet, offers the opportunity to precompute "static" (i.e., independent of outlet location) raster maps of connectivity and normalized connectivity, for all the pixels over large areas. In this way, later, when a user selects a specific outlet location, the final computation of $HCIU$ only involves the much-quicker weighted averaging of the precomputed at-a-cell normalized connectivities, only considering those cells within the selected basin and their along-the-stream-network distances to that desired outlet."

**Other minor issues:**

1. **Figure 4: What do the blue bars represent?**

   *They indicate the proportion (expressed in percent) of cells within a given range of $n$ (or $CN$, or $S$, depending on the considered row in the subplots), with respect to the total number of basin cells for each homogeneous region (also see lines 311-313). For instance, for the VA case study, a little more than 25% of all basin cells (from all basins of that region) have a value of $n$ between 0.7 and 0.8 (Fig. 4c). To address this as well as a comment from Reviewer 1, we will expand the legend to also include the description of the blue bars, "Proportion of cells (%)", consistent with the associated y-axis label.*

2. **No comment exists about how the urban drainage structure could affect urban hydrology.**
   *To address this and a similar comment from Reviewer 1, we will include an additional paragraph in the Introduction, highlighting that the proposed methodology currently considers topography as the only driver of hydrologic connectivity. This may be a*

*limitation for highly urbanized basins, typically characterized by the presence of a dense stormwater drainage system, possibly including detention tanks and sections where stormwater may be pumped against topographic gradients. We will therefore note that, for highly urbanized basins, it may be necessary to consider these additional sources of connectivity, to reliably obtain estimates of HCIU; we will also briefly mention that adaptations to the current methodology to incorporate the effects of the stormwater drainage network are straightforward, as explained in more detail in the Discussion section.*

---

## Author Response (AR1)

We thank the Reviewers once again for their constructive comments. We believe they have helped improve our manuscript.

Please find below a point-by-point explanation of how we addressed each comment in our revised manuscript.

**Comments made by Referee #1**

1. **It would be beneficial if the authors made it more clear why they opted to compare their approach to TIA, compared to other more robust metrics. For example, the authors mention the use of the effective impervious area (EIA) and the directly connected impervious area (DCIA) however do not compare to these approaches. I would recommend adding some additional justification as to why the authors compared the HCIU index to TIA only.**
   We addressed this comment in a new paragraph in the Introduction (lines 122-134 of the revised manuscript). "We benchmark our hydrologic-connectivity-based index of urbanization (HCIU) against the traditional fraction of TIA, by alternatively using one of these two metrics as a predictor in regional peak-flow equations for urbanized basins. Imperviousness descriptors expressed as a fraction of the total basin area (e.g., TIA, EIA, and DCIA) are still among the most popular approaches to quantify the effects of land development in lumped hydrologic and regional models (Bell et al., 2016; Yang et al., 2023). Among these, we choose TIA as a benchmark because HCIU and TIA both condense distributed surface basin information (i.e., LULC and the topographic structure, and LULC only, respectively) into a lumped urbanization metric, making their comparison conceptually straightforward. On the other hand, EIA is an indirect estimate of the impacts of urbanization, based on retrospective analyses of concurrent historic flow and precipitation data for the case-study watersheds (Ebrahimian et al., 2016b). In preliminary tests, we found much uncertainty with EIA values, possibly due to the challenges involved in reliably estimating precipitation depths across basins with varying sizes and, for the same watershed, across distinct storm events (depending, e.g., on the areal footprint and location of the storm relative to basin extent). We also discarded DCIA, as its estimation would require knowing the configuration of the stormwater sewer network for each case-study basin, which was unfeasible, as mentioned above."

2. **Although the paper is technically sound, I think that it would be better if the authors did not wait until the end of the paper to address the limitations of the proposed methodology. I think it is very important to address that this approach does not currently account for the urban drainage system. The urban drainage systems in heavily urbanized watersheds are a major runoff routing component and are important for accurately evaluating hydrologic connectivity. I think it is**

**important to address that for heavily urbanized watersheds differences in connectivity may be controlled by the presence and capacity of storm and combined sewers. This component is missing from the methodology based on the limitations addressed in the discussion. However, based on the methodological approach could be added relatively easily, which is excellent. I agree with the authors that it may be difficult to acquire sewer attribute information, however this is not this case in all Countries. In Canada for example much of the storm sewer attribute information is now being made publicly accessible via OpenData government portals. I think making this clear in the introduction may make the authors approach more accessible to individuals who may have access to this information.**

We included in the Introduction considerations about the effects of stormwater sewer infrastructure on connectivity and stressed the limitations of the current formulation of HCIU, which ignores these effects:

Lines 104-115 (Introduction): "As the traditional definition of the connectivity index only accounts for topographically induced runoff pathways (Borselli et al., 2008), additional adjustments may be needed, depending on the level of urbanization and the scope of the analysis, to also include the effects of underground stormwater drainage infrastructure, typically present in urban environments. Underground pipe flows may be regarded as an additional source of connectivity, which can alter and sometimes even reverse the connectivity induced by topography (e.g., when stormwaters are pumped against topographic gradients).

In this work, we derive a lumped metric of urbanization effects on hydrologic response, incorporating only topographically induced connectivity (i.e., neglecting any effects of underground storm sewer infrastructure), and test its performance as a predictor in regional peak-flow equations. […] While considering the additional source of connectivity introduced by the underground drainage network would be straightforward, as explained in the Discussion, we could not account for it here, because it was impossible to obtain stormwater sewer data for the hundreds of watersheds involved in our regional scale analyses."

3. **The proposed HCIU index is presently heavily dependent on topography, in some heavily urbanized regions (impervious cover above 80%) stormwater pumping across topographic gradients, and stormwater detention tanks may impact outcomes. In the case study component, the authors have a few catchments with high levels of imperviousness but these catchments seem to be relatively small and were only part of the VA case-study. I would suggest addressing this as a limitation in the paper.**

We mentioned this limitation in the subsection dedicated to the strengths and weaknesses of the proposed methodology (in the Discussion, lines 758-771): "Another scenario where it is highly recommended to explicitly consider underground connectivity is when dealing with heavily urbanized watersheds, typically characterized by the presence of extensive drainage infrastructure. In these basins, detention tanks and sections of the minor system pumping stormwater against topographic gradients may completely change the connectivity determined by topographically driven surface runoff pathways. Our results are for basins with heterogeneous LULC characteristics, where urbanized sectors with varying development rates are mixed with natural LULC patches, typically displaying a distribution of land-development intensities more skewed towards lower values (Fig. 4j, 4k, and 4l), as is common for residential areas. Among the three studied regions, only VA included watersheds with TIA above 50%, but all of those were of small size. Because our dataset may not be representative of large, highly urbanized basins, for these cases (e.g., in countries where cities present generally higher land-development intensities, as compared to the U.S.) we recommend considering the effects of the minor, underground stormwater drainage infrastructure as well, when deriving HCIU. If stormwater sewer data are not available for the study region, and HCIU is estimated only considering topographically induced connectivity, some preliminary testing of its predictive power on gauged basins should be required (e.g., using the validation approach depicted in Fig. 8), before using the index for systematically generating peak-flows in ungauged, highly urbanized watersheds."

4. **I found the explanation and use of figure 1b, for creating a totally impervious copy to be complicated. I think it would be nice if the authors could simplify this text or provide a different graphic to support this process.**
   We improved Fig. 1b accordingly. Additionally, we broke down the methodological steps also in the caption.

5. **I am curious to why the authors evaluate the predictive power of their approach (HCIU) using peak flow only. Why did the authors not consider other important metrics of hydrologic response?**
   In the revised manuscript, we motivate our choice in the Introduction (lines 109-112): "In this work, we derive a lumped metric of urbanization effects on hydrologic response, incorporating only topographically induced connectivity (i.e., neglecting any effects of underground storm sewer infrastructure), and test its performance as a predictor in regional peak-flow equations. Peak flows are among the hydrologic-response variables of greatest interest in urban flooding risk (Feng et al., 2021), and the most important for design purposes (Vogel and Castellarin, 2017)."

6. **With regards to the case studies could you provide what type of flow data was used to generate the flow statistics? Daily, or sub-daily?**

   We clarified this aspect in subsection 3.1, lines 338-341: "For each basin in their case-study regions, Southard (2010), Austin (2014), and Feaster et al. (2014) extracted annual maxima series from instantaneous discharge records (typically with 15-minute or hourly temporal resolution) and performed flood frequency analyses to estimate peak flow values for a range of return periods (see Appendix A), following the U.S. national guidelines provided in Bulletin 17B (Interagency Advisory Committee on Water Data, 1982)."

7. **The authors mention in the acknowledgments the need for computing power, is this a limitation of the approach? It would be nice to know what technical equipment they had access to for completing the analysis. Is this approach feasible for local governments or conservation groups to do?**

   Throughout the duration of the project, we generated increasingly faster code to perform the proposed methodology. For our analyses, we ran the computations in parallel, for multiple watersheds, instead of sequentially processing one basin at a time. To do so, we used the High-Performance Computer at the University of Memphis, allocating different basins to different processors, simultaneously. At the time of manuscript preparation, the computational time per basin varied from coffee-break duration to days, depending on size (with a few days required for basins of thousands of km2). We are developing a newer version of our code, such that computing power will no longer be a limitation and the approach will be feasible for local governments. We expect to cut down the processing times required for the largest basins (in the order of thousands of square kilometers) to a few minutes. Because newer versions of the code will be publicly shared with our future articles, we did not mention information on computational times in the revised manuscript, as these are expected to change soon with future versions of the code.

8. **I think it could be beneficial for the authors to discuss the role of major vs minor system with regards to event size and hydrologic connectivity, especially for the peak flows represented by the extreme flood values.**

   We added considerations about the different roles of major and minor systems of stormwater drainage infrastructure in handling the more frequent, less intense, and the larger, more extreme events. Please find below a list of additional paragraphs and sentences in the revised manuscript.

Lines 115-121 (Introduction): "However, for the scope of our investigation, which focuses on hydrologic response under severe flooding (peak flows with return periods from 2 to 500 years) considering only topographically induced connectivity should be acceptable. This approach allows us to capture the impacts of land development on the surface and near-surface phases of a basin's response, as well as the effects of streams and watercourses, including the artificial ditches and canals that make up the so-called major drainage system of stormwater infrastructure (i.e., excluding the underground network, also known as the minor system; Martins et al., 2017). During severe flooding, it is surface dynamics that predominantly govern hydrologic response, as the underground stormwater infrastructure's capacity is typically exceeded."

Lines 599-602 (Results): "[Referring to Fig. 7] A generalized decrease in model performance is also observed when moving from intermediate to smaller quantiles, except for TIA in VA. This trend may be due to the increasing influence of the minor drainage system on hydrologic response during smaller events, overshadowing surface runoff dynamics. However, both $TIA$ and $HCIU$ primarily focus on aspects related to surface runoff."

Lines 741-758 (Discussion): "Specifically, in its current version, the methodology does not capture the effects of underground stormwater sewer networks (also referred to as the minor system of stormwater infrastructure; Martins et al., 2017), although these are typically present in urban environments, especially in highly developed areas. However, stormwater drainage infrastructure usually includes not only underground pipe networks but also surface flow pathways and canals, which make up the so-called major system. The major system is critical for handling larger, less-frequent storm events. When calculating HCIU, major drainage system sections connected to natural channels are treated as part of the stream network (assuming that excess flow from the major system is poured directly into the stream network). This means that the connectivity of hillslope cells draining to the major system is calculated referring to the pour points along the major system. The contributions of these hillslope cells are then weighted based on the "along-the-stream-network" distance to the outlet, measured starting from the major-system pour point and following both the major-system and any subsequent natural-stream-network links downstream, when averaging the (normalized) connectivities to compute HCIU. This approach captures the effects of the stream network and major drainage systems, which have a stronger influence on the hydrologic response to extreme rainfall events compared to the minor system, whose capacity is typically overwhelmed by large runoff volumes. Consequently, the proposed HCIU should be a more reliable predictor of hydrologic-response variables under severe flooding conditions, as also suggested by the increase in model performance moving from small to intermediate peak quantiles, observed in Fig. 7.

On the other hand, when the analysis focuses on basin response to regular storms (e.g., in water-quality studies), the effects of the minor system should not be neglected, as the underground network may be able to handle most of the (smaller) runoff volumes."

Lines 808-811 (Conclusions): "Depending on the scope of the analysis, an expanded version of the current formulation to account for the additional source of connectivity introduced by underground storm sewer infrastructure may be necessary; in highly urbanized watersheds, the latter may be a stronger control of basin response than topographically induced connectivity, especially in the case of less intense, more frequent events."

9. **Do you think the poor overall performance of the HCIU(CN) could be due to the quality of the raster product you are using?**
While the map of hydrologic soil groups that we considered has the coarsest resolution (250 m, versus 10 m for the DEM and 30 m for the LULC map), it is also the hydrologic variable with the smallest expected spatial variability, therefore we do not consider this to be the main explanation. We suspect that the main issue is that, based on existing CN tables, some developed (i.e. urbanized) cells can get CN values that are similar to those for cells with natural land covers, depending on the type of soil, which does not help in discriminating some of the flood-mitigating effects of such natural conditions.
In the revised manuscript, we consistently kept the same discussion from the previous manuscript version about the overall lower performance of HCIU(CN). However, we will also consider this suggestion in future work about improving HCIU(CN)'s predictive power for peak flows. We thank the Reviewer for providing this additional idea.

10. **Line 560: This is interesting. Please elaborate on this point.**

We expanded that sentence into the following paragraph (lines 650-660): "We suggest that $HCIU$ should also increase our explanatory power when predicting other event-related variables such as lag times and times of concentration. $HCIU$ is indeed sensitive not only to the presence and spatial arrangement of LULC patches with different hydrologic characteristics but also to those locations where flows tend to concentrate, locally decreasing surface runoff travel times, as conceptually reflected in the upslope component $D_{up,k}$ (Eq. 4). $HCIU$ also considers the distance of these surface runoff "hotspots," where stormwater tends to concentrate and travel faster, to the stream network, as reflected by the downslope component $D_{dn,k}$. This in turn determines how easily those locations with accumulating flows will contribute to the overall basin

response. Ultimately, *HCIU* conceptually summarizes in a single number the effects of all potential runoff travel paths occurring on the basin surface, moving towards the stream, including interactions among converging surface flow paths, following a hydrologically driven approach. Because other response variables, such as lag time and time of concentration, are emergent basin properties arising from the interactions of all individual travel paths, their correlations with *HCIU* or other connectivity-based descriptors should be investigated in future research."

**Technical corrections:**

a) **Line 159: I would like to see some examples of LULC types.**

We modified that paragraph as follows (lines 189-195): "Among the options discussed above, when deriving HCIU we recommend choosing $W$ values that primarily depend on the LULC type of each basin cell, considering both developed (urbanized) and more natural (e.g., barren, croplands, forested, etc.) categories, possibly differentiating across distinct intensities of land-development and dominant vegetation types, for the developed and vegetated categories, respectively. In this way, the effects of pixels with different surface characteristics can be differentially weighted depending on their potentials for either generating and quickly transmitting surface runoff (e.g., in the case of developed cells) or else retaining, detaining, or infiltrating water (e.g., in the case of cells with vegetated land cover), depending on the distinct hydrologic dynamics associated with different LULC types."

b) **Line 124: The references are repeated multiple times.**

We eliminated repeated references (lines 155-157): "Borselli et al. (2008) proposed a widely used GIS-based index of connectivity to assess sediment erosion and transport, which was then modified by Cavalli et al. (2013), Persichillo et al. (2018), Zanandrea et al. (2019), Hooke et al. (2021), and Husic & Michalek (2022), among others, to focus on other basin dynamics, such as runoff generation or landslide occurrence."

c) **Figure 1: Suggest expanding on your figure caption to help explain the process in more detail.**

We expanded the caption, breaking down the explanation for each step: "Figure 1: Methodological steps for obtaining the hydrologic-connectivity-based index of urbanization (HCIU): a) scheme for calculating Borselli et al.'s (2008) connectivity index at generic cell $k$; b) create a virtual, totally impervious copy of the basin, with the same shape, topography, and stream network, but different LULC, i.e., fully developed at all cells; c) separately calculate the raster maps of connectivity for both the actual basin and its totally impervious copy; d) calculate the raster map of normalized connectivity for the

basin by dividing the connectivity of the actual basin by the connectivity of the totally impervious copy, on a cell-by-cell basis; e) assign a weight $w_k$ to each basin cell $k$ depending on its distance to the outlet, as measured along the stream network, starting from the cell's pour point; f) calculate $HCIU$ as a weighted average of the normalized connectivities at each basin cell."

**d) Figure 2: What Ecoregion is this?**

In the original manuscript (lines 247-252), we explained in the text that "EPA Ecoregion" is a short name, used in our work, to refer to the hydrologically homogeneous region corresponding to the "Piedmont" and a small part of "Ridge and Valley" EPA ecoregions, consistent with the USGS case-study region from Feaster et al. (2014). In the revised manuscript, we kept the same explanation (now at lines 328-333). In Figure 2, we used that short name for space-related reasons, as well as to be consistent with the rest of the manuscript.

**e) Line 252-254: Suggest adding a reference to Appendix Table A1.**

We specified that TIA information is reported in Appendix Table A1 (lines 333-335 in the revised manuscript): "The VA and EPAE case studies only include basins with at least 10% of TIA (Austin, 2014; Feaster et al., 2014), while the MO study considers a lower threshold, with all basins above 5% TIA, except for one, with only 2.33% (Southard, 2010). TIA values for all case-study basins are reported in Appendix A."

**f) Figure 4: add legend item for shaded blue bars as you have one for the basin averages.**

In the revised manuscript, Fig. 4 now includes a legend item for both types of bars.

**g) Line 329-330: Suggest defining the LULC ranges for the different percentages in the text as you have done in the figure.**

We expanded the sentence (now at lines 412-416) as follows: "Figure 4 also illustrates the mix of developed LULC types in the basins, by showing the distributions (boxplots) of the extents of the four developed NLCD categories in each watershed, for the three homogenous regions (Fig. 4j, 4k, and 4l, respectively). Those categories include "Developed, Open Space", "Developed, Low Intensity", "Developed, Medium Intensity", and "Developed, High Intensity", associated with ranges of impervious area of less than 20%, 20%-49%, 50%-79%, and 80% or more, respectively."

h) **Figure 6: with all the reference lines I found this confusing. Perhaps you could just color code the points you are highlighting based on the outline of the box color.**

We removed the arrows and used letters instead (we also kept the colors), to connect basins to points. Figure 6 in the revised manuscript looks tidier now, after following this suggestion. Thank you.

**Comments made by Referee #2**

1. **Lines 217-230—It is unclear how implementing an "along-the-stream network" differs from the well-known IC_outlet approach from Cavalli et al. (2013) and several other researchers/papers. It is necessary to explain why to chose this new approach over IC_outlet.**

To clarify this aspect and better place our methodology in the context of previous applications of the connectivity index, we added the following text (lines 285-321): "In summary, the proposed methodology provides a lumped metric ($HCIU$) that is able to conceptually capture the varied hydrologic effects arising from the spatial arrangement of different LULC patches, both natural and developed, depending on their relative location with respect to each other, the stream network, and the basin outlet. First, hillslope-to-stream connectivities, weighted depending on the hydrologic effects of distinct LULC types, are normalized with respect to a fully impervious benchmark (Fig. 1a, 1b, 1c, and 1d), which allows to compare the effects of heterogeneous levels of urbanization both across and within basins. Then, $HCIU$ is obtained as a weighted average of normalized connectivities across the entire watershed, assigning different weights to each pixel depending on the "along-the-stream-network" distance of that cell's pour point to the basin outlet (Fig. 1e and 1f).

The proposed two-step formulation – where the flow paths of hillslope cells to the pour points along the stream network and then the distances of those pour points to the basin outlet are considered separately – is different from other established, outlet-focused applications of the connectivity index, such as the $IC\_outlet$ distributed metric proposed by Cavalli et al. (2013). The latter is calculated following Borselli et al. (2008; with some adaptations to the weighting coefficient and the flow direction algorithm) but considering flow paths all the way to the outlet (hence, considering both overland flows and subsequent channelized flows within the same path), instead of flow paths to the closest stream link, following only hillslope surfaces. The two main components of a basin's hydrologic response, i.e., overland and channel flow, generally involve quite different

temporal scales, because of the different orders of magnitude in roughness and water depths. The $IC\_outlet$ metric is able to capture these differences, as $IC\_outlet$ raster maps typically exhibit the highest connectivity values along the watershed stream network (comparable only to connectivities in the hillslope sectors closest to the outlet), followed by connectivities in zero-order valleys or hollows adjacent to channels (Cavalli et al., 2013). On the other hand, the focus of our methodology is on the hydrologic effects of land development, which mostly influences the surface and near-surface components of basin response by locally decreasing infiltration and increasing runoff speeds. Considering only the hillslope-to-stream connectivity in our first step allows us to enhance the method's sensitivity to the effects of land development on hydrologic response, by focusing on how runoff interacts with the distinct LULC patches encountered along the hillslope path, which control (i.e., enhance or mitigate) the connectivity. Once runoff reaches the stream network, the effects of travel distance along the stream network must still be accounted for, but this is performed in the separate, second step, considering a narrower range for the weights. This ensures that $HCIU$ displays adequate sensitivity to urbanized sectors that are adjacent to the stream network, but at reaches located far upstream from the outlet.

Breaking down the calculations for $HCIU$ in two parts (the hillslope-to-stream and then stream-to-outlet flow paths) also presents a practical advantage, particularly for large-scale implementation of the index. To ensure broad applicability of the proposed methodology, we need to be able to quickly compute $HCIU$ for any basin (in a region, country, province, state, etc.), as selected by the final user. If we were to use a "cell-to-outlet" scheme, such as the $IC\_outlet$ metric, we would need to recompute everything from scratch, every time a user chooses a different basin (i.e., a different outlet location along the stream network). Splitting the computations from cell to pour point, and then pour point to outlet, offers the opportunity to precompute "static" (i.e., independent of outlet location) raster maps of connectivity and normalized connectivity, for all the pixels over large areas. In this way, later, when a user selects a specific outlet location, the final computation of $HCIU$ only involves the much-quicker weighted averaging of the precomputed, at-a-cell, normalized connectivities, only considering those cells within the selected basin and their along-the-stream-network distances to its outlet."

**Other minor issues:**

a) **Figure 4: What do the blue bars represent?**
They indicate the proportion (expressed in percent) of cells within a given range of $n$ (or $CN$, or $S$, depending on the considered row in the subplots), with respect to the total number of basin cells for each homogeneous region (also see lines 394-396). For instance, for the VA case study, a little more than 25% of all basin cells (from all basins of that region) have a value of $n$ between 0.7 and 0.8 (Fig. 4c). To address this

as well as a comment from Reviewer 1, we expanded the legend to also include the description of the blue bars, "Proportion of cells", consistent with the associated y-axis label.

b) **No comment exists about how the urban drainage structure could affect urban hydrology.**
To address this as well as comments 2, 3, and 8 from Reviewer 1, we included considerations about how the urban drainage structure could affect urban hydrology. Here are some key passages addressing this, in the revised manuscript:

Lines 104-108 (Introduction): "As the traditional definition of the connectivity index only accounts for topographically induced runoff pathways (Borselli et al., 2008), additional adjustments may be needed, depending on the level of urbanization and the scope of the analysis, to also include the effects of underground stormwater drainage infrastructure, typically present in urban environments. Underground pipe flows may be regarded as an additional source of connectivity, which can alter and sometimes even reverse the connectivity induced by topography (e.g., when stormwaters are pumped against topographic gradients)."

Lines 112-121 (Introduction): "While considering the additional source of connectivity introduced by the underground drainage network would be straightforward, as explained in the Discussion, we could not account for it here, because it was impossible to obtain stormwater sewer data for the hundreds of watersheds involved in our regional scale analyses. However, for the scope of our investigation, which focuses on hydrologic response under severe flooding (peak flows with return periods from 2 to 500 years) considering only topographically induced connectivity should be acceptable. This approach allows us to capture the impacts of land development on the surface and near-surface phases of a basin's response, as well as the effects of streams and watercourses, including the artificial ditches and canals that make up the so-called major drainage system of stormwater infrastructure (i.e., excluding the underground network, also known as the minor system; Martins et al., 2017). During severe flooding, it is surface dynamics that predominantly govern hydrologic response, as the underground stormwater infrastructure's capacity is typically exceeded."

Lines 756-761 (Discussion): "On the other hand, when the analysis focuses on basin response to regular storms (e.g., in water-quality studies), the effects of the minor system should not be neglected, as the underground network may be able to handle most of the (smaller) runoff volumes. Another scenario where it is highly recommended to explicitly consider underground connectivity is when dealing with heavily urbanized watersheds,

typically characterized by the presence of extensive drainage infrastructure. In these basins, detention tanks and sections of the minor system pumping stormwater against topographic gradients may completely change the connectivity determined by topographically driven surface runoff pathways."

Lines 808-811 (Conclusion): "Depending on the scope of the analysis, an expanded version of the current formulation to account for the additional source of connectivity introduced by underground storm sewer infrastructure may be necessary; in highly urbanized watersheds, the latter may be a stronger control of basin response than topographically induced connectivity, especially in the case of less intense, more frequent events."

**Other changes**

1. To address a request from the Editorial Team about ensuring that the color schemes used in our Figures allow readers with color vision deficiencies to correctly interpret our findings, we changed the color map used in Fig. 5c and 5e to a perceptually uniform color map, in the revised manuscript.